



Earth System
Dynamics

# ESD Reviews: Climate feedbacks in the Earth system and prospects for their evaluation

**Christoph Heinze**[1,2], **Veronika Eyring**[3,4], **Pierre Friedlingstein**[5], **Colin Jones**[6], **Yves Balkanski**[7], **William Collins**[8], **Thierry Fichefet**[9], **Shuang Gao**[1,a], **Alex Hall**[10], **Detelina Ivanova**[11,b], **Wolfgang Knorr**[12], **Reto Knutti**[13], **Alexander Löw**[c,†], **Michael Ponater**[3], **Martin G. Schultz**[14], **Michael Schulz**[15], **Pier Siebesma**[16,17], **Joao Teixeira**[18], **George Tselioudis**[19], and **Martin Vancoppenolle**[20]

[1]Geophysical Institute and Bjerknes Centre for Climate Research, University of Bergen, Postboks 7803,
5020 Bergen, Norway
[2]NORCE Norwegian Research Centre, Bergen, Norway
[3]Institut für Physik der Atmosphäre, Deutsches Zentrum für Luft- und Raumfahrt (DLR), Oberpfaffenhofen,
Germany
[4]Institute of Environmental Physics (IUP), University of Bremen, Bremen, Germany
[5]College of Engineering, Mathematics and Physical Sciences, University of Exeter, Exeter, UK
[6]National Centre for Atmospheric Science (NCAS), University of Leeds, Leeds, UK
[7]7Laboratoire des Sciences du Climat et de l'Environnement, CEA-CNRS-UVSQ-UPSaclay, Gif-sur-Yvette,
France
[8]Department of Meteorology, University of Reading, Reading, UK
[9]Université catholique de Louvain, Earth and Life Institute, Georges Lemaître Centre for Earth and Climate
Research, Louvain-la-Neuve, Belgium
[10]Department of Atmospheric and Oceanic Sciences, University of California, Los Angeles, USA
[11]Nansen Environmental and Remote Sensing Center (NERSC), Bergen, Norway
[12]Department of Physical Geography and Ecosystem Science, Lund University, Lund, Sweden
[13]Institute for Atmospheric and Climate Science, ETH Zürich, Zurich, Switzerland
[14]Forschungszentrum Jülich, Jülich, Germany
[15]Norwegian Meteorological Institute, Oslo, Norway
[16]Royal Netherlands Meteorological Institute, De Bilt, the Netherlands
[17]Department of Geoscience & Remote Sensing, Delft University of Technology, Delft, the Netherlands
[18]Jet Propulsion Laboratory, California Institute of Technology, Pasadena, California, USA
[19]NASA Goddard Institute for Space Studies, New York City, USA
[20]Sorbonne Université, Laboratoire d'Océanographie et du Climat, Institut Pierre-Simon Laplace,
CNRS/IRD/MNHN, Paris, France
[a]current address: Institute of Marine Research, Bergen, Norway
[b]current address: Scripps Institution of Oceanography, La Jolla, USA
[c]formerly at: Department for Geography, Ludwig Maximilian University, Munich, Germany
[†]deceased, 2 July 2017

**Correspondence:** Christoph Heinze (christoph.heinze@uib.no)

Received: 21 November 2018 – Discussion started: 5 December 2018
Revised: 10 May 2019 – Accepted: 10 May 2019 – Published:

**Abstract.** TS1 CE1 TS2 Earth system models (ESMs) are key tools for providing climate projections under different scenarios of human-induced forcing. ESMs include a large number of additional processes and feedbacks such as biogeochemical cycles that traditional physical climate models do not consider. Yet, some processes such as cloud dynamics and ecosystem functional response still have fairly high uncertainties. In this article, we present an overview of climate feedbacks for Earth system components currently included in state-of-the-art

ESMs and discuss the challenges to evaluate and quantify them. Uncertainties in feedback quantification arise from the interdependencies of biogeochemical matter fluxes and physical properties, the spatial and temporal heterogeneity of processes, and the lack of long-term continuous observational data to constrain them. We present an outlook for promising approaches that can help to quantify and to constrain CE2 the large number of feedbacks in ESMs in the future. The target group for this article includes generalists with a background in natural sciences and an interest in climate change as well as experts working in interdisciplinary climate research (researchers, lecturers, and students). This study updates and significantly expands upon the last comprehensive overview of climate feedbacks in ESMs, which was produced 15 years ago (NRC, 2003).

## 1   Introduction: the Earth system model dilemma – complexity vs. uncertainty

Anthropogenic emissions of greenhouse gases (GHGs) and aerosols (as well as respective precursor tracers) have altered the radiative balance of the Earth and induce changes in the climate on top of natural variations (IPCC, 2013). International negotiations agreed on keeping the maximum global increase in global mean surface temperatures below $+2\,\mathrm{K}$ relative to pre-industrial levels through reductions in GHG emissions (see discussions in Randalls, 2010, and Knutti et al., 2015), while the signatory countries pledged to make efforts to keep warming below $+1.5\,\mathrm{K}$ (UNFCCC, 2015). In recent years, many atmosphere–ocean general circulation models (AOGCMs; see glossary entry on general circulation models) have been extended to Earth system models (ESMs) that are used to project the extent, characteristics, and timing of climate change under given future scenarios (ENES, 2012). ESMs also contribute to the design of feasible mitigation pathways (e.g. through the computation of allowable emissions in order to achieve a certain climate target; Ciais et al., 2013; Collins et al., 2013). ESMs are climate models CE3 which in addition to physical processes, also simulate a range of relevant biogeochemical cycles (land–biosphere, ocean biogeochemistry, atmospheric chemistry, and aerosols). Special attention in current ESMs is given to the carbon cycle (Bretherton, 1985; Flato, 2011; Jones et al., 2016) (Fig. 1). Compared to conventional, purely physical, CE4 coupled AOGCMs, ESMs include more process representations, variables, and also climate-relevant feedbacks on both short (instantaneous to a few years) and long (decades to centuries to millennia) timescales. ESMs are being continuously expanded to include additional processes. For example, the ESMs which form part of the Coupled Model Intercomparison Project Phase (CMIP) 6 (Eyring et al., 2016a) will for the first time include interactive ice sheets (Nowicki et al., 2016), and several models will have interactive chemistry and aerosols (Collins et al., 2017). Multi-model ensembles of ESM simulations driven by GHG emissions show a larger spread in projections of climate variables (such as surface temperature; see Meehl et al., 2007a) than do physics-only simulations driven by GHG concentrations. This increase in uncertainty is a result of simulating a bigger part of the climate system interactively, including the carbon cycle and atmospheric trace species. Such complex model simulations reveal prevailing deficiencies in our capability to project the evolution of the full Earth system. These deficiencies need to be overcome. How can we assess the quality of the ESM simulations and how might we eventually reduce uncertainties? From observations, we have identified many of the physical and biogeochemical processes operating within the Earth system, yet our understanding of these processes and their interactions on a global scale is still emerging. Observational data are often sparse, and observational time series rarely extend over climate timescales. Many important processes or mechanisms in the Earth system cannot be well constrained through measured parameters. Furthermore, many parameters of known relevance in the Earth system cannot be observed directly. The situation is particularly challenging for feedbacks acting on timescales longer than a decade due to sparse data coverage or a lack of high-quality measurements from the instrumental record. The lack of observational constraints underlines the need for employing models in order to make any useful statement about the future evolution of the climate system at all. This presents challenges concerning the methods and strategies used in assessing ESM performance with respect to the real world.

This article summarizes the major climate-relevant feedbacks to be considered for such an analysis and provides an outlook for constraining feedback in Earth system models. We focus on climatic changes occurring over typical "scenario timescales" (i.e. several hundred years from the pre-industrial period). We thus do not consider, for instance, the long-term effects of ice sheet variations and changes in the land–sea distribution due to sea-level change and tectonics. The goal is to familiarize the reader with the various known major climate feedbacks and to show that there are strategies and tools available for understanding and constraining those feedbacks.

The last major review of climate feedbacks covering several Earth system reservoirs was carried out in 2003 (NRC, 2003). In addition, Bony et al. (2006) provided a review on how well we understand and evaluate climate change feedback processes but focusing on physical feedbacks. Since 2006, considerable progress has been made in Earth system modelling. Similarly to the National Research Council

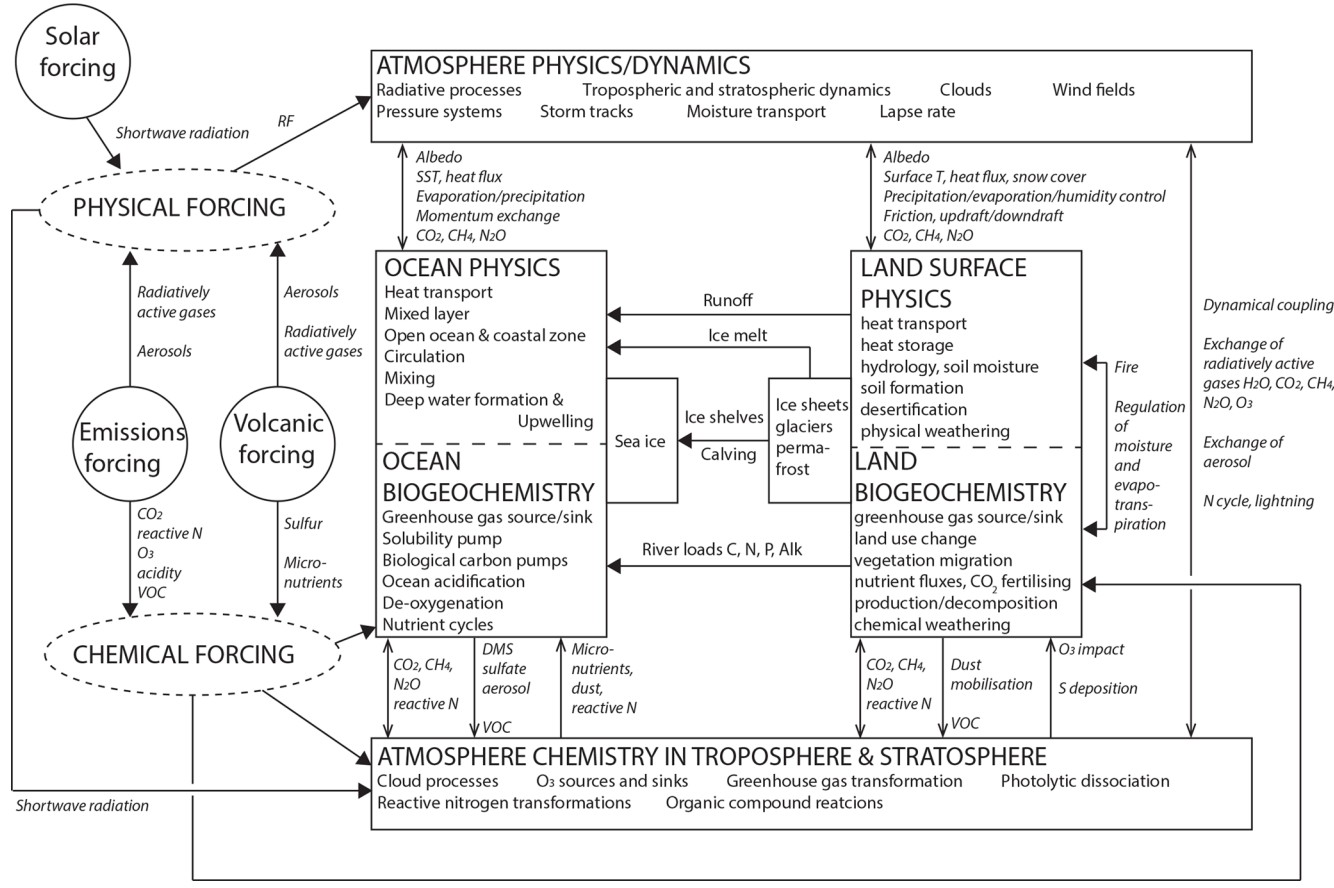

**Figure 1.** The Earth system as an extension of the physical climate system (inspired by Bretherton, 1985 TS3).

(NRC) report (NRC, 2003), the target group of our article is generalists with a background and interest in climate science as well as experts working in interdisciplinary climate research (researchers, lecturers, and students). Compared to this report, we additionally provide feedback diagrams, a more detailed conceptual framework of climate forcings and feedbacks, and an overview of options for evaluating feedbacks. We are aware that, when summarizing climate feedbacks in the Earth system, we must make compromises between comprehensiveness and desirable detail and between instructive conciseness and accounting for inevitable complexity. We have tried to find a feasible balance here. What this article does not aim at is a quantification of Earth system feedbacks and a corresponding uncertainty analysis. Opportunities to address this in detail arise from the experimental design of CMIP6 (Eyring et al., 2016a), which asks how the Earth system responds to forcing as one of three broad scientific questions that are specifically addressed in this phase of CMIP.

## 2 From traditional climate feedbacks to Earth system feedbacks: what is forcing and what is system response?

The external forcing of the climate system is the solar insolation and variations therein (Matthes et al., 2017). Internal forcings (all forcings within the Earth system itself) include human-caused emissions of excess GHGs (Meinshausen et al., 2017) and excess aerosols into the atmosphere due to fossil fuel and biofuel burning and due to industrial, agricultural, and transportation activities. Due to chemical transformations in the atmosphere, it is not only emissions of radiatively active forcing gases or aerosols that must be considered, but emissions of respective precursor gases should be accounted for as well (Hoesly et al., 2018; Lamarque et al., 2010). Further, human-induced land use change (Hurtt et al., 2011) has to be taken into account as it affects many climate-relevant parameters (surface albedo, surface energy budget, hydrological cycle, $CO_2$ respiration and photosynthesis, emission of reactive trace gases, etc.). Internal forcings such as changing atmospheric GHG as well as aerosol concentrations can also be altered by natural processes, for example, as consequences of glacial–interglacial cycles (Lüthi et al., 2008;

Siegenthaler et al., 2005) and volcanic eruptions (Thompson, 1995). Overall, the warming effect due to human-induced GHG and aerosol emissions into the atmosphere leads to ongoing and increasingly positive net radiative forcing in the atmosphere. Aerosol can be of a cooling nature or can have a warming effect depending on the ratio of scattered to absorbed light and the impact of clouds. Anthropogenic driving factors, such as albedo changes from deforestation, agriculture, and urbanization, and perturbations of the nitrogen cycle contribute to the overall forcing of climate change. The climate system reacts to changes in forcing through a response. This response can be amplified or damped through positive or negative feedbacks. We will now briefly describe the term "feedback" in the context of climate and the Earth system and will explain with which reference forcing these feedbacks can be quantified CE5.

## 2.1 Climate sensitivity and feedbacks in a purely physical climate model

Let us first consider a purely physical climate system, where a change in radiative forcing would occur, for example, due to an increase in solar insolation. The so-called climate sensitivity parameter $S$ describes the expected globally averaged equilibrium change in surface temperature $\Delta T_s$ for this given change in globally averaged radiative forcing $\Delta F$ relative to a baseline forcing. In the absence of feedbacks (indicated by subscript and superscript "0"), the climate sensitivity parameter is

$$S_0 = \frac{\Delta T_s^0}{\Delta F}, \tag{1}$$

expressed in kelvin per watts per square metre. In the literature, this term is often expressed as "equilibrium climate sensitivity" (see also Knutti et al., 2017, and Stevens et al., 2016), where a forcing of about $3.7\,\mathrm{W\,m^{-2}}$ is assumed and the sensitivity is therefore given in units of temperature (K). The $3.7\,\mathrm{W\,m^{-2}}$ result from doubling the pre-industrial atmospheric $CO_2$ concentration of 278 ppm around 1750 (parts per million is equivalent to $\mu\mathrm{mol\,mol^{-1}}$). If the Earth were a perfectly absorbing solid sphere without feedback, then $\Delta T_s$ could be computed from the new balance between incoming net shortwave radiation and outgoing thermal radiation according to the Stefan–Boltzmann law (black body radiation). If the colour of the simple Earth were to change with temperature, then $\Delta T_s$ would also depend on the corresponding change in albedo (reflectivity). If the surface temperature changes linearly with changing albedo, then one can add a corresponding correction term $c_1 \cdot \Delta T_s$ to the forcing, thus formally making the overall forcing $\Delta F^* = (\Delta F + c_1 \cdot \Delta T_s)$ a function of the response, while the reference climate sensitivity $S_0$ would not change:

$$S_0 = \frac{\Delta T_s}{\Delta F + c_1 \cdot \Delta T_s}. \tag{2}$$

The total response with this new process is then

$$\Delta T_s = \frac{S_0 \cdot \Delta F}{(1 - c_1 \cdot S_0)}, \tag{3}$$

while the overall climate sensitivity $S$ would be

$$S \equiv \frac{\Delta T_s}{\Delta F} = S_0 \cdot \left( \frac{1}{1 - S_0 \cdot c_1} \right). \tag{4}$$

If $c_1$ were zero, the reference sensitivity and the new overall sensitivity would be identical. If $c_1 > 0$ (reduced albedo), then $S > S_0$, and if $c_1 < 0$ (brightened albedo), then $S < S_0$. In analogy with electrical engineering, $f = S_0 \cdot c_1$ is called *feedback factor*, and the quantity $1/(1 - S_0 \cdot c_1)$ is called *gain* $G$, i.e. the ratio of the new overall climate sensitivity $S$ with respect to the reference sensitivity $S_0$:

$$G = \frac{\Delta T_s}{\Delta T_s^0} = \frac{S}{S_0} = \frac{1}{1 - S_0 \cdot c_1} = \frac{1}{1 - f}. \tag{5}$$

In the climate literature, the terms gain and feedback factor are sometimes used with the opposite meaning in reference to Hansen et al. (1984). Let us assume that the surface of the original Earth was grey. If it were to turn towards black with increasing temperature, both feedback factor and gain would be larger than 1 and $\Delta T_s > \Delta T_s^0$. If it were to turn towards white with increasing temperature, both the feedback factor and gain would be smaller than 1, and $\Delta T_s < \Delta T_s^0$ (see Fig. 2). In the first case the colour-changing process would provide positive feedback (amplifying the temperature change for positive forcing), in the second case the process would cause negative feedback (reducing the temperature change for positive forcing). Likewise, for positive feedback, the climate sensitivity would increase, while for negative feedback, the climate sensitivity would decrease. For the purpose of this paper, climate feedback is defined in this sense, i.e. as a process that changes climate sensitivity (e.g. Manabe and Wetherald, 1967). Other definitions of climate feedback in the literature refer to changes in climate stability, for example, when feedbacks enhance or damp an initial perturbation, and in the statistical mean, extreme weather with greater frequency and/or amplitude – or the opposite – would result from climate change (see discussion in Bates, 2007).

In a more advanced model world of the climate system including atmosphere, ocean, and land surface, more feedback processes would need to be added. The change in radiative forcing may, for example, come from emissions of greenhouse gases and aerosols. From Eqs. (3) and (5), there follows (Hansen et al., 1984; Roe, 2009)

$$\Delta T_s = \frac{S_0 \cdot \Delta F}{1 - S_0 \cdot (c_1 \cdot \Delta T_s + c_2 \cdot \Delta T_s + \ldots + c_n \cdot \Delta T_s)}; \tag{6}$$

$$G = \frac{1}{1 - \sum_{i=1}^{n} f_i}; \, f_i = c_i \cdot \Delta T_s. \tag{7}$$

Equation (7) shows that the feedback factors combine in a linear way, while the gains from the various processes do not.

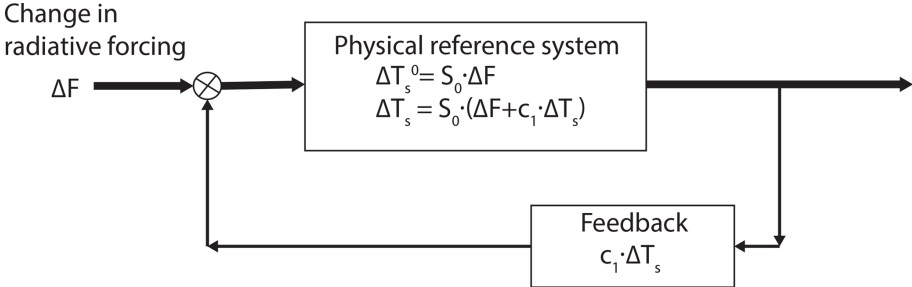

**Figure 2.** Illustration of a climate feedback mechanism. An increase $\Delta F$ in radiative forcing leads to a change in the surface temperature depending on the magnitude and sign of the sensitivity parameter $S_0$. The temperature change $\Delta T_s$ in the presence of feedback is different from the case without feedback ($\Delta T_s^0$). If, for example, the albedo (reflectivity) decreases/increases with rising temperatures (i.e. $c_1 < 0$ or $c_1 > 0$), the surface temperature change will be larger/smaller than in the absence of the feedback.

Therefore, even simple approximations of the climate system with few feedback processes become quite complex when it comes to the quantification of the overall climate sensitivity.

Generally, changes in spatially averaged forcing and spatially averaged surface temperature are time dependent. Because of the large inertia of the climate system (in particular through the ocean), the equilibrium climate sensitivity $S$ denotes the rise in average global surface air temperatures (land and ocean) for a doubling of the atmospheric $CO_2$ concentration (e.g. with respect to the pre-industrial period) after the climate system has reached a new, warmer quasi-steady state after quite a long interval of several thousand years or even longer:

$$S = \frac{\Delta T_s^{t=\infty}}{\Delta F^{CO_2 \times 2 = \text{constant}}}. \tag{8}$$

Firstly, the equilibrium climate sensitivity has been applied to compare results from climate models that include the atmosphere only (while the sea surface temperatures are prescribed) or from atmosphere models coupled to simplified ocean models where the three-dimensional ocean is replaced by a swamp ocean, slab ocean, or mixed-layer ocean. In fully coupled general circulation models of the atmosphere and ocean, the calculation of $S$ becomes a formidable task with long simulation times due to the slow equilibration of the deep ocean. Note that the above framework makes many simplifying assumptions, such as linearity and additivity of forcing and responses and feedbacks independent of the state of the system and the type of forcing; none of these are completely valid in the real world. The carbon cycle and other biogeochemical feedbacks, chemistry feedbacks, and slow feedback-like changes in vegetation types and ice sheets are deliberately not included in the concept of equilibrium climate sensitivity (Knutti and Hegerl, 2008; Knutti and Rugenstein, 2015), which was developed mainly to intercompare the performance of physical climate models consisting of an atmospheric general circulation model (GCM) and a simplified representation of the upper ocean only (for a recent discussion of the equilibrium climate sensitivity, see Stevens

et al., 2016). For example, in an ESM, the ocean would be forced to continuously take up large amounts of $CO_2$ from the atmosphere if the $CO_2$ forcing were to be held constant at double present-day $CO_2$ relative to the pre-industrial period. This would render an unrealistic ocean biogeochemical state after several hundred years leading also to unrealistic fluxes of DMS (dimethyl sulfide, $(CH_3)_2S$, a major contributor to cloud condensation nuclei). The ESM could never be run to a meaningful equilibrium.

## 2.2 Climate sensitivity, transient climate response, and feedbacks in an Earth system model

The *transient climate response* (TCR; unit °C) is an estimate of the global mean surface temperature change in response to $CO_2$ doubling after a prescribed $1\% \, \text{yr}^{-1}$ increase in atmospheric $CO_2$ concentration (where the $CO_2$ doubling is reached after 70 years):

$$\text{TCR} = \Delta T_s^{t=t1}(\Delta F(t)), \tag{9}$$

where for $\Delta F(t)$, a specified threshold is reached at $t = t_1$. This concept is already more aligned to real-world situations. However, the concepts of equilibrium climate sensitivity and TCR are restricted to cases where atmospheric $CO_2$ concentrations (or equivalent $CO_2$ concentrations, i.e. concentrations of CFCs (chlorofluorocarbons) and other greenhouse gases expressed in units of $CO_2$) are the forcing. It is not applicable to emission-driven runs, where biogeochemical cycles interact with the $CO_2$ concentration levels. For the concepts of forcing, feedback, and sensitivity to be useful in an Earth system context, they need to be generalized. We will not look at time-dependent and non-linearly interacting feedback factors here but at state variables other than temperature and forcings other than purely radiative forcing. The concept of sensitivity applies not only to the surface temperature change as a state variable but also to other physical and biogeochemical state variables. To illustrate this, we look at the basic form of an Earth system model, i.e. a physical climate model (atmosphere, ocean, land surface), to which the

carbon cycle has been coupled. Let us assume that human-induced $CO_2$ emissions are the only forcing agent. The emission of $CO_2$ introduces a change in radiative forcing as well as a change in biogeochemical matter cycling – even if there is no change in radiative forcing. Oceanic $CO_2$ uptake and $CO_2$ fertilization of terrestrial vegetation would react to the changing $CO_2$ concentrations in the atmosphere by readjustment of the carbon cycle even without any physical climate change. If we disregard the radiative greenhouse gas forcing for the moment and only focus on $CO_2$-concentration-driven biogeochemical feedbacks, one can define a "material" sensitivity $M_0$ (reference sensitivity without feedback) and $M$ (sensitivity taking into account one or more feedbacks in analogy with Eqs. 1–4):

$$M_0 = \frac{\Delta C_a^0}{\Delta E}; M_0 = \frac{\Delta C_a}{\Delta E + d_1 \cdot \Delta C_a};$$
$$\Delta C_a = \frac{M_0 \cdot \Delta E}{1 - M_0 \cdot d_1}; M \equiv \frac{\Delta C_a}{\Delta E} = M_0 \cdot \frac{1}{1 - M_0 \cdot d_1}, \quad (10)$$

where $\Delta C_a$ and $\Delta C_a^0$ are the atmospheric $CO_2$ concentration change with and without feedback, $\Delta E$ is the change in atmospheric $CO_2$ concentration due to emissions from human activities (cumulated emissions of $CO_2$ since the beginning of industrialization), and $d_1$ is a linear factor changing $\Delta C_a$, such as that due to $CO_2$ fertilization or enhanced respiration. Trivially, $\Delta C_a^0 \equiv \Delta E$ and $M_0 = 1$. The material sensitivity is thus the ratio of the change in atmospheric $CO_2$ concentration (with feedback) to the change in atmospheric $CO_2$ concentration due to emissions (these being expressed as parts per million change in concentration without climate change in the carbon cycle) for a specific time interval, i.e. the change in the *airborne fraction* of $CO_2$ for a given biogeochemical $CO_2$ forcing.

In reality, the physical sensitivity $S$ and the material sensitivity $M$ are not independent. The change in atmospheric $CO_2$ concentration due to biogeochemical feedbacks (chemically forced by $CO_2$), $\delta(\Delta C_a) = \Delta C_a^0 - \Delta C_a$, also causes feedback in the physical system as the greenhouse gas radiative forcing is modified. Such a feedback term $c_2 \cdot \Delta T_s = d^* \cdot \delta(\Delta C_a)$ (see below) can formally be entered into the denominator of Eq. (4) as radiative feedback:

$$S_0 = \frac{\Delta T_s}{\Delta F + c_1 \cdot \Delta T_s + c_2 \cdot \Delta T_s}$$
$$= \frac{\Delta T_s}{\Delta F + c_1 \cdot \Delta T_s + d^* \cdot \delta(\Delta C_a)}. \quad (11)$$

Note that we deviate here from the classical definition of climate sensitivity, which is formulated so that it is independent of biogeochemical feedbacks through the reference value of twice pre-industrial $CO_2$ for change in radiative forcing. The coefficient $d^*$ represents a conversion function for translating the change in $CO_2$ concentration into a modification of surface temperature via an alteration of the radiative forcing ($d^*$

is the combination of modules in ESMs that convert greenhouse gas concentration changes into surface air temperature changes). The overall change in the physical climate system due to greenhouse gas warming, for simplicity represented here by $\Delta T_s$, will additionally feed back to the change in atmospheric $CO_2$ concentration. Alongside other causes, this can occur due to enhanced soil respiration in a warmer world. A respective feedback term can be added to the biogeochemical system:

$$M_0 = \frac{\Delta C_a}{\Delta E + d_1 \cdot \Delta C_a + d_2 \cdot \Delta C_a}$$
$$= \frac{\Delta C_a}{\Delta E + d_1 \cdot \Delta C_a + c^* \cdot \Delta T_s}, \quad (12)$$

where $d_2 \cdot \Delta C_a = c^* \cdot \Delta T_s$. Coefficient $c^*$ includes the temperature-dependent process causing the additional release or storage of $CO_2$ in the biogeochemical system. The couplings of the biogeochemical and physical reference systems as described in Eqs. (11) and (12) are illustrated in Fig. 3. Please note that the coupling of the physical system to the biogeochemical system takes place due to the total response in climate state variables to the entire greenhouse gas forcing, while the coupling of the biogeochemical system to the physical system takes place due to the biogeochemical feedbacks (and not total greenhouse gas forcing). Previdi et al. (2013) suggested expanding upon the term climate sensitivity through the addition of biogeochemical and long-term physical feedback processes to *Earth system sensitivity*, although its practical realization is challenging. In a holistic view, one would have to combine all radiative and concentration feedbacks into one common framework. This can be done formally through extended expressions following Eqs. (11) and (12). Further related sensitivities can be determined for other substances that are involved in the radiative forcing and are also coupled, in parallel, to (bio-)geochemical cycles. This would include the sensitivities of $N_0$ for non-$CO_2$ compounds, $O_o$ for tropospheric ozone including interactions with $CH_4$, and $A_0$ for aerosols including temperature- and carbon-cycle-induced changes in DMS emissions from the oceans. The sensitivity $S_e$ of the Earth system would then be a tensor of different sensitivities, where every component would depend on all others (or, at least, all would be related to thermal sensitivity):

$$\mathbf{S}_e = \{S, M, N, O, A\ldots\}.$$

For model intercomparisons, these sensitivities can be determined at a certain fixed point in time for a prescribed forcing scenario in a similar way as TCR. In practice, however, this is not easy to do due to the multiple interdependencies, the non-equilibrium situation, and the many different timescales involved in feedback processes and forcing agents.

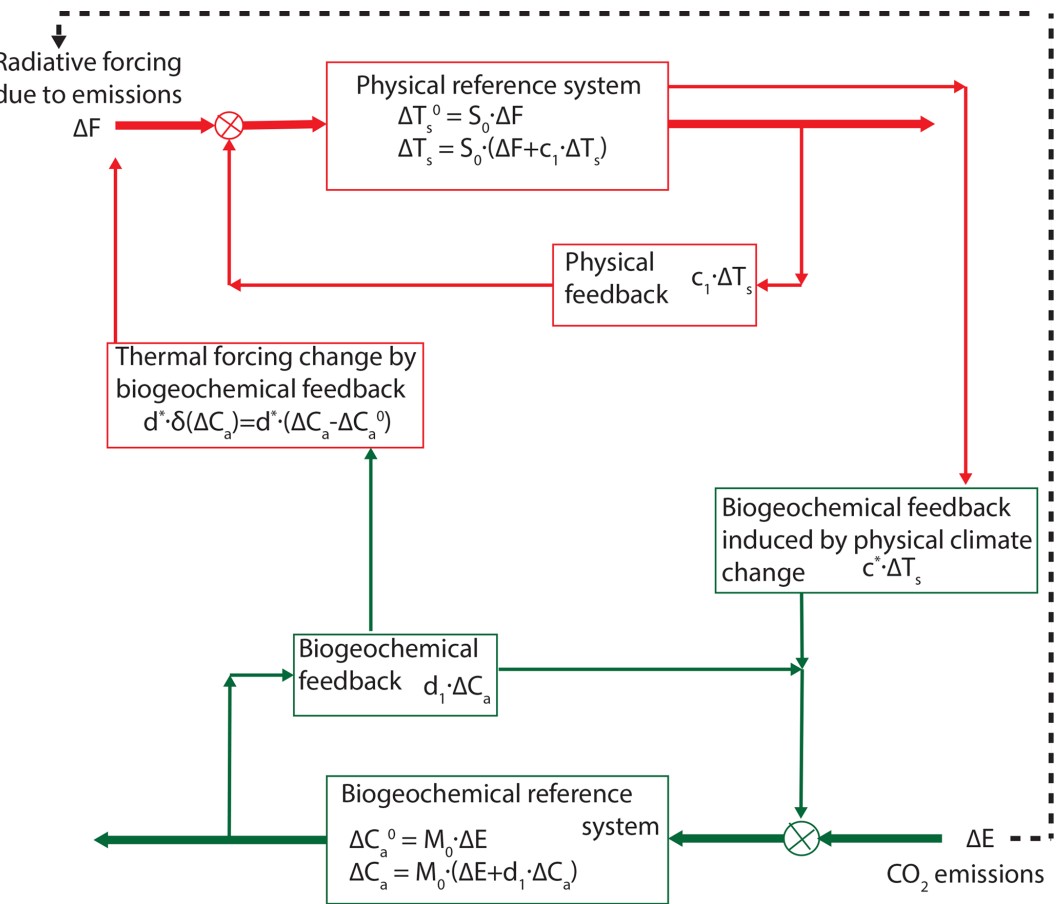

**Figure 3.** The physical (red) and biogeochemical carbon cycle systems (green) within the Earth system are governed by their respective physical and biogeochemical feedback loops and through couplings between the physical and biogeochemical worlds. The coupling of physics to biogeochemistry is induced by the total response of physical climate state variables to the imposed forcing change (represented by $\Delta T_s$), while the coupling of biogeochemistry to physics is induced by the greenhouse gas concentration change $\delta(\Delta C_a)$ caused by biogeochemical feedbacks. The change in greenhouse gas concentration $\Delta C_a$ in the presence of biogeochemical feedback (e.g. the fertilization of plants through higher $CO_2$ levels and respective increased growth) is different to a case without this feedback. $M_0$ is the climate sensitivity parameter for the reaction of the biogeochemical system due to a biogeochemical forcing change. The symbols are explained in more detail in Sects. 2.1 and 2.2. The dashed black arrow illustrates that the $CO_2$ emissions are initially the same for the chemical and the radiative forcing.

## 2.3 Choosing a reference forcing for Earth system feedbacks

When quantifying anthropogenic climate change through the results of multi-model ESM simulations, one has to decide (a) which processes contribute to the additional reference forcing (relative to the unperturbed state) applied to the model systems and (b) which processes contribute to feedbacks amplifying or reducing the response and which ones increase or decrease the actual forcing relative to this reference forcing. Two concepts are currently being used, the classical definition of stratospherically adjusted radiative forcing (RF; see IPCC AR4 Forster et al., 2007) and the more recent definition of *effective radiative forcing* (ERF; see IPCC AR5 Myhre et al., 2013; Sherwood et al., 2015; Fig. 4). The instantaneous radiative flux change induced by a perturbation (Fig. 4a) was discovered to be unsuitable for

providing a sensible reference forcing for the expected climate change (Hansen et al., 2005). Quick adjustments of the stratosphere would start to substantially alter this forcing even before the surface temperature begins to change. Hence, RF (the stratospherically adjusted RF) has been defined as the radiative flux change at the tropopause (see glossary) after the temperature above the tropopause has been allowed to adjust to the changed radiative heating rates under the constraint of fixed dynamic heating rates (Fig. 4b). ERF is also defined as being at the top of the atmosphere (TOA; see glossary) but additionally includes further contributions that are counted as feedbacks under the RF definition (Fig. 4c–d). After an instantaneous addition of a greenhouse gas to the atmosphere, rapid adjustments of various atmospheric variables occur, leading to a further modification of the Earth's radiative budget. RF is computed by keeping tro-

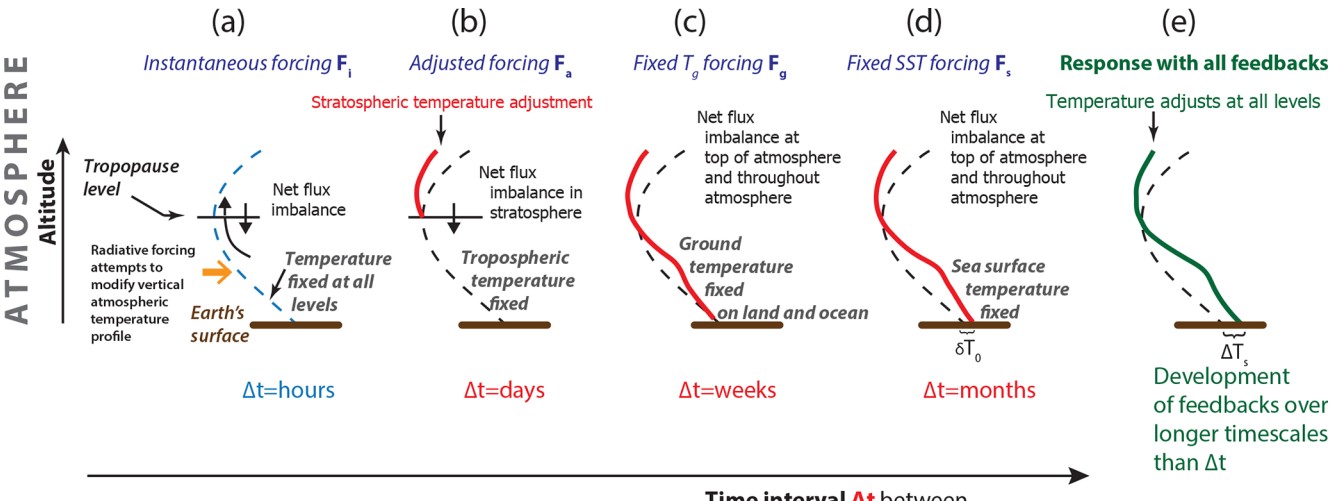

**Figure 4.** Sketch illustrating the different definitions of forcing (drawn anew and extended following Myhre et al., 2013, and Hansen et al., 2005). **(a)** Instantaneous forcing. **(b)** Forcing at tropopause with only stratospheric temperatures adjusted (corresponding to RF). **(c)** Fixed ground temperature forcing. **(d)** Fixed SST forcing including adjustment of land temperature and atmospheric temperature (corresponding to ERF). **(e)** Full response including all feedbacks, also the slow ones.

pospheric temperatures (and also state variables such as water vapour and cloud cover) fixed at their unperturbed profile; ERF is the ensuing radiative forcing once all rapid adjustments for temperature (including the stratospheric domain), water vapour, surface albedo, and clouds are taken into account in response to a change in a forcing agent such as increasing GHG concentrations (Fig. 4c–d). When using ERF, in the optimal case all ground temperature components (of land, ice, and ocean, Fig. 4c) are held fixed at their levels in a reference state, but for pragmatic reasons (e.g. Shine et al., 2003) sometimes only the SSTs (sea surface temperatures) are fixed (Fig. 4d). Feedbacks to ERF are then the radiative flux changes that develop in response to changing sea surface temperature and other slower climatic variables of radiative relevance (Fig. 4e). In IPCC AR5, the following ERF quantification concept was adopted (Myhre et al., 2013):

> we take ERF to mean the method in which sea surface temperatures and sea ice cover are fixed at climatological values unless otherwise specified. Land surface properties (temperature, snow and ice cover and vegetation) are allowed to adjust in this method. Hence ERF includes both the effects of the forcing agent itself and the rapid adjustments to that agent (as does RF, though stratospheric temperature is the only adjustment for the latter).

ERF is model dependent, as it includes a model-specific rapid adjustment simulation. Longer and more complex model simulations are required to quantify ERF than is the case for RF, as the rapid adjustments of clouds and aerosols and their interactions also have to be included (Forster et al.,

2013; Zelinka et al., 2012a). It should also be noted that the fast adjustments differ for the type of forcing even if the total amount of energy added through this forcing does not change. Fast adjustment differences from physical feedbacks have been quantified for solar and $CO_2$ forcings of similar magnitude (Bala et al., 2010) and have also been identified and attributed to various feedback mechanisms for $CH_4$ and aerosol forcings (Smith et al., 2018). An actual procedure for computing ERF is given in Pincus et al. (2016) within RFMIP (Radiative Forcing Model Intercomparison Project). While the ERF approach works for purely physical climate models, it has significant limitations when Earth system models with biogeochemical cycles are employed. In concentration-driven scenarios, the carbon cycle feedback to the climate system can be diagnosed through the respective compatible emissions. These emissions are the emissions necessary to achieve the prescribed atmospheric $CO_2$ concentration trajectory. The lower the compatible emissions, the stronger the underlying positive carbon cycle climate feedback. This is illustrated by the following definition of compatible emissions in a model projection framework using prescribed atmospheric $CO_2$ (see also Box 6.4 in Ciais et al., 2013):

$$\text{Emissions}_{\text{compatible}} = \left( \frac{dCO_2}{dt} \right)^{\text{prescribed}}_{\text{atmosphere}}$$
$$+ (\text{carbon uptake})_{\text{land}} + (\text{carbon uptake})_{\text{ocean}}.$$

For a projection with increasing carbon uptake by land and ocean under rising atmospheric $CO_2$ concentrations, high compatible emissions would result. In a projection with decreasing carbon uptake by land and ocean under rising atmo-

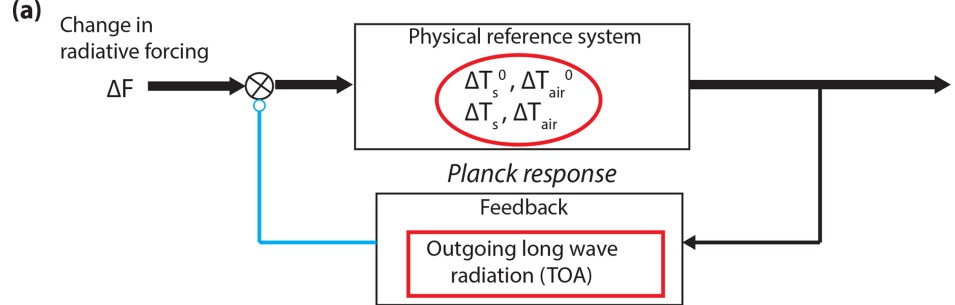

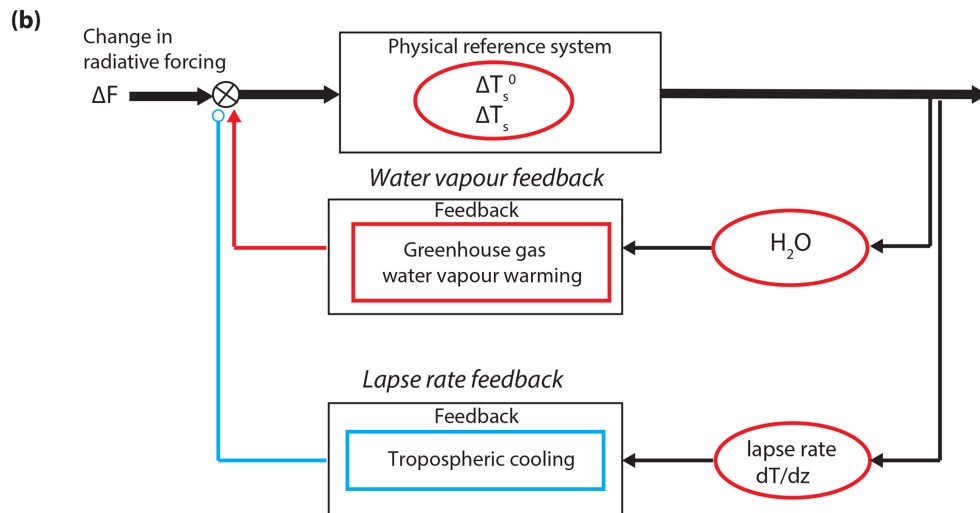

**Figure 5.** **(a)** Negative Planck response feedback. **(b)** Combined water vapour lapse rate feedback. Arrows indicate positive coupling; open circles indicate negative coupling. Changes in state variables are indicated in ellipses. The combined water vapour lapse rate feedback is still positive. Red indicates increasing variable values, strengthening of processes, or positive feedback; blue indicates the opposite. The temperature change $\Delta T$ in the presence of feedback is different from the change $\Delta T^0$ without feedback.

spheric $CO_2$ concentrations, the compatible emissions would be smaller.

The case where the Earth system models would show exactly the same compatible emissions as those emissions that were prescribed in the underlying forcing scenario for the projections could be considered as the reference for positive or negative feedbacks. However, ambiguities already start when splitting up the compatible emissions into contributions from land and ocean with different feedback processes – models could agree in total diagnosed compatible emissions but for different reasons. The most realistic overall experiment set-up for future projections is using emissions-driven forcing. In this framework, there is no suitable reference forcing framework except the pre-industrial situation with no anthropogenic emissions. The entire uptake of carbon by land and ocean would be regarded as a feedback in an emissions-driven Earth system context (as in Friedlingstein et al., 2003, 2006, and Gregory et al., 2009). The feedback of the carbon sink within such a reference framework is by far the most important quantitatively. Alternatively, a refer-

ence land and ocean $CO_2$ uptake pathway could theoretically be defined and only deviations from this "standard" uptake would count as feedbacks; this would be a system response as described in Previdi et al. (2013). Unfortunately, such a standard uptake is not known. The computation of a respective Earth system forcing would also need to take into account fast biogeochemical adjustments including the quasi-instantaneous annual $CO_2$ uptake rates, leaving only longer-term processes such as mixing of carbon into the deep ocean and slow soil processes such as feedbacks for the overall thermal and chemical forcing. As the annual uptake rates differ significantly from model to model, the definition of such a baseline uptake does not look feasible in practice.

## 3   Summary of fast physical climate feedbacks

We will now describe the major feedback processes and the options that currently exist to evaluate them. A general note is appropriate at first. We first briefly summarize the fast physical feedbacks that are already part of conven-

tional physical AOGCMs and then discuss the Earth system feedbacks (Sect. 4) which have been included in climate simulations through the increasing model complexity of ESMs. The following feedbacks will not be considered in detail: (a) ice sheet feedbacks, due to their long timescale (though we will mention the freshwater release from melting glaciers and its impact on ocean circulation) and (b) socio-economic feedbacks (see van Vuuren et al., 2012), as rigorous mechanisms to interpret these are still under development. Table 1 provides a general overview of the most important feedbacks (both short and long term). The feedbacks considered (regardless of whether they are fast or slow) can be grouped into four basic types (please see Table 1, right-hand side): (1) thermodynamic shortwave radiation feedbacks (to a large degree these are the albedo feedbacks), (2) thermodynamic longwave (LW) radiation feedbacks (including dynamics of water vapour and heat redistribution through circulation, though these can also affect shortwave radiation), (3) atmospheric-composition-altering feedbacks due to GHGs (such as $CO_2$, $CH_4$, $N_2O$, and $O_3$, in addition to water vapour, which is already mentioned in (2)CE6), and (4) atmospheric-composition-altering feedbacks involving non-GHGs and particles or droplets (such as $NO_x$ and aerosols). For each family of feedbacks described in the following sections, we provide more details on the respective observational constraints in Appendix A.

Fast feedbacks cover a timescale of months to a few years, where the upper end of the timescale spectrum (few years) would be defined by the mixing timescale of the upper ocean down to the thermocline (of course, equilibration times with the entire deep ocean would also be longer by up to several thousand years). Fast feedbacks are key to decadal climate prediction efforts, while slow feedbacks mainly come into play after a few decades.

## 3.1 Atmospheric thermodynamic feedbacks

The largest fast atmospheric thermodynamic feedbacks are the Planck response (see feedback diagram in Fig. 5a) and the combined water vapour lapse rate feedback (see feedback diagram in Fig. 5b). The Planck response and the water vapour feedback are also considered to be the most certain feedbacks. Cloud feedbacks are also part of the key atmospheric thermodynamic feedbacks. They are discussed separately in Sect. 3.2 because of their complexity. Cloud feedbacks are among the largest contributors to the uncertainty of the total Earth system feedback. Tropical responses of the coupled atmosphere–ocean system to a warming climate are discussed in fast ocean feedbacks (Sect. 3.4).

### 3.1.1 Planck response

A general strong fast negative feedback to surface and tropospheric air temperature warming is the Planck feedback, often referred to as Planck response. The warmer a body

gets, the more energy it radiates (see feedback diagram in Fig. 5a). This feedback has long been understood; it is based on the Stefan–Boltzmann law. For the atmosphere, it is described, for example, in Jonko et al. (2013): "The Planck feedback is the response of LW TOA [longwave at the top of the atmosphere] flux to a perturbation in surface temperature that is applied to each vertical layer of the troposphere." The Planck response is the strongest negative feedback (see quantifications in Bony et al., 2006; Jonko et al., 2013; Soden and Held 2006) and has been found to stabilize the surface temperature response to realistic forcings towards a new equilibrium state. Only if other – positive – feedbacks grew to much larger levels than currently expected could the Planck feedback be overcome, and a runaway greenhouse effect would result. In principle, the Planck feedback could also work in the absence of an atmosphere.

### 3.1.2 The combined water vapour lapse rate feedback

In a warmer world, the atmosphere is expected to hold more water vapour, which is itself an important greenhouse gas. The strongly positive water vapour feedback is defined as the response of column-integrated atmospheric moisture to changes in climate resulting from an external perturbation in radiative forcing (see feedback diagram in Fig. 5b). For example, when the tropical ocean warms as a result of a $CO_2$-induced increase in downwelling LW radiation, the Clausius–Clapeyron relationship (see glossary) leads to an increased ability of the atmosphere to carry water vapour that evaporated from the ocean (Bohren and Albrecht, 1998). As water vapour absorbs radiation across a large part of the infrared spectrum (Tipping and Ma, 1995), increased water vapour leads to increased atmospheric absorption of surface-emitted radiation, a reduction in outgoing LW radiation (OLR), and an increase in downwelling LW radiation to the surface. If atmospheric relative humidity remains constant when temperature increases, as suggested by observations and models, then the water vapour feedback approximately doubles the Earth's equilibrium climate sensitivity to a doubling of $CO_2$ concentrations (relative to a theoretical no-feedback case) (Manabe and Wetherald, 1967).

The moist adiabatic lapse rate is the vertical gradient of tropospheric temperature with altitude (due to vertical pressure changes and taking condensation or freezing into account). As the moist adiabatic lapse rate decreases with increasing surface temperature, the first-order effect of a *lapse rate feedback* is expected to be negative (Cess, 1975; Wetherald and Manabe, 1986) (see feedback diagram in Fig. 5b). Often, the addition of a longwave absorber tends to cool the atmosphere but warms the surface, thus increasing the vertical lapse rate. This is balanced by convection (see glossary) stabilizing the atmosphere back towards a moist adiabatic profile. Especially in tropical regions, a stronger warming of the troposphere as compared to the surface occurs under increased greenhouse gas concentrations in the atmosphere.

**Table 1.** Classification of specific feedbacks (left vertical column) with respect to general "archetypes" of feedbacks. Feedbacks can be summarized as thermodynamic feedbacks and composition-altering feedbacks. Aerosol feedbacks are among the most complex feedbacks. The numbers in front of the specific feedbacks refer to the headers or sub-headers of the respective sections in the text. `CE7`

| | Thermal SW - reflectivity/albedo | Thermal LW - heat re-distribution including water vapor and moisture | Atmospheric composition - greenhouse gases without water vapor | Atmospheric composition - non-GHG and particles |
|---|---|---|---|---|
| **Fast physical climate feedbacks** | | | | |
| *3.1 Atmospheric thermodynamic feedbacks* | | | | |
| 3.1.1 Planck response | | ■ | | |
| 3.1.2 Combined water vapour lapse rate feedback | | ■ | | |
| *3.2 Cloud feedbacks* | | | | |
| 3.2.1 Rise of cloud tops feedback | | ■ | | |
| 3.2.2 Tropical low cloud feedback | ■ | ■ | | |
| 3.2.3 Mid-latitude cloud reflectance feedback | ■ | | | |
| 3.2.4 Cloud water phase feedback | ■ | | | |
| *3.3 Fast land surface feedbacks* | | | | |
| 3.3.1 Snow albedo feedback | ■ | | | |
| 3.3.2 Soil moisture evapotranspiration feedback and $CO_2$–stomata–water feedback | | ■ | | |
| *3.4 Fast ocean feedbacks* | | | | |
| 3.4.1 Fast ocean feedbacks: ocean mixed layer and ocean thermocline feedbacks | | ■ | | |
| 3.4.2 Tropical circulation responses to a warming climate | | ■ | | |
| *3.5 Sea ice feedbacks* | | | | |
| 3.5.1 Sea ice albedo feedback | ■ | | | |
| 3.5.2 Sea ice negative feedbacks | | ■ | | |
| **Earth system feedbacks** | | | | |
| *4.1 Slow vegetation-land-surface-climate feedbacks* | | | | |
| 4.1.1 Vegetation-snow-masking feedback | ■ | | | |
| 4.1.2 Vegetation-evapotranspiration-albedo feedback | ■ | ■ | | |
| *4.2 Slow ocean-atmosphere feedbacks* | | | | |
| 4.2.1 Ocean overturning feedbacks | | ■ | | |
| *4.3 Land biogeochemistry feedbacks* | | | | |
| 4.3.1 Feedback between net ecosystem productivity, climate change, and rising $CO_2$ | | ■ | ■ | |
| 4.3.2 Feedback between climate change and $CO_2$ emissions from fires | | ■ | ■ | ■ |
| *4.4 Marine biogeochemical feedbacks* | | | | |
| 4.4.1 Inorganic ocean carbon cycle feedbacks due to changes in carbon chemistry | | ■ | ■ | |
| 4.4.2 Feedbacks due to changes in marine ecosystems | | ■ | ■ | |
| *4.5 Aerosols-climate feedbacks* | | | | |
| 4.5.1 Feedbacks between marine aerosol emissions and climate change | ■ | ■ | | ■ |
| 4.5.2 Feedbacks between dust mobilization and climate, including fertilizing effects | ■ | ■ | | ■ |
| 4.5.3 Secondary aerosol feedbacks | ■ | ■ | ■ | ■ |
| 4.5.4 Aerosol-cloud feedbacks | ■ | ■ | ■ | ■ |
| *4.6 Tropospheric gas-phase chemistry feedbacks* | | | | |
| 4.6.1 Feedback through changes to chemical reaction rates | | ■ | ■ | ■ |
| 4.6.2 Feedbacks between natural emissions and climate change | | ■ | ■ | ■ |
| *4.7 Stratospheric composition feedbacks* | | | | |
| 4.7.1 Stratospheric ozone feedback | | ■ | ■ | |
| 4.7.2 Stratospheric water vapour feedback | | ■ | | |

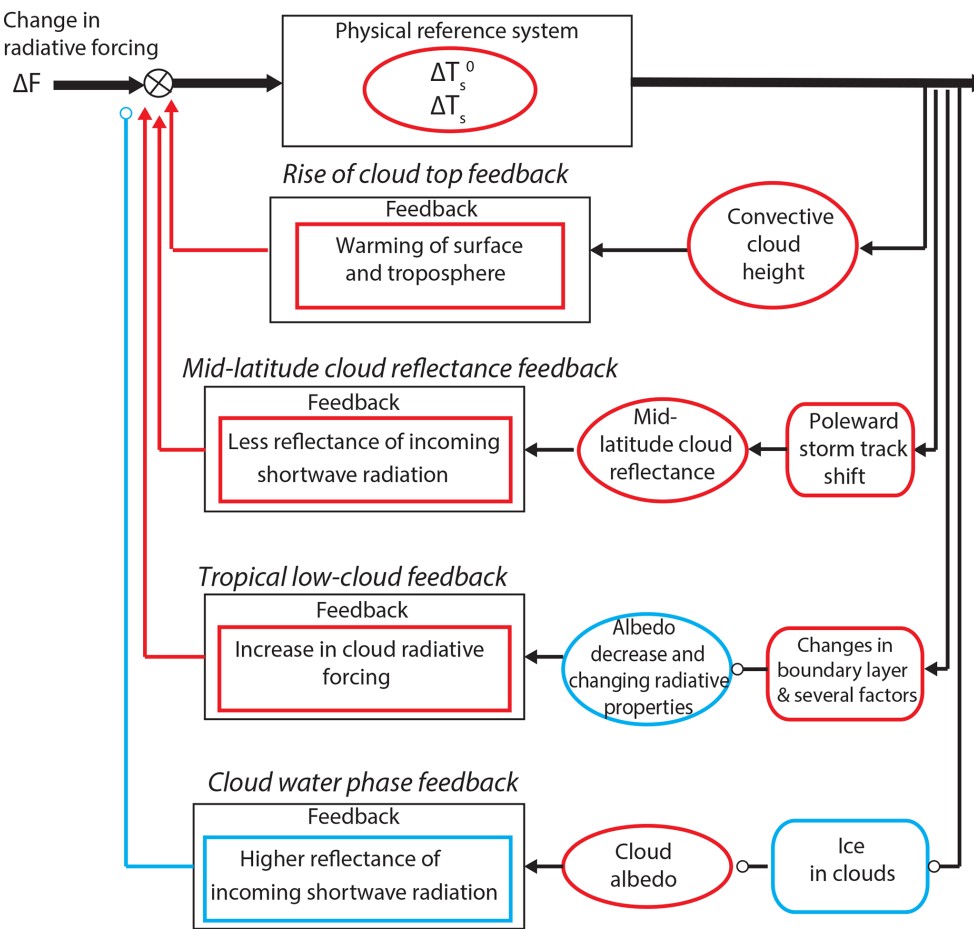

**Figure 6.** Schematic of important cloud feedbacks. Arrows indicate positive coupling; open circles indicate negative coupling. Changes in state variables are indicated by ellipses; changes in processes by rounded rectangles. Red indicates increasing values of variables, strengthening of processes, or positive feedbacks; blue indicates the opposite. The change in surface air temperature $\Delta T_s$ in the presence of feedback is different from the change $\Delta T_s^0$ without feedback.

This effect results in a negative feedback to climate due to an increase in thermal emission to space (Boucher et al., 2013; Bony et al., 2006).

The offsetting nature of the water vapour and lapse rate feedbacks has long been understood (Cess, 1975) though the details of their origin are still a current research topic (Po-Chedley et al., 2018). Using the radiative kernel of the NCAR (National Center for Atmospheric Research) model, Vial et al. (2013) and Caldwell et al. (2016) estimate a positive multi-model mean water vapour feedback of $+1.71 \, \mathrm{Wm^{-2} \, {}^\circ C^{-1}}$ (standard deviation of 0.13) and a negative lapse rate feedback of $-0.66 \, \mathrm{Wm^{-2} \, {}^\circ C^{-1}}$ (standard deviation of 0.17), leading to a combined mean positive water vapour lapse rate feedback of $+1.05 \, \mathrm{Wm^{-2} \, {}^\circ C^{-1}}$. These numbers compare well with the previous estimate of Soden and Held (2006). Multi-model comparison experience emphasizes the interdependent and generally offsetting nature of the water vapour and lapse rate feedbacks in GCMs; e.g. models with large upper tropical troposphere (UTT) warm-

ing (negative lapse rate feedback) generally also have high UTT moistening (positive water vapour feedback) and vice versa.

## 3.2    Cloud feedbacks

Clouds have a strong effect on the Earth's present-day top-of-atmosphere radiation budget as can be inferred from satellite data by comparing upwelling radiation in cloudy and non-cloudy conditions (e.g. Ramanathan et al., 1989). Since cloud albedo is in general much larger than the albedo of the underlying surface, cloudy conditions exert a global annual shortwave radiative cooling effect (SWCRE) of close to $-50 \, \mathrm{Wm^{-2}}$. On the other hand, the atmosphere emits less outgoing longwave radiation under cloudy conditions than under cloud-free conditions. Predominantly due to high-altitude clouds, this results in a longwave cloud radiative effect (LWCRE) of approximately $+30 \, \mathrm{Wm^{-2}}$ (Loeb et al., 2009). Therefore, clouds have a strong net cooling effect

on the current climate, associated with a net global mean CRE (cloud radiative effect) of $-20\,\mathrm{Wm}^{-2}$. Changes in the cloud albedo can occur through changes in cloud amount but also through changes in cloud opacity, which in turn depend on the cloud optical thickness or more precisely on the cloud condensation mass, on the phase of condensed water, and on the cloud droplet number concentration. Changes in the LWCRE can occur mainly through changes in cloud top height but also through changes in cloud amounts, particularly at middle and high levels.

Lacking understanding of cloud processes and difficulties in simulating cloud feedbacks are CE8 the prime sources of uncertainty in climate sensitivity estimates and have been so for a few decades (e.g. Charney, 1979; Sherwood et al., 2014; Stevens et al., 2016). Progress has been slow because the grid resolutions of typical GCMs are too coarse to resolve the fundamental physical processes that control clouds. Essential physical processes such as turbulence, moist convection, cloud macrophysics and microphysics, and the interaction with the land and ocean surfaces (for example through evapotranspiration and evaporation) must be parameterized. These parameterizations still have deficiencies that lead to a large spread between the models. Recently, however, more realistic and unified parameterizations of turbulence, convection, and clouds have started to be successfully implemented in GCMs. Using simulations with more recent cloud parameterizations, and based on high-resolution cloud models and observations, it has been concluded that "the net radiative feedback due to all cloud types is judged likely to be positive" (IPCC, 2013). The overall cloud feedback strength from multi-model averaging has been determined to be $+0.6\,\mathrm{Wm}^{-2}\,^{\circ}\mathrm{C}^{-1}$, with a large range of uncertainty between $-0.2$ and $+2.0\,\mathrm{Wm}^{-2}\,^{\circ}\mathrm{C}^{-1}$ based on large inter-model spread as well as on additional processes that are not included in GCMs (Boucher et al., 2013). Progress in understanding the reasons for these positive cloud feedbacks has been achieved due to a combination of model analysis techniques, hypothesis testing efforts using cloud-resolving models (CRMs) and large-eddy simulation (LES) models (e.g. Bretherton, 2015; Tonttila et al., 2017), and observations. Of the greenhouse warming cloud feedbacks that are supported by multiple GCMs and that can be understood by physical reasoning and using observations, the one that appears the most robust is that which includes the rise of high clouds and the melting layer at all latitudes leading to positive longwave radiation feedback. The increase in the mid-latitude cloud amount resulting in positive feedback, the expected positive tropical low-cloud feedback and the negative cloud water phase feedback are more uncertain (see feedback diagram in Fig. 6).

### 3.2.1 Rise of cloud top feedback

The most robust and well-understood positive cloud response is the longwave cloud feedback related to an upward shift in cloud height. This contribution can be largely attributed to the so-called "fixed anvil temperature" (FAT) mechanism (Hartmann and Larson, 2002), which states that the outflow level of deep convective clouds occurs at a fixed temperature as the climate warms. This leads to a rise of high clouds associated with deep convection and a corresponding positive cloud longwave feedback. The clouds are not warming synchronously with the surface temperature. Therefore, the warming tropics become less efficient at radiating away heat. As a consequence, the clouds induce a positive feedback to climate (Zelinka and Hartmann, 2011). It is expected that this mechanism occurs at all latitudes and has been estimated to give rise to a positive longwave feedback of $+0.2\,\mathrm{Wm}^{-2}\,^{\circ}\mathrm{C}^{-1}$ (Zelinka et al., 2016), explaining roughly half of the mean cloud feedback of GCMs.

### 3.2.2 Tropical low-cloud feedback

Climate models tend to produce a widespread positive low-cloud feedback, causing most of the overall spread in climate sensitivities among GCMs (Klein et al., 2017; Sherwood et al., 2014; Vial et al., 2013; Zelinka et al., 2016). The low-cloud feedback from GCMs participating in CMIP5 ranges from $-0.09$ to $0.63\,\mathrm{Wm}^{-2}\,^{\circ}\mathrm{C}^{-1}$ (Webb et al., 2013) with a mean of $+0.35\,\mathrm{Wm}^{-2}\,^{\circ}\mathrm{C}^{-1}$ (Zelinka et al., 2016). This spread is largely attributable to the representation of marine stratocumulus and shallow cumulus clouds and the transitions between them (Williams and Webb, 2009; Xu et al., 2010). Recent studies using LES models (which capture the physics of these boundary-layer clouds in a realistic manner) have provided a deeper understanding of and helped isolate key mechanisms behind low-cloud feedbacks (Bretherton, 2015). Changes in reflective properties, humidity, and convection contribute as well (Medeiros et al., 2008; Qu et al., 2015; Zhang et al., 2013). Also, total cloud amount observations in combination with model results support a positive, temperature-driven low-level cloud feedback (Clement et al., 2010; Klein et al., 2017; Myers and Norris, 2016; Qu et al., 2015). While the overall confidence in the tropical low-cloud feedback had been generally low (Grise and Medeiros, 2016), it has been increased by the recent observational and high-resolution modelling results (Klein et al., 2017).

### 3.2.3 Mid-latitude cloud reflectance feedback

Several observational studies (Bender et al., 2012; Eastman and Warren, 2013) have reported poleward shifts in the mid-latitude cloud field over the past 40 years. These studies attribute the poleward cloud shifts to mid-latitude jet (see glossary) shifts, even though later studies have shown that they are more strongly related to the expansion of the Hadley cell (Tselioudis et al., 2016). Similar poleward jet and cloud shifts are also simulated by most GCMs, although with a weaker strength than observed (e.g. Yin, 2005). These shifts in optically thick storm clouds to higher latitudes with

weaker incoming solar radiation make them less efficient radiation reflectors and thus induce a positive feedback of an uncertain amount and of medium confidence in AR5 models (Boucher et al., 2013) but have subsequently been shown to vary greatly by ocean basin, season, and model (e.g. Grise and Polvani, 2014; Kay et al., 2014), thus reducing the level of confidence in the feedback. The misrepresentation of extratropical low-level clouds in models also has implications for the cloud water phase feedback (see following section).

### 3.2.4    Cloud water phase feedback

Greenhouse warming will cause an elevation of the melting level and of liquid cloud water at the expense of the amount of ice in clouds (Senior and Mitchell, 1993; Tsushima et al., 2006). The resulting poleward shift in the freezing isotherm and consequent change in cloud phase is expected to induce negative cloud water phase feedback (e.g. Mitchell et al., 1989; Wall and Hartmann, 2015). Due to the larger reflectivity of liquid water clouds over ice clouds (cloud cover and water mass unchanged), a change from ice to liquid clouds must induce a negative (shortwave) cloud radiative feedback (Tan et al., 2016). Extratropical low-level clouds are often misrepresented in Earth system models, contributing to the double intertropical convergence zone problem and to shortwave radiation biases (too much heating of the Southern Ocean) (Hwang and Frierson, 2013). A correction of this bias is likely to decrease the negative cloud water phase feedback (and introduces a positive low-cloud feedback that is similar in mechanism to our tropical low-cloud feedback in Sect. 3.2.2) (Frey and Kay, 2018). This misrepresentation of extratropical low-level clouds reduces the confidence in the magnitude of the feedback.

### 3.3    Fast land surface feedbacks

We will now address how the lower boundary condition of the atmosphere influences climate. Climate and land use changes together are expected to strongly influence the state of land surfaces and land–atmosphere interactions affecting the surface energy, water, and carbon fluxes (snow cover, surface albedo, land cover, soil moisture, turbulent fluxes, and growing season) (Davin et al., 2007; Pitman et al., 2009; Seneviratne et al., 2006, 2013). Slow physical land surface feedbacks are discussed in Sect. 4.1 further below. The most important fast land surface feedbacks are that of snow albedo, the positive soil moisture evapotranspiration feedback, and the positive $CO_2$–stomata–water feedback (see feedback diagrams in Fig. 7). The latter is not a purely physical feedback due to the involvement of vegetation, but it is nevertheless discussed here in the context of the hydrological or thermodynamic feedbacks (the land biogeochemistry feedbacks involving the carbon cycle are discussed in Sect. 4.3).

### 3.3.1    Snow albedo feedback

Snow cover is projected to decrease in a warmer climate. Since snow is generally more reflective of sunshine than the underlying land surface, this will cause an increase in the net incoming solar radiation flux. Moreover, the reflectivity of the remaining snow is also projected to decrease due to the fact that highly reflective fresh snow (Robock and Kaiser, 1985; Wiscombe and Warren, 1980) occurs less frequently in a warmer climate, and less reflective melting snow (Robock, 1980) is more common. The snow albedo feedback is therefore positive in all past and current climate models (mean of ca. $0.08\,\mathrm{W\,m^{-2}\,^\circ C^{-1}}$). In spite of this consensus on the sign of the feedback, there is a nearly 5-fold spread in its strength (Qu and Hall, 2014) with significant consequences for the magnitude of climate change in Eurasia and North America (Hall et al., 2008) and for Northern Hemisphere atmospheric circulation (Fletcher et al., 2009). The spread in estimated snow albedo feedback was also not reduced between the models used in CMIP3 and CMIP5 (Qu and Hall, 2007, 2014).

### 3.3.2    Soil moisture evapotranspiration feedback and $CO_2$–stomata–water feedback

Warming leads to an increase in evaporation from soils. This negative soil moisture anomaly leads to a positive surface temperature anomaly through the reduction in latent heat flux (Seneviratne et al., 2010). The result is a positive feedback. In addition to this physical feedback, there is chemically forced feedback. Under rising atmospheric $CO_2$ concentrations, plants open their stomata (plant stomata; see glossary) less widely (Farquhar et al., 1980; Woodward, 1987) (see Sect. 4.3). This leads to a reduction in evapotranspiration over land, a decrease in latent heat flux, and respective warming. This overall positive feedback is somewhat reduced by a secondary negative feedback: $CO_2$ fertilization (see Sect. 4.3) will lead to an increase in carbon assimilation and a respective increase in LAI (leaf area index – area covered by leaf canopy in relation to ground area) and a slight increase in surface albedo (Willeit et al., 2014). An uncertainty associated with this feedback is the original underlying surface albedo (if this were high, then the feedback could even become reversed).

### 3.4    Fast ocean feedbacks

The ocean differs strongly from the atmosphere with respect to fundamental physical properties such as heat capacity, viscosity, and timescales of motion. The progress of climate change depends critically on the penetration rate of the global warming signal into the ocean and the capacity of the ocean to uptake heat from the atmosphere. Ocean–climate feedback timescales range from the synoptic to seasonal, decadal, or even centennial. The transient short-term ocean heat uptake feedback is negative. This holds for the ocean surface

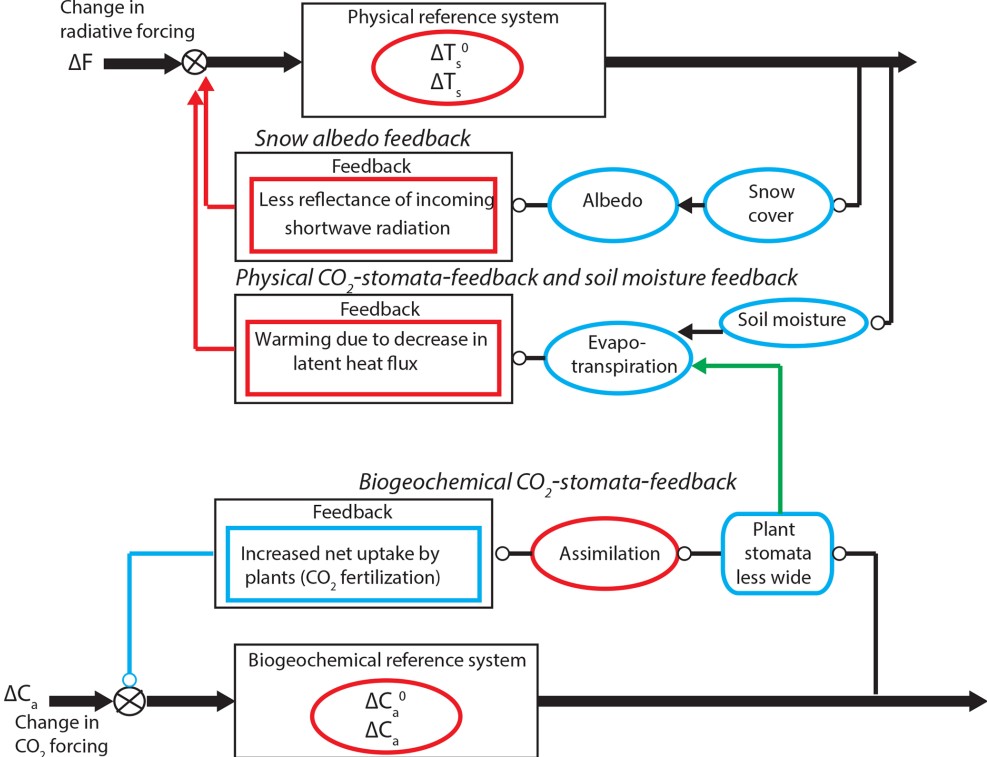

**Figure 7.** Schematic of important fast land surface feedbacks. Arrows indicate positive coupling; open circles indicate negative coupling. Changes in state variables are indicated by ellipses, changes in processes by rounded rectangles. Red indicates increasing values of variables, strengthening of processes, or positive feedbacks; blue indicates the opposite. Green arrows and lines indicate couplings of biogeochemical processes to physical processes. The change in surface air temperature $\Delta T_s$ in the presence of feedback is different from the change $\Delta T_s^0$ without feedback (the change in greenhouse gas concentration $\Delta C_a$ in the presence of a biogeochemical feedback is different from that without such a feedback, $\Delta C_a^0$).

feedback as well as the thermocline-related feedbacks of the tropical coupled ocean–atmosphere system (see feedback diagrams in Fig. 8). Slow ocean feedbacks are discussed in Sect. 4.2.

### 3.4.1 Fast ocean feedbacks: ocean mixed-layer feedbacks and ocean thermocline feedbacks

Heat and momentum fluxes alter the ocean mixed-layer depths, which in turn affect sea surface temperatures with implications for atmospheric circulation and precipitation patterns in the tropics (Bernie et al., 2007, 2008). The transient short-term ocean heat uptake feedback is negative; i.e. the ocean has a cooling effect on the air temperatures in the troposphere (Rose et al., 2014) (Fig. 8). One attempt to combine some of the complex dynamical–thermodynamical processes into a simple feedback-type relationship was that of Haney (1971), who derived a relationship between ocean surface heat flux and surface temperature in the form $Q = \kappa(T_a - T_o)$, where $Q$ is the downward heat flux from atmosphere into ocean, $T_a$ is a representative atmosphere temperature, and $T_o$ is a measure of ocean surface temperature. Based on observations available at the time, Haney computed an average

"coupling coefficient", $\kappa$, with a value of 30–45 $\mathrm{W\,m^{-2}\,°C^{-1}}$. Subsequent work by Chu et al. (1998) suggests a value of 65–70 $\mathrm{W\,m^{-2}\,°C^{-1}}$, whereas Frankignoul et al. (1998) suggest a value of 20 $\mathrm{W\,m^{-2}\,°C^{-1}}$. Frankignoul and Kestenare (2002) further quantified the oceanic feedback to the anomalous surface heat flux by explicitly separating the surface heat flux anomaly into atmospheric and oceanic forced components: $Q' = q' - k \cdot T_o'$, where $q'$ is the anomalous atmospheric heat flux and $-k \cdot T_o'$ is the anomalous heat flux induced by sea surface temperature anomaly. Using COADS (Comprehensive Ocean–Atmosphere Data Set) ship-based observational data and NCEP reanalysis data, the authors estimated the heat flux feedback $\kappa$ to be in the range of 10 to 35 $\mathrm{W\,m^{-2}\,°C^{-1}}$. This estimate was refined by Park et al. (2005) by exploiting ship-derived observations EECRA (Extended Edited Cloud Report Archive; Hahn and Warren, 1999) together with a satellite-derived data set of turbulent fluxes (ISCCP, Zhang et al., 2004). They obtained seasonal ranges for the North Pacific of between 5 and 28 $\mathrm{W\,m^{-2}\,°C^{-1}}$ and ranges for the North Atlantic of between 9 and 33 $\mathrm{W\,m^{-2}\,°C^{-1}}$. We will now look at how the basic ocean processes are integrated into the coupled climate framework.

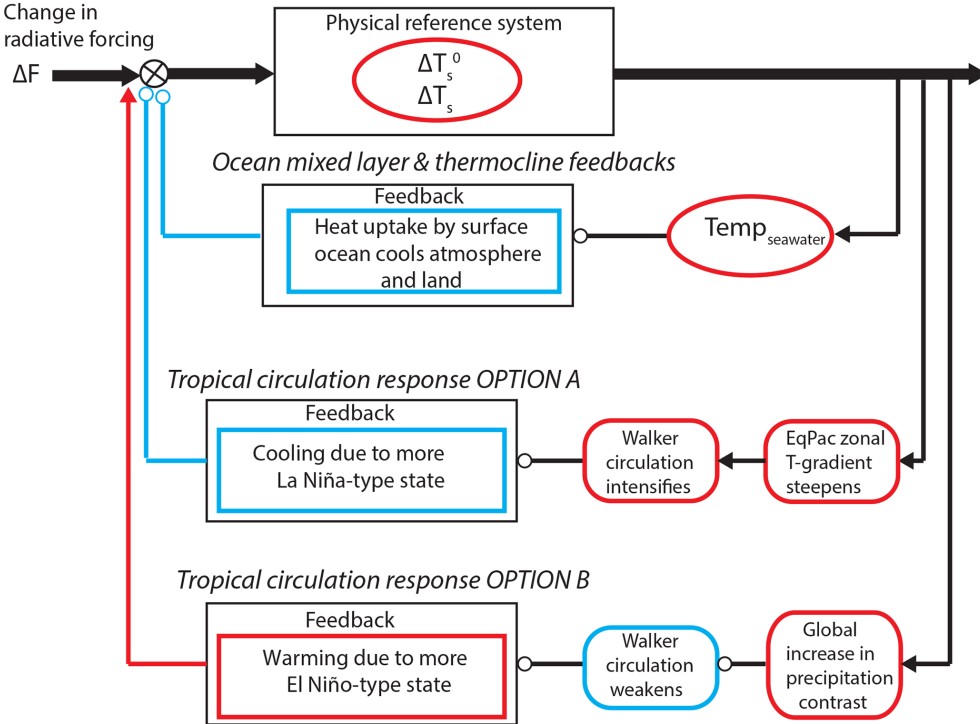

**Figure 8.** Schematic of important fast ocean feedbacks. The sign of the tropical circulation feedback has not been clarified yet. Arrows indicate positive coupling; open circles indicate negative coupling. Changes in state variables are indicated by ellipses, changes in processes by rounded rectangles. Red indicates increasing values of variables, strengthening of processes, or positive feedbacks; blue indicates the opposite. The change in surface air temperature $\Delta T_s$ in the presence of feedback is different from the change $\Delta T_s^0$ without feedback.

### 3.4.2 Tropical circulation responses to a warming climate

The transient response of the tropical circulation to an increase in atmospheric GHG concentrations remains an unresolved issue. Two possible opposing responses are as follows. (1) The Walker circulation (see glossary) intensifies and the tropical Pacific shifts to a more La Niña-like (see glossary) mean state with a relatively cold tropical surface ocean, hence causing negative feedback (Fig. 8). This arises primarily through the ocean thermostat mechanism (Cane et al., 1997; Clement et al., 1996), whereby heating leads to a steepening of the zonal temperature gradient across the equatorial Pacific, due to warming in the east being partially balanced by oceanic upwelling, while west Pacific warming follows a surface thermodynamic response. The increased temperature gradient induces stronger surface easterly winds, further upwelling and cooling in the equatorial east or central Pacific, and a strengthening of the temperature gradient. Vecchi et al. (2008) show that coupled models, with simplified atmospheric dynamics, exhibit an ocean thermostat response and a strengthening of the Walker circulation. (2) Most coupled GCMs respond to increasing GHGs with a weakening of the Walker circulation, i.e. an El Niño-like response (Vecchi et al., 2006, 2008) inducing positive feedback (Bjerknes, 1969). This occurs through a differential response of global

mean precipitation and atmospheric humidity to a warming climate (Held and Soden, 2006). Tokinaga et al. (2012), however, attribute the weakening of the Walker circulation with climate warming mainly to the ocean (SST changes). Atmospheric models run with slab oceans exhibit an even stronger decrease in the Walker circulation (El Niño-like response), inducing positive feedback (Fig. 8). Fully coupled GCMs in principle contain both feedback mechanisms as described above. Vecchi et al. (2008) suggest that the atmospheric El Niño-like response dominates in these models, resulting in overall positive tropical circulation feedback. Sandeep et al. (2014) argued that SST changes during the 20th-century warming even led to an overall strengthening of the Pacific Walker circulation (while this strengthening was to some degree compensated for by variability induced by the El Niño–Southern Oscillation climate variability mode).

### 3.5 Sea ice feedbacks

A major climate feedback involving sea ice is the positive sea ice albedo feedback. This feedback is counteracted at least in part by several negative feedbacks (see feedback diagrams in Fig. 9). The study of sea ice feedbacks has not been systematic and the corresponding definitions and conceptual models vary among authors. A reason for this is that sea ice feedbacks are very often linked in a non-linear way and are there-

fore state dependent (Goosse et al., 2018). We consider here only those sea ice feedbacks which would have an impact on the Earth's surface temperature.

### 3.5.1 Sea ice albedo feedback

In analogy with the snow albedo feedback (Sect. 3.3.1), a melting of sea ice implies more open water, less snow, darker ice, and enhanced melt ponds, all reducing the large-scale albedo and increasing the absorption of shortwave radiation, which further melts the ice, leading to positive feedback (Curry et al., 1995; Holland et al., 2006; Winton, 2006). The ice albedo feedback is known to be one of the largest contributors to polar amplification (see glossary) (Pithan and Mauritsen, 2014), and its contribution to climate uncertainties is weak (Bitz, 2008). Long thought to be a potential source of sea ice tipping points (e.g. Lindsay and Zhang, 2005), the ice albedo feedback is in practice most likely counterbalanced by negative feedbacks.

### 3.5.2 Sea ice negative feedbacks

The fact that sea ice decrease is not self-accelerating (Notz and Marotzke, 2012) in the presence of the ice albedo feedback leads to the conclusion that negative sea ice feedbacks must exist. There are at least three potential mechanisms which lead to sea ice negative feedbacks. First, thinner ice is warmer and has a higher winter open-water fraction, which induces more LW emission. Second, thinner ice is less insulating. Third, thinner ice has less snow (Hezel et al., 2012), further decreasing the insulation power of the sea ice cover. Overall, these three mechanisms drastically (and non-linearly) increase the growth rate for thin ice (Bitz and Roe, 2004) contributing to rapidly bringing sea ice back to its equilibrium thickness in response to a perturbation (Tietsche et al., 2011). In the Southern Ocean, where the stratification of the water column is weaker than in the Arctic, two competing ice–ocean feedbacks have been documented (Goosse et al., 2018). The first feedback is negative and is termed ice production–entrainment feedback. It arises because brine rejection during freezing deepens the ocean mixed layer, bringing to the surface warmer water from deeper levels, melting a part of the ice initially formed and inhibiting ice production. The second feedback is positive and termed ice-production–ocean-heat-storage feedback. It stems from the fact that anomalous sea ice production induces vertical exchanges of salt, a higher stratification, storage of heat at depth, and finally lower oceanic heat fluxes that favour further ice production.

## 4 Introducing Earth system feedbacks

We will now look at the Earth system feedbacks, which are included in ESMs in addition to the classical physical feedbacks. Again, for each family of feedbacks as described in the following sections, we provide more details on the respective observational constraints in Appendix A.

### 4.1 Slow vegetation–land-surface–climate feedbacks

The feedbacks discussed in this section involve the transition of the vegetation cover from one form to another induced by climate change, which, through land–atmosphere interaction, leads to further alteration of the regional climate with consequences for the global climate and related feedbacks. The feedbacks operate at timescales of years to centuries. Three main factors control the character of the vegetation feedbacks: albedo, evapotranspiration, and shifts in forests to grassland or shrubs and vice versa. Important vegetation feedbacks include the positive vegetation snow-masking feedback, and the either negative or positive vegetation–evapotranspiration–albedo feedback (see feedback diagrams in Fig. 10).

### 4.1.1 Vegetation snow-masking feedback

This feedback mechanism is similar to the snow albedo and sea ice albedo feedbacks (Sect. 4.2) as it is caused by surface albedo changes. An increase in temperatures in high-latitude regions favours the growth of evergreen conifers compared to low tundra shrubs (Kaplan et al., 2003; Port et al., 2012). Coniferous forests typically mask the underlying snow. As a consequence, the surface albedo decreases with increasing forest cover, resulting in positive feedback. A northward expansion of boreal forests also induces an increase in evapotranspiration and in latent and sensible heat fluxes, due to surface roughness changes (Gustafsson et al., 2004). This negative feedback, however, is unlikely to compensate for the overall positive feedback due to the albedo change.

### 4.1.2 Vegetation–evapotranspiration–albedo feedback

The combination of changes in vegetation, hydrological cycle, and albedo at lower latitudes seems more uncertain than the high-latitude feedback described in the vegetation snow-masking feedback above. For desert areas such as the Sahel region, positive precipitation albedo feedback has been suggested. If rainfall increases with climate warming (as may have been the case during the Holocene climatic optimum 9000–6000 years before present; de Noblet-Ducoudre et al., 2000), then vegetation, soil moisture, evapotranspiration, and precipitation also increase, while albedo decreases, leading to an overall positive rainfall and temperature feedback (if albedo change dominates the thermal effect over evapotranspiration) (Brovkin, 2002). In other tropical areas such as the Amazon with progressive deforestation forcing, precipitation together with soil moisture and evapotranspiration could decrease, while albedo could increase.

Nevertheless, here the warming effect of decreasing evapotranspiration could dominate the cooling effect due to the

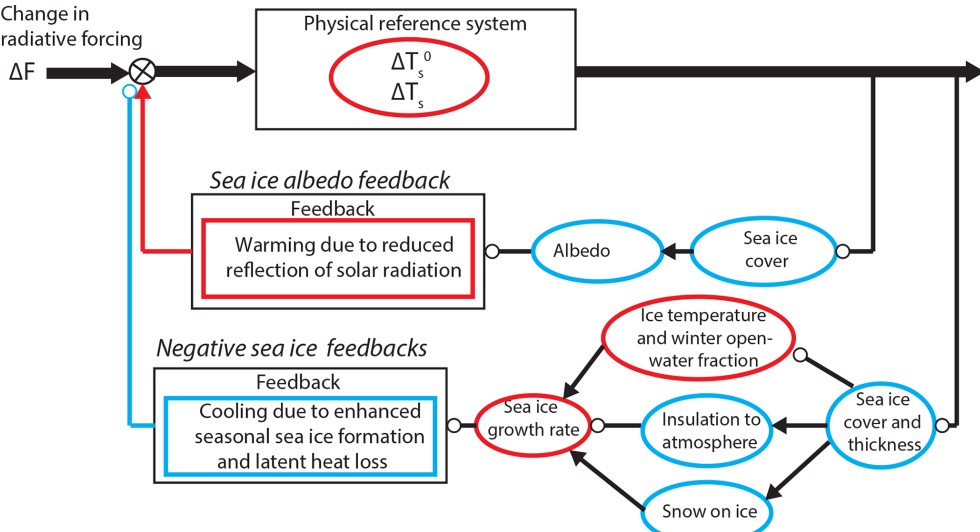

**Figure 9.** Schematic of the sea ice feedbacks. Arrows indicate positive coupling; open circles indicate negative coupling. Changes in state variables are indicated by ellipses, changes in processes by rounded rectangles. Red indicates increasing values of variables, strengthening of processes, or positive feedbacks; blue indicates the opposite. The change in surface air temperature $\Delta T_\mathrm{s}$ in the presence of feedback is different from the change $\Delta T_\mathrm{s}^0$ without feedback.

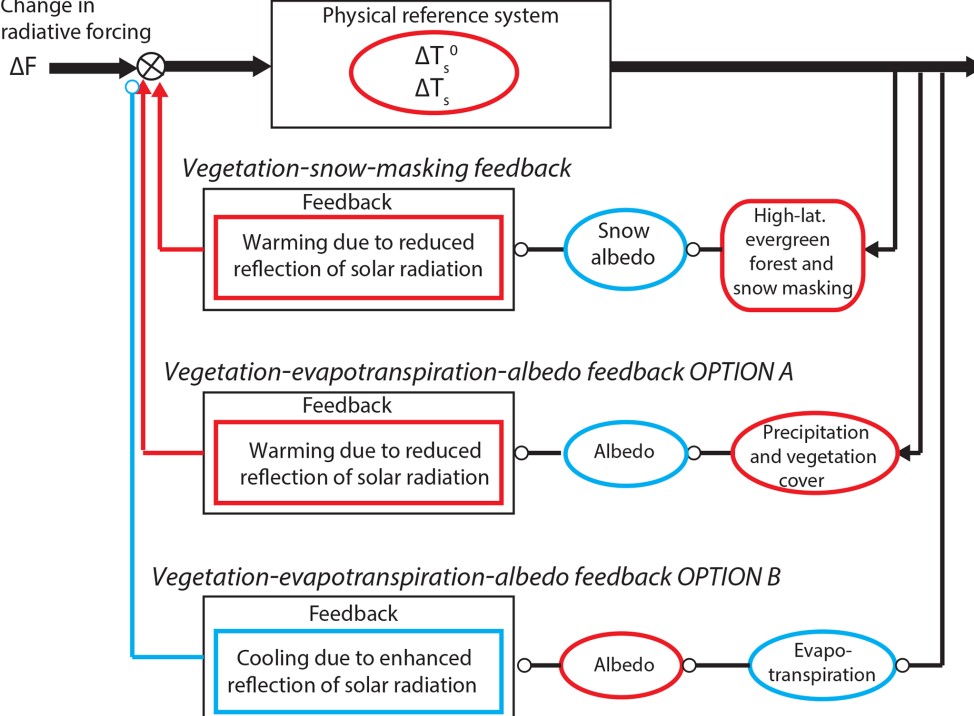

**Figure 10.** Schematic of important slow land surface feedbacks. The sign of the vegetation–evapotranspiration–albedo feedback can be positive if, for example, a desert starts to green due to increased precipitation. It can, however, also be negative in the case of progressing desertification (or due to extreme developments such as an Amazonian forest dieback). Arrows indicate positive coupling; open circles indicate negative coupling. Changes in state variables are indicated by ellipses, changes in processes by rounded rectangles. Red indicates increasing values of variables, strengthening of processes, or positive feedbacks; blue indicates the opposite. The change in surface air temperature $\Delta T_\mathrm{s}$ in the presence of feedback is different from the change $\Delta T_\mathrm{s}^0$ without feedback (see Sect. 2.2).

increase in albedo (Port et al., 2012). Even a dieback of the Amazonian rain forest under progressing human-induced global warming has been suggested by model simulations (Cox et al., 2004; Huntingford et al., 2008). Whether the coupled vegetation–evapotranspiration–albedo feedback results in negative or positive feedback to radiative forcing thus depends on the local conditions of respective precipitation changes with warming and on whether the thermal effects induced by evapotranspiration (changes in latent and sensible heat fluxes modulated by soil moisture availability and prevailing vegetation) changes or albedo changes dominate.

## 4.2 Slow ocean–atmosphere feedbacks

On decadal to centennial timescales, the most important ocean-related climate feedback is the ocean overturning circulation feedback (see feedback diagrams in Fig. 11). For the fast ocean feedbacks, see Sect. 3.4.

### 4.2.1 Ocean overturning feedbacks

Driven by density gradients and wind forcing, the global thermohaline circulation (THC) redistributes heat, salt, carbon, and other tracers. Due to the large heat capacity of water as compared to air, ocean heat uptake and transport of heat in the vast ocean interior via the large-scale overturning circulation are considered to be the main long-term response to a radiative forcing perturbation (Levitus et al., 2005). It can be considered to be a significant negative feedback (Fig. 11 option a).

Changes in this overturning circulation induced by climate change can provide positive or negative feedbacks. Seawater density at the sea surface decreases under warming. Higher temperatures, however, can drive a counteracting increase in density due to enhanced evaporation (due to a respective salinity increase). Likewise, regional cooling can increase surface density. Regionally increased freshwater fluxes decrease density, enhance the hydrostatic stability of the ocean water column, and hamper deep convection. In this context, the melting of ice sheets and glaciers plays an increasingly important role, especially in the Northern Hemisphere (Lique et al., 2015; Swingedouw et al., 2006; Yang et al., 2016). The Atlantic meridional overturning circulation (AMOC) is the most dynamic segment of THC and most vulnerable with respect to changes in forcing. When warm and salty tropical waters are advected to the high latitudes (losing heat on the way), they increase in density and destabilize the local density vertical stratification. This results in convective sinking, and climate-change-induced low-latitude evaporation will accelerate the deep overturning circulation. An opposite effect is observed when freshening (i.e. a decrease in salinity) at the high latitudes is in place, stabilizing the density vertical stratification and thus reducing the sinking rates and in turn decreasing the strength of the AMOC. ESMs quite consistently show a decrease in the meridional over-

turning circulation due to warming and freshening (i.e. a stronger density stratification) at the deep-water production areas during GHG-induced warming (Collins et al., 2013; Meehl et al., 2007b). It is therefore likely that the slowing down of the overturning circulation will result in positive feedback (Fig. 11 option b), i.e. a diminishing of the otherwise effective heat transport in the ocean's interior.

Such sensitivity-altering feedbacks may not be the only ones at work (though this paper focuses on these feedbacks). Stability-altering feedbacks are also possible (see discussion in Bates, 2007). Multiple possible equilibriums of THC suggest potential transitions and shifts between the different states when some threshold is reached, likely causing abrupt climate change (Alley et al., 2003; Marotzke, 2000; Stommel, 1961; Weaver et al., 1993). Conceivable triggers are the elements of the freshwater cycle in the North Atlantic. Palaeo-records from Greenland ice cores and North Atlantic sediments (Clark et al., 2002; McManus et al., 2004) have shown that there were abrupt climate changes in the past when the THC switched between different regimes, thereby changing the climate in North America and Europe. However, it is deemed unlikely that the AMOC will collapse or shift regimes and result in abrupt climate change by the end of the 21st century (Collins et al., 2013; Meehl et al., 2007b). However, all projections agree that in the warming climate, increased heating and freshening at the ocean surface in the high latitudes will enhance the stability of the water column at the deep-water formation sites, resulting in positive feedback due to weaker mixing of excess heat downward. An apparent monostable mode of the AMOC in a state-of-the-art ESM may, however, be due to the tropical ocean salinity bias in these models (Liu et al., 2014) so that no final conclusion can be drawn on the possibility of stability-altering feedback, with potentially complex implications for sensitivity-altering feedbacks as well.

So far, we have mainly looked at the physical processes in the climate system. We will now introduce the chemical and biogeochemical components of the climate system.

## 4.3 Land biogeochemistry feedbacks

Currently, about half of the $CO_2$ emitted annually through human activities is taken up in about equal portions by land and ocean processes (Le Quéré et al., 2014, 2016). Carbon uptake by land and ocean has a fast response component. In addition, a series of slow processes contribute to significant long-term responses on decadal to multi-millennial timescales. Land uptake of $CO_2$ is a biogeochemical process that involves an increase in organic matter stored in vegetation and soils. $CO_2$ fertilization – enhanced plant growth due to elevated atmospheric $CO_2$ concentrations – is thought to be one of the underlying mechanisms. In addition, other mechanisms such as changes in nutrient cycles, for example through increased reactive nitrogen deposition, or changes in land use management (and climate-induced vegetation

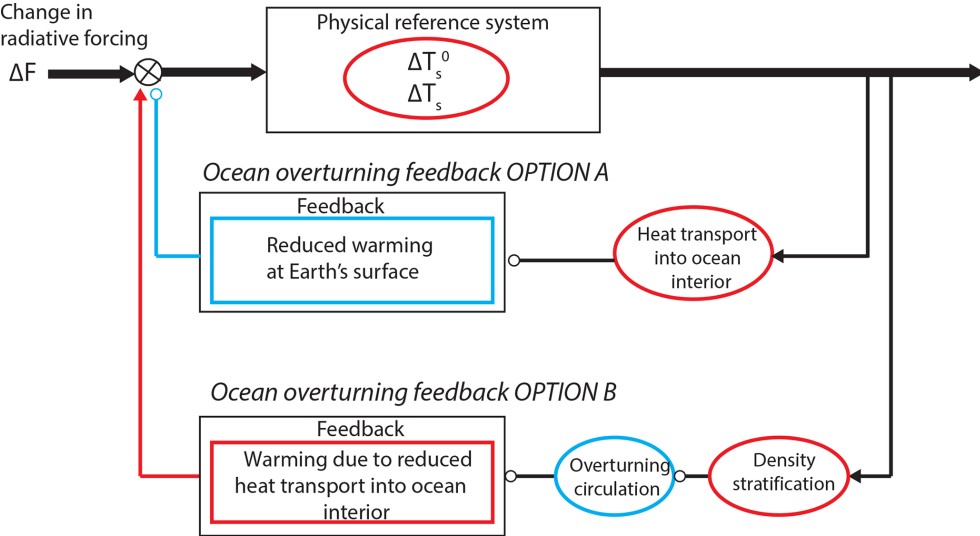

**Figure 11.** Schematic of the slow ocean–atmosphere feedback. Arrows indicate positive coupling; open circles indicate negative coupling. Changes in state variables are indicated by ellipses, changes in processes by rounded rectangles. Red indicates increasing values of variables, strengthening of processes, or positive feedbacks; blue indicates the opposite. The change in surface air temperature $\Delta T_s$ in the presence of feedback is different to the change $\Delta T_s^0$ without feedback.

changes; see Sect. 4.1) contribute to variations in the land carbon uptake. Whether the total land carbon uptake is considered to be feedback (Friedlingstein et al., 2003, 2006) or a system response (with only deviations from a reference uptake case counted as a feedback) (Previdi et al., 2013) is a conceptual question. As such, a reference case is very arbitrary, so we suggest including the entire land carbon uptake as a strong feedback (see the Sect. 2.2). Currently this feedback is negative, but it can change to a strong positive feedback in the course of climate change and changing $CO_2$ emissions (Friedlingstein et al., 2006). Important land biogeochemistry–climate feedbacks (see feedback diagrams in Fig. 12) are associated with changes in net ecosystem productivity (NEP; that is, gross primary productivity and plant and soil respiration) and with permafrost thawing, $CO_2$ fertilization, and fires. Arneth et al. (2010) provide a comprehensive overview of a suite of terrestrial carbon cycle climate feedbacks.

### 4.3.1 Feedback between net ecosystem productivity, climate change, and rising $CO_2$

The NEP of the land biosphere represents the difference between the large ($> 100\,\mathrm{PgC\,yr^{-1}}$) gross flux due to gross primary production (GPP, equivalent to photosynthesis) and the oxidative processes of autotrophic respiration ($R_A$, by green plants), heterotrophic respiration ($R_H$, by soil bacteria, fungi, and animals), and the combustion of biomass in wildfires. Climate change can affect all of these gross fluxes. Numerous studies show that warming will induce an increase in both autotrophic and heterotrophic respiration, leading to a

larger $CO_2$ loss by the ecosystems and thus positive feedback. Recent studies confirm that roots and fungi in soils can indeed lead to outgassing and less effective storage of carbon on land (Cheng et al., 2014; Phillips et al., 2012). However, the study by Clemmensen et al. (2013) suggests the opposite effect in boreal forests, namely the stimulation of carbon sequestration by roots and fungi in these regions. Further, in a warmer world, permafrost areas may shrink and more wetlands may appear due to this process or develop in already humid regions if these receive additional precipitation. The latter is projected for many regions (the contrast between dry and wet regions is likely to increase under global warming; Collins et al., 2013). Increasing wetland formation has the potential to lead to substantial additional $CH_4$ release to the atmosphere (Gedney et al., 2004). The inclusion of wetland and permafrost dynamics in ESMs is still in its infancy (Koven et al., 2011; Lawrence and Slater, 2008; Schuur et al., 2015). It is expected that the thawing of permafrost areas leads to a substantial outgassing of $CO_2$ and $CH_4$ (methane) depending on the biogeochemical conditions (such as oxygen availability) (Lee et al., 2012). The overall quantitative partitioning of permafrost carbon release into $CO_2$ and $CH_4$ is uncertain (Ciais et al., 2013). See the discussion in Sect. 4.6 for implications of increased $CH_4$ emissions for the tropospheric gas-phase chemistry.

Regarding production, the picture is quite complex, since warming would, on the one hand, eventually lead to the limitation of productivity as plants would suffer from heat or water stress, but, on the other hand, increased soil decomposition also leads to greater mineralization of nitrogen which would reduce plant nitrogen limitation and hence would en-

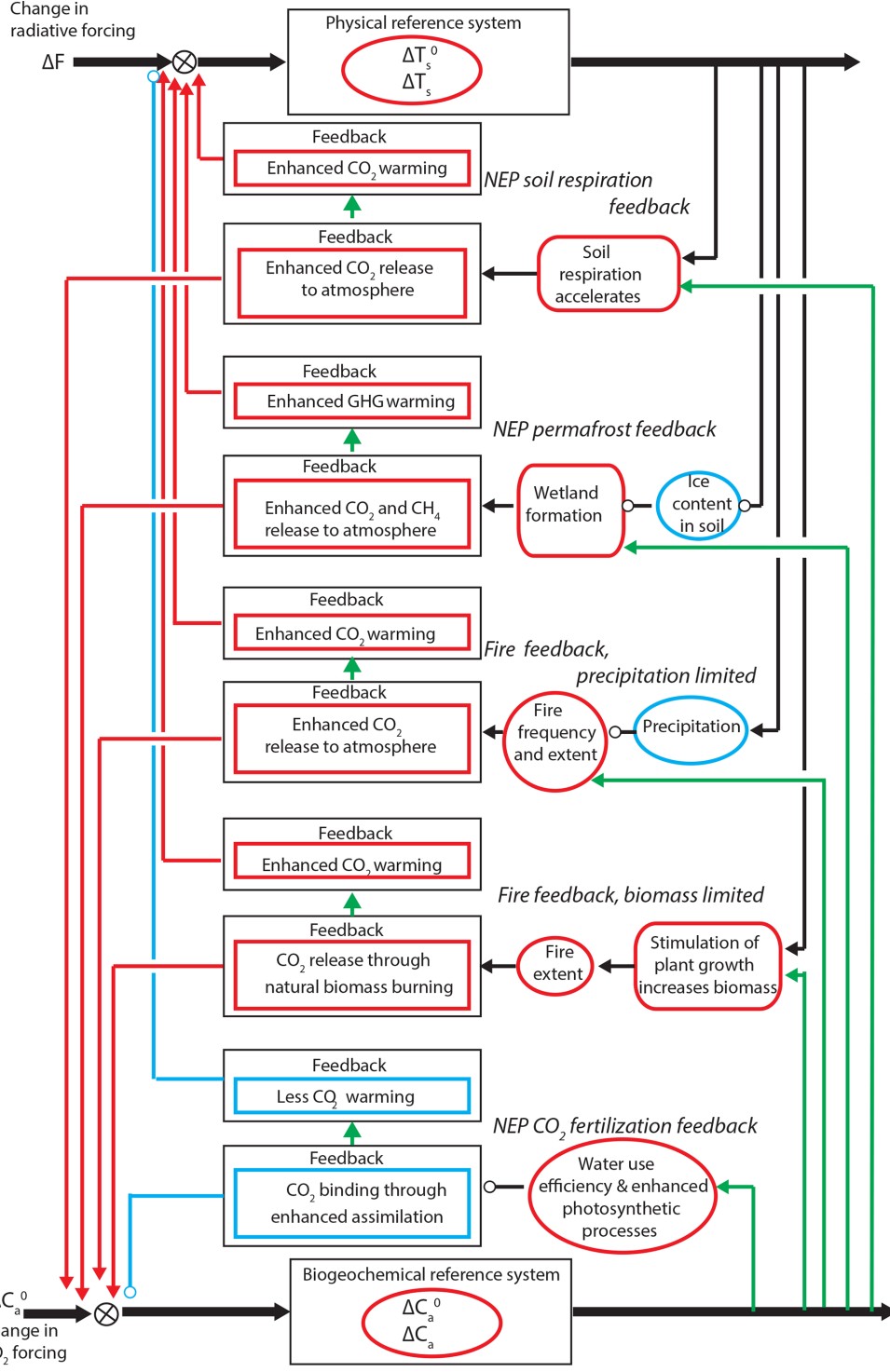

**Figure 12.** Schematic of important land biogeochemistry feedbacks. The fire feedback depends on the local change in precipitation or available biomass for burning. Arrows indicate positive coupling; open circles indicate negative coupling. Changes in state variables are indicated by ellipses, changes in processes by rounded rectangles. Red indicates increasing values of variables, strengthening of processes, or positive feedbacks; blue indicates the opposite. Green arrows and lines indicate couplings of biogeochemical processes to physical processes. The change in surface air temperature $\Delta T_{\mathrm{s}}$ in the presence of feedback is different from the change $\Delta T_{\mathrm{s}}^0$ without feedback. The change in greenhouse gas concentration $\Delta C_{\mathrm{a}}$ in the presence of biogeochemical feedback is different from that without this feedback ($\Delta C_{\mathrm{a}}^0$).

hance photosynthesis (Thornton et al., 2009). Higher $CO_2$ levels in the atmosphere can have a fertilizing effect on plant growth (Field, 2001) due to improved water use efficiency (Farquhar et al., 1980; Woodward, 1987) and enhanced photosynthetic processes (Liberloo et al., 2009; Norby et al., 2005). Systematic field experiments show a partial confirmation of this behaviour though the response may depend on hydrological conditions and seems to be species dependent (Nowak et al., 2004). In contrast to the study by Thornton et al. (2009), the $CO_2$ fertilization effect can be reduced when nitrogen limitation of plant growth is included in terrestrial biosphere models (Zaehle et al., 2010). In fact, the two models included in IPCC AR5, which have coupled carbon and nitrogen representations, estimated less anthropogenic carbon uptake than other models (Ciais et al., 2013). Details of the approach for C–N coupling as applied in these models may need improvements (Bonan and Levis, 2010). For the role of changes in nitrogen-related aerosols and gas-phase nitrate and ammonium deposition within this context, see Sect. 4.5. For the potential change in terrestrial micronutrient supply from changes in atmospheric dust deposition, see Sect. 4.5. For changes in emissions of biogenic volatile organic compounds (BVOCs), see Sects. 4.5 and 4.6.

In addition, nitrous oxide ($N_2O$) is an important radiatively active gas, which is influenced by the land biosphere. $N_2O$ emissions are caused by fossil fuel burning, biomass burning, and artificial fertilizer (its industrial production and use in agriculture). In addition, natural emissions from soils (and oceans) contribute to the budget. Increases in precipitation and temperature influence the release of $N_2O$ from soils (Li et al., 2013). Rising temperatures can induce a higher or lower rate of soil $N_2O$ emissions (Avrahami et al., 2003; Barnard et al., 2005) as such emissions due to bacterial activity peak at certain soil temperatures. Climate-induced drying of soil layers may lead to enhanced outgassing of $N_2O$ from nutrient-rich soils (Avrahami et al., 2003; Barnard et al., 2005; Martikainen et al., 1993).

### 4.3.2 Feedback between climate change and $CO_2$ emissions from fires

Climate change might alter the frequency and intensity of wildfires (non-deforestation fires in contrast to purposefully created fires) (Pechony and Shindell, 2010). The global response is not clear as regional responses will differ depending on the base state and the future change in precipitation as well as moisture (of fuel and soil) (Arora et al., 2013; Kloster et al., 2010; Randerson et al., 2006; Thonicke et al., 2001). Fires can be limited either by an excess of precipitation or by a lack of biomass as fuel to spread a fire (Spessa et al., 2005). In regions where fire is limited by high amounts of precipitation, future reductions in precipitation would increase fire risk and hence lead to positive feedback, provided that increased fire frequency leads to a reduction in standing biomass in addition to the effect of less precipitation alone

(and thus a net release of $CO_2$ to the atmosphere) and that higher atmospheric $CO_2$ levels lead to further reductions in precipitation (Fig. 12). In regions where fire is biomass limited, climate change, as well as increased atmospheric $CO_2$, might increase the fire risk if plant productivity is stimulated, thus leading to larger biomass (Knorr et al., 2016). **CE11** The major feedback associated with fires is that the release of additional $CO_2$ to the atmosphere through the effect of non-deforestation fires due to climate change may be small relative to purposeful biomass burnings because most biomass burned in wildfires is dead plant material, so that fire simply provides an accelerated decomposition route compared to microbial decomposition (Bowman et al., 2009; Landry et al., 2015). Furthermore, fire occurrence, spread, and intensity depends a lot on human behaviour so that changes in population density, fire suppression or firefighting strategies, and technology (for example, the use of electrical stoves instead of wood-burning stoves) may have large impacts on future biomass burning. The overall feedback between fires and climate is complex because it is not only changes in the $CO_2$ budget that are involved but also variations in heat and moisture fluxes, and changes in aerosol forcings with related alterations of cloud properties (Jacobson, 2014; Landry et al., 2015). Wildland fire activity also leads to emissions of $CH_4$, CO, VOCs (volatile organic compounds), and $NO_x$ with implications for tropospheric ozone concentrations (see Sect. 4.6).

### 4.4 Marine biogeochemical feedbacks

Ocean uptake of anthropogenic $CO_2$ from the atmosphere is mainly driven by the concentration gradient across the ocean surface and the inorganic buffering ability of seawater for $CO_2$. Again, whether one includes this process altogether as a feedback (Friedlingstein et al., 2003, 2006) or as a system response accounting for feedbacks for deviations relative to a reference standard uptake case (Previdi et al., 2013) is a conceptual question (see Sects. 2.2 and 4.3). In analogy with the ocean overturning feedback, we tend to regard the total uptake of anthropogenic $CO_2$ by the oceans as a strong negative feedback. $CO_2$ at the sea surface is kept at a comparatively low concentration by the biological production of organic matter and related vertical particle flux. Major marine biogeochemical feedbacks to climate change and high $CO_2$ (see feedback diagrams in Fig. 13) include positive feedback via the reduction in $CO_2$ solubility in seawater at increasing temperatures and positive or negative feedback through a change in the seawater buffer capacity (increasing with rising temperature, decreasing with rising $CO_2$). A reduction in ocean overturning will lead to positive biogeochemical feedback through increased accumulation of anthropogenic carbon in the upper ocean.

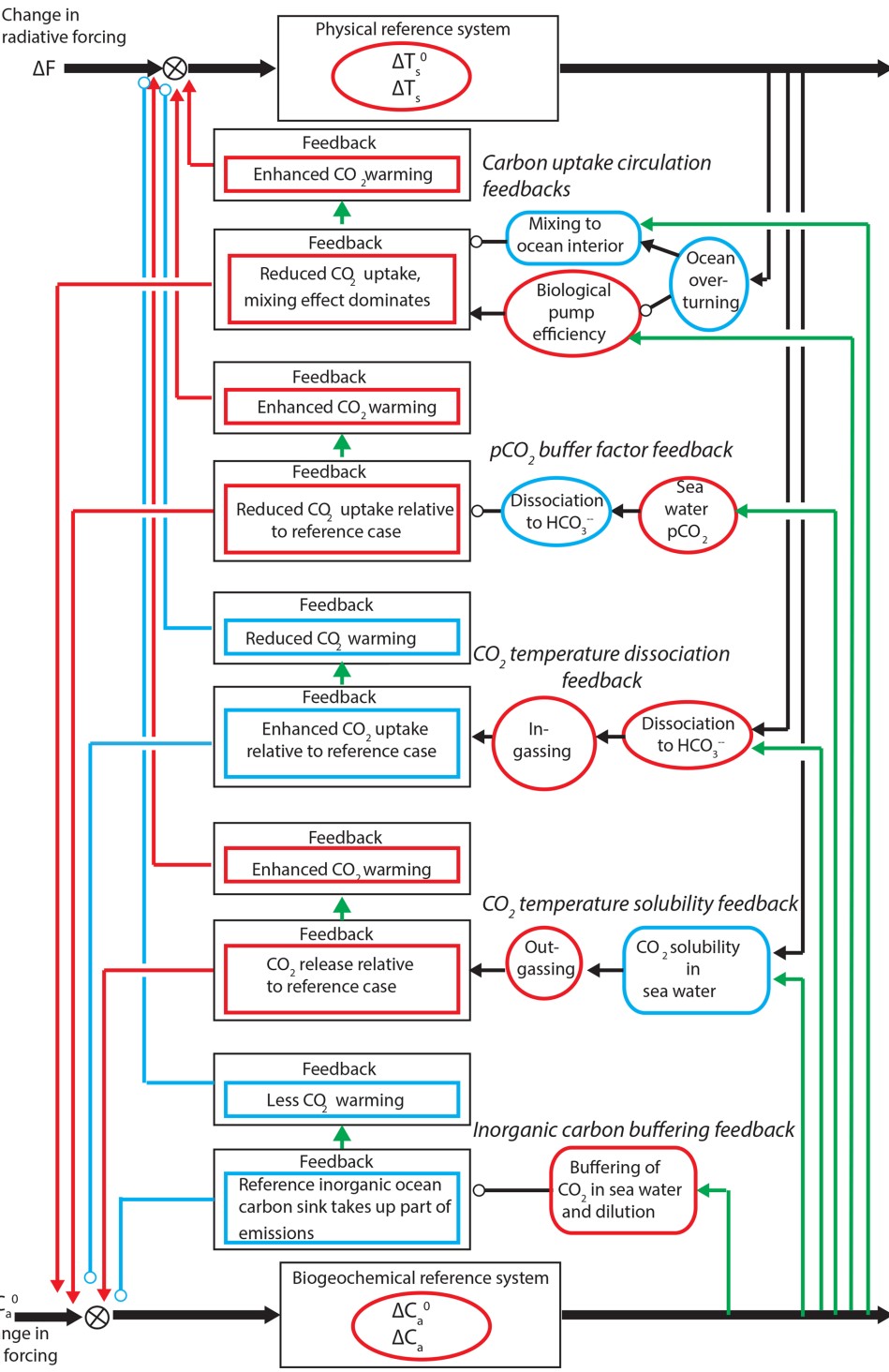

**Figure 13.** Schematic of important ocean biogeochemistry feedbacks. Arrows indicate positive coupling; open circles indicate negative coupling. Changes in state variables are indicated by ellipses, changes in processes by rounded rectangles. Red indicates increasing values of variables, strengthening of processes, or positive feedbacks; blue indicates the opposite. Green arrows and lines indicate couplings of biogeochemical processes to physical processes. The change in surface air temperature $\Delta T_s$ in the presence of feedback is different from the change $\Delta T_s^0$ without feedback. The change in greenhouse gas concentration $\Delta C_a$ in the presence of a biogeochemical feedback is different from that without this feedback ($\Delta C_a^0$).

### 4.4.1   Inorganic ocean carbon cycle feedbacks due to changes in carbon chemistry

The ocean takes up $CO_2$ through air–sea gas exchange. In contrast to fresh water, seawater dissociates most of the $CO_2$ into bicarbonate $HCO_3^-$, while the seawater carbonate $CO_3^{2-}$ is reduced and the proton concentration rises (Dickson, 2007; Zeebe and Wolf-Gladrow, 2001). The inorganic carbon equilibration is the major negative biogeochemical feedback in the Earth system (Fig. 13). Its strength depends partially on how fast the excess carbon is transported to the ocean interior, and hence this feedback is related to the physical ocean overturning feedback. $CO_2$ solubility in seawater depends strongly on temperature. The reduction in $CO_2$ solubility due to ocean warming provides positive feedback, which will reduce the oceanic uptake of anthropogenic carbon by 9 %–15 % by the end of the 21st century as compared to an ocean with pre-industrial temperatures (Joos et al., 1999; Matear and Hirst, 1999; Plattner et al., 2001; Sarmiento and Le Quéré, 1996) (Fig. 13). The effect of increased freshwater supply to the surface ocean on $CO_2$ solubility is small and the resulting feedback to anthropogenic $CO_2$ uptake is less than 1 % with an uncertain sign (Plattner et al., 2001). The ocean has a large $CO_2$ uptake capacity because $CO_2$ is not simply dissolved in water like other gases, but it reacts with water and dissociates into bicarbonate $HCO_3^-$ and carbonate $CO_3^{2-}$ ions (Zeebe and Wolf-Gladrow, 2001). Negative feedback arises with increasing temperature as the dissociation of $CO_2$ into bicarbonate and carbonate is enhanced in warmer waters (Fig. 13). On the other hand, rising atmospheric $CO_2$ and respective oceanic $CO_2$ uptake lower the buffering ability of seawater as decreasing amounts of $CO_3^{2-}$ ions become available for neutralizing the additional $CO_2$ (for details about the buffer factor or Revelle factor, see Zeebe and Wolf-Gladrow, 2001). This induces positive feedback (Fig. 13). Orr et al. (2005) suggested that the change in surface ocean dissolved inorganic carbon (the sum of the $CO_2$, $HCO_3^-$, and $CO_3^{2-}$ concentrations) per unit change in atmospheric $CO_2$ would be approximately 60 % lower in the year 2100 than it is today. These positive feedbacks due to the $CO_2$ solubility and the $CO_2$-dependent buffer factor dominate the antagonistic temperature dependency of the $CO_2$ dissociation, leading globally to an overall positive feedback to atmospheric $CO_2$. In a more sluggish ocean, sinking particles that have been produced at the surface tend to sink deeper, as gravity acceleration is not changed. The surface-to-deep gradient of the carbon and nutrient concentrations steepens; i.e. the biological carbon pump (see glossary) is more efficient (Boyle, 1988; Heinze et al., 1991). This negative feedback (Fig. 13) is expected to be considerably smaller than the feedback due to reduced physical downward transport of surface waters with high anthropogenic carbon loadings (Broecker, 1991; Maier-Reimer et al., 1996; Plattner et al., 2001). The stronger partial retention of waters with high anthropogenic $CO_2$ burdens at the sea surface will thus dominate over the more efficient biogenic downward particle flux in a more slowly overturning ocean.

### 4.4.2   Feedbacks due to changes in marine ecosystems

Conversion of inorganically dissolved carbon to organic matter through photosynthesis and phytoplankton growth increases with rising temperature (Eppley, 1972). On the other hand, the degradation of organic matter involving bacterial activity also accelerates with warming (Bendtsen et al., 2002; Rivkin and Legendre, 2001). These two effects result in partially compensating feedbacks. The present progressive increase in atmospheric $CO_2$ concentrations may slow down the biotic production of shell material built of calcium carbonate ($CaCO_3$) in the surface ocean via lowering pH values and the $CaCO_3$ saturation state (Raven et al., 2005; Riebesell et al., 2000; Zondervan et al., 2001). A net decrease in $CaCO_3$ production, however, would provide only a small feedback to atmospheric $CO_2$ as compared with the projected accelerating fossil fuel $CO_2$ emissions (Gehlen et al., 2007; Heinze, 2004; Hofmann and Schellnhuber, 2009; Ridgwell et al., 2007). This slightly negative feedback could be counteracted by positive feedback due to the lack of $CaCO_3$ particles as an efficient ballast of sinking marine particles (Heinze, 2004; Klaas and Archer, 2002). Over very long timescales (10 000–100 000 years), marine $CaCO_3$ sediment exposed to waters with high anthropogenic carbon loads will dissolve and release carbonate ions into the water column and thus provide an important long-term negative feedback to atmospheric $CO_2$ concentrations (Archer, 2005; Broecker and Takahashi, 1977). Carbon overconsumption (a change in the average stoichiometry of $C : N : P$ for producing new biomass towards the higher binding of carbon per nutrient unit available) by marine primary producers has been observed in mesocosm experiments as potentially significant negative feedback to rising $pCO_2$ (Riebesell et al., 2007). It is not yet clear whether this result can be extrapolated to a larger scale, as it would contradict the marine $^{13}C$ sediment core record that has been used for studying potential processes leading to the low glacial $pCO_2$ levels (e.g. Heinze and Hasselmann, 1993; Shackleton and Pisias, 1985; Zahn et al., 1986). For potential changes in ocean micronutrient supply from changes in atmospheric dust deposition and related stimulations of plankton growth, see Sect. 4.5. For changes in emissions of BVOCs, see Sects. 4.5 and 4.6. Feedbacks through marine $N_2O$ production can arise through changes in nitrification rates at the ocean surface and denitrification processes at the transition from oxygenated to oxygen-poor areas in both the water column and the sediment (Voss et al., 2013; Zehr and Ward, 2002). Marine $N_2O$ sources seem to be dominated by nitrification (Freing et al., 2012), but outgassing of $N_2O$, for example, in CE12 coastal upwelling regimes, is difficult to observe due to its confined spatial extent (Kock et al., 2016; Rhee et al., 2009). Climate-induced changes in upwelling rates, mixing, and oxygenation (deoxy-

genation due to lower $O_2$ solubility at higher temperatures) can induce $N_2O$ feedbacks. These are not yet quantified at the global scale.

## 4.5 Aerosol–climate feedbacks

Feedbacks between climate and atmospheric composition are expected due to modifications in the burden and distribution of atmospheric aerosols, which in turn affect clouds and radiation and thus climate. By adding nutrients and pollutants, aerosols also influence the carbon cycle and other biogeochemical cycles (Mahowald et al., 2011). Biological processes in the atmosphere itself usually play no major quantitative role in climate feedback processes in contrast to the elemental cycles on land and in the ocean. Aerosols, in contrast to long-lived greenhouse gases (such as $CO_2$, $CH_4$, $N_2O$, and CFCs), exert a radiative forcing in both the shortwave and the longwave spectrum due to the wide spread of their particle size distribution (Tegen and Lacis, 1996). Whereas the radiative forcing of long-lived greenhouse gases is relatively uniform spatially, the aerosol forcing is made of sharp gradients that are highly correlated with the atmospheric column loads. Aerosols can be classified as different types such as urban–industrial aerosol, aerosol from biomass burning, mineral dust, sulfate aerosol produced from volcanic eruptions, organic aerosol from terrestrial plants, and marine or maritime aerosol (Gregory, 2010; Russell et al., 2014). Several components of the Earth system (e.g. ocean, bare soils, terrestrial and marine ecosystems, and tectonics or volcanism) make important natural contributions to global aerosol loads. Important aerosol climate feedbacks (see feedback diagrams in Fig. 14) are the positive DMS–sulfate–cloud albedo feedback in coupling to ocean acidification, the potentially negative dust mobilization and fertilization feedback, and the negative secondary aerosol feedback.

### 4.5.1 Feedbacks between marine aerosol emissions and climate change

DMS $(CH_3)_2S$ is a by-product of marine biological productivity. Marine DMS is one of the largest natural sulfur sources to the atmosphere (Bates et al., 1992). Marine biology could thus be involved in a climate feedback through changes in DMS emissions from the ocean, the subsequent formation of sulfate aerosol – and hence cloud condensation nuclei (CCN) – and the alteration of cloud properties as well as radiative fluxes (Carslaw et al., 2010; Charlson et al., 1987). It is now realized that the feedback could operate in numerous ways through changes in temperature, solar radiation dose, mixed-layer depth and nutrient recycling, sea ice extent, wind speed, shift in marine ecosystems due to ocean acidification and climate change, or atmospheric processing of DMS into CCN (Ayers and Cainey, 2007; Bopp et al., 2004; Kloster et al., 2007). Changes in surface UV radiation may also feed back to DMS production (Vallina

and Simo, 2007). However, no study to date has included all the relevant effects, and the processes involved are complex and not all understood. The sign and magnitude of the DMS–sulfate–cloud albedo feedback thus remains uncertain. On the basis of controlled marine field experiments (mesocosm experiments), recently a decrease in marine DMS production with progressing ocean acidification (declining pH value of seawater) has been suggested. This could lead to a sizable positive feedback to chemical and climatic anthropogenic climate forcing of 0.23–0.48 K for business-as-usual scenario conditions by 2100 (Schwinger et al., 2017; Six et al., 2013) (Fig. 14).

### 4.5.2 Feedbacks between dust mobilization and climate, including fertilizing effects

Elevated $CO_2$ concentrations and climate change will increase the extent of dry land during this century (Huang et al., 2016; Rajaud and de Noblet-Ducoudre, 2017) and may induce further changes in soil moisture and wind speed. This will increase natural dust emissions. The current understanding of the radiative properties of atmospheric dust (Balkanski et al., 2007; Di Biagio, 2017) suggests that negative radiative forcing goes along with dust increases. However, regional effects can also be in the direction of positive feedback depending on the reflectivity of the Earth's surface conditions and the radiative properties of dust particles (Miller et al., 2014). Miller et al. (2014) also point out the potential impact of mineral dust on precipitation changes. Mahowald et al. (2010) suggested that the largest dust radiative forcing in the 20th century in a particular dusty decade is $-0.57\,\mathrm{Wm}^{-2}$ less than in a less dusty decade. A large range in future dust conditions has been suggested, predicting a range from a 60 % less dusty future (Mahowald and Luo, 2003) to a moderate increase of 10 %–20 % in dust (Tegen et al., 2004). Atmospheric deposition of aerosols provides nutrients over land and oceans. The addition of soluble iron, nitrogen, and phosphorus to marine and terrestrial ecosystems in regions that are limited in these nutrients can lead to related feedbacks. Climate-related changes in dust uplift and transport can, for example, supply nutrients to the Amazon Basin (Kaufman et al., 2005; Yu et al., 2015). The deposition of micronutrients (such as bio-utilizable iron) as well as macronutrients (such as phosphate) may regionally enhance terrestrial and marine biological production (Aumont et al., 2008; Jickells et al., 2005; Ridame et al., 2014; Ridgwell, 2002; Swap et al., 1992; Tagliabue et al., 2014). The related negative feedback (Fig. 14) may be enhanced by the additional ballast to biogenic marine particle aggregates (Bressac et al., 2014). In oceanic regions where iron is limiting or co-limiting, additional inputs through atmospheric deposition from mineral dust or combustion and transfer from continental shelves stimulate phytoplankton growth (Martin et al., 1991; Moore et al., 2013). Gabric et al. (2010) suggest that nitrogen utilization is affected in regions where soluble iron inputs are high. The

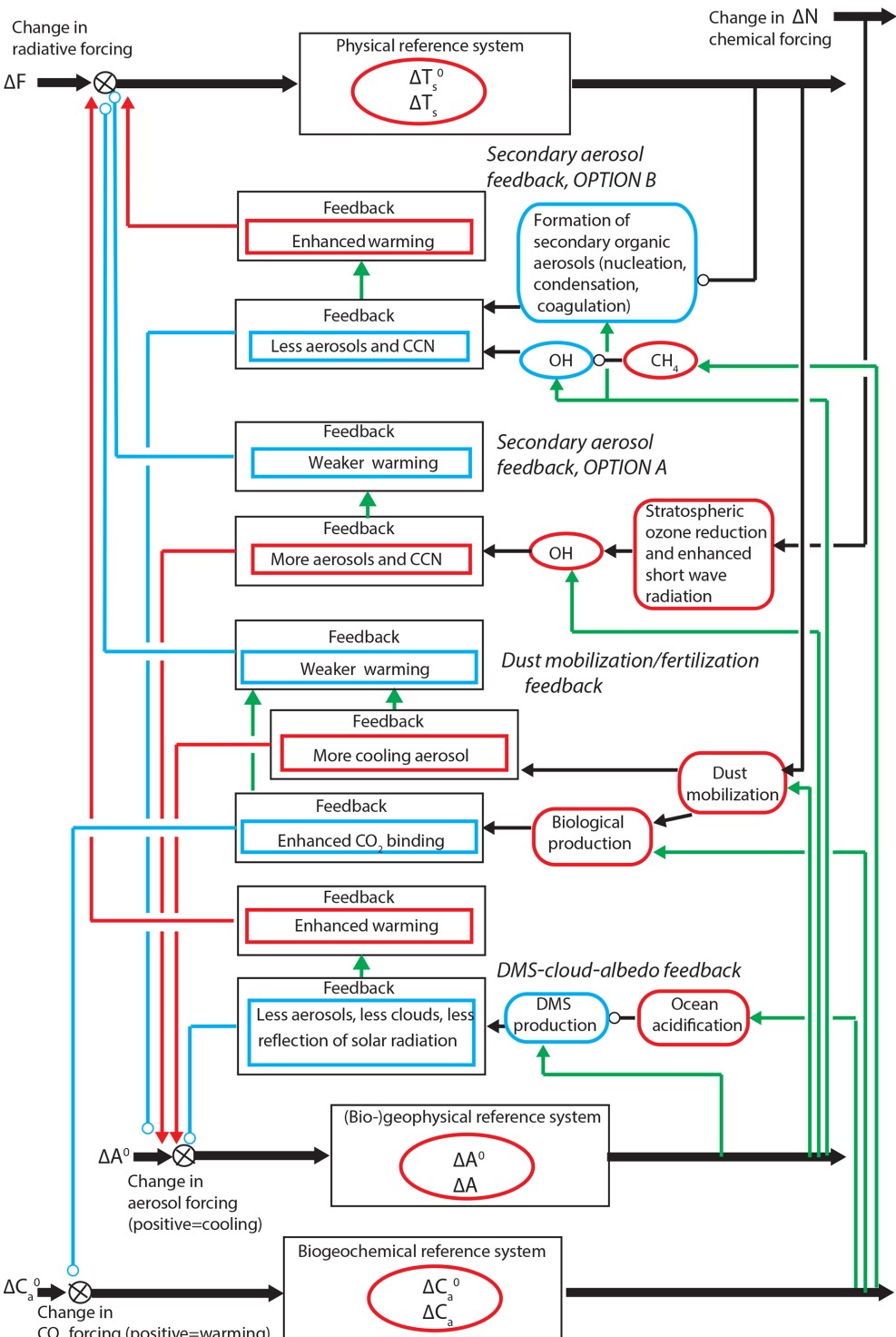

**Figure 14.** Schematic of important aerosol feedbacks. Arrows indicate positive coupling; open circles indicate negative coupling. Changes in state variables are indicated by ellipses, changes in processes by rounded rectangles. Red indicates increasing values of variables, strengthening of processes, or positive feedbacks; blue indicates the opposite. Green arrows and lines indicate couplings of biogeochemical processes and aerosol processes to physical processes. The change in surface air temperature $\Delta T_s$ in the presence of feedback is different from the change $\Delta T_s^0$ without feedback. The change in greenhouse gas concentration $\Delta C_a$ in the presence of a biogeochemical feedback is different from that without this feedback ($\Delta C_a^0$). Likewise, the change in aerosols $\Delta A$ in the presence of aerosol feedback processes is different from that without this feedback ($\Delta A^0$).

stimulation of phytoplankton growth draws $CO_2$ from the atmosphere to the ocean (see Sect. 4.4). As much as 45 ppm of $CO_2$ could be drawn out of the atmosphere through this process by 2100 (Mahowald et al., 2011).

Future projections of aerosol components for the 21st century indicate a decrease in the aerosol components in the nitrate–ammonium–sulfate system except for ammonia (Bellouin et al., 2011; Hauglustaine et al., 2014). Emissions of ammonia, one of the two main precursors of ammonium nitrate alongside $NO_x$, have risen over the 20th century and will continue to rise due to agricultural practices. Nitrogen deposition stimulates plant growth (Zaehle et al., 2010) and seems to inhibit soil respiration (Janssens et al., 2010) (see Sect. 4.3). The present and future increase in nitrogen deposition changes the extent to which regions are nitrogen limited. Note that nitrogen is deposited from both aerosol and gas-phase nitrate and ammonium (see Sect. 4.6).

### 4.5.3 Secondary aerosol feedbacks

Secondary aerosols (nitrate, sulfate, and organic matter) are produced within the atmosphere from precursor gases and influence the radiative budget of the climate system. Secondary aerosol formation depends on the oxidizing capacity (see glossary) of the atmosphere (Murray et al., 2014), i.e. the concentrations of ozone $O_3$, the hydroxyl radical OH, and hydrogen peroxide $H_2O_2$ (Thompson, 1992). The hydroxyl radical OH and its distribution play an important role in this respect and are subject to multiple possible feedback loops (see Sect. 4.6). An increase in OH would increase aerosol loads and CCN concentrations and thus the negative aerosol radiative forcing, thus decreasing surface temperature and thus exhibiting negative feedback (Fig. 14). However, this process could be counteracted by a further process. Semi-volatile species such as ammonium nitrate and secondary organic aerosols (SOAs) are likely to partition more towards the gas phase in a warmer climate (resulting in a reduced negative forcing from less aerosol). The corresponding gaseous species such as $HNO_3$ are also believed to have a shorter lifetime than, for example, particulate nitrate (Xu and Penner, 2012). Nitrate particles and organic semi-volatile components are reduced as they shift from the particle phase to the gas phase (Sheehan and Bowman, 2001; Tsigaridis and Kanakidou, 2007). Together this would decrease the negative aerosol radiative forcing and hence exhibit positive climate feedback. Details concerning the role of VOCs and BVOCs for SOA formation can be found in Sect. 4.6. Due to warming, an increase in BVOC emissions is expected with the potential for enhanced SOA formation and thus negative climate feedback.

### 4.5.4 Aerosol–cloud feedbacks

Aerosols have the potential to significantly modify cloud distribution through several effects (Boucher et al., 2013). Some aerosol species, such as black and brown carbon, are efficient absorbers of solar radiation in the atmosphere and may therefore lead to substantial radiative heating near cloud layers inducing a semi-direct effect (Hansen et al., 1997; Ramanathan et al., 2001). Aerosols have been observed to increase and decrease cloud cover depending on whether condensation or evaporation effects dominate (Koren et al., 2008). Li et al. (2011) suggest that high aerosol loads invigorate upward motion in clouds, thus altering regional circulation patterns. Even though substantial progress has been made in investigating aerosol–cloud interactions, the increasingly complex picture makes providing an overall conclusive statement on the related climate feedback difficult (Boucher et al., 2013).

### 4.6 Tropospheric gas-phase chemistry feedbacks

Ozone ($O_3$) and methane ($CH_4$) are both important greenhouse gases and reactive enough for their concentrations to be controlled by tropospheric chemistry processes. These species also play a major role in controlling the concentration of the OH radical, which is the most important oxidant in the troposphere. OH is generated and lost via many different chemical reactions and is therefore involved in many different feedback loops (Isaksen et al., 2009). The most important parameters driving tropospheric OH sensitivity to climate change are water vapour, methane, and stratospheric ozone (Lelieveld et al., 2004). Stratospheric ozone reductions will increase tropospheric UV radiation and hence increase OH. Significant future methane increases would counteract and reduce OH levels.

Changes in atmospheric $CH_4$ concentrations are induced by emission changes or by changes in OH concentrations, which control the major sink reaction. Changes in $O_3$, which is not directly emitted but only produced chemically in the atmosphere, depend on a variety of factors such as water vapour, UV radiation, reactive nitrogen oxides ($NO_x$), and hydrocarbon emissions (Fiore et al., 2012). Increases in $O_3$ concentrations lead to increases in OH concentrations, thus reducing the $CH_4$ lifetime; increases in $CH_4$ (or its oxidation rate) lead to increases in $O_3$. Atmospheric chemistry involves many processes that are strongly linked to climate change via temperature, water vapour, precipitation, and clouds (Fiore et al., 2009). In terms of chemistry–climate feedback loops, these can be positive or negative and involve changing GHGs through atmospheric chemistry (see feedback diagram in Fig. 15). These processes have so far mostly been quantified with chemistry–climate models with little or no constraints from observations.

Other established processes that link atmospheric chemistry with the carbon cycle on land that need to be included in ESMs are the deposition of nitrates (nutrients) and ozone (plant damage) (Ashmore, 2005; Sitch et al., 2007), which can influence carbon uptake and release from terrestrial ecosystems and biome distributions. The deposition of nitrate and ammonium occurs partly in the gas phase and

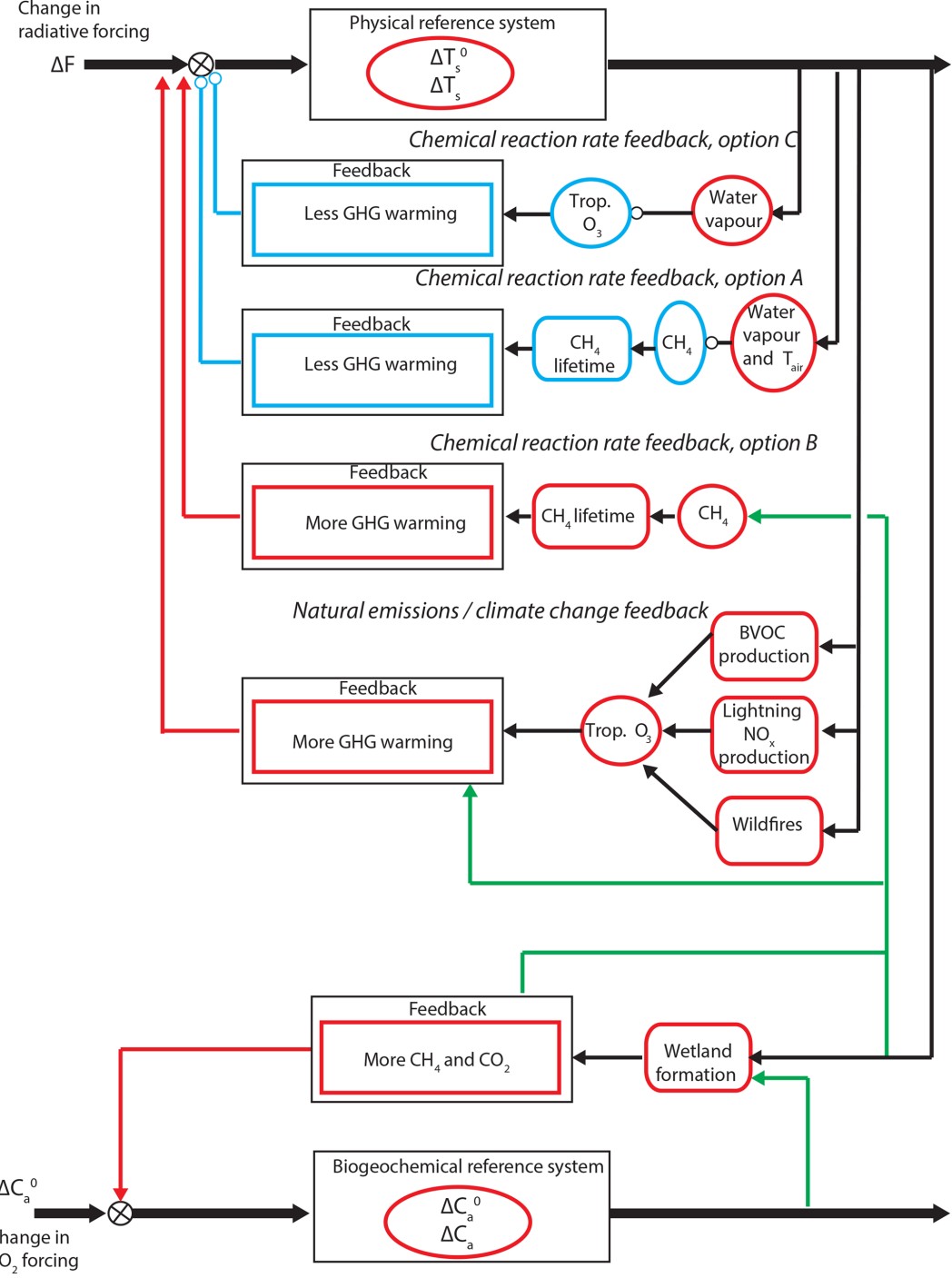

**Figure 15.** Schematic of important tropospheric chemistry feedbacks. Arrows indicate positive coupling; open circles indicate negative coupling. Changes in state variables are indicated by ellipses, changes in processes by rounded rectangles. Red indicates increasing values of variables, strengthening of processes, or positive feedbacks; blue indicates the opposite. Green arrows and lines indicate couplings of biogeochemical processes to physical processes. The change in surface air temperature $\Delta T_s$ in the presence of feedback is different from the change $\Delta T_s^0$ without feedback. The change in greenhouse gas concentration $\Delta C_a$ in the presence of a biogeochemical feedback is different from that without this feedback ($\Delta C_a^0$).

partly in the aerosol phase (see Sect. 4.3). Xu and Penner (2012) suggest that for $NH_x$ the total deposition in gas and aerosol phases is similar, whereas for nitrate, gas-phase deposition is larger by almost a factor of 2.

### 4.6.1 Feedback through changes to chemical reaction rates

Changing reaction rates in atmospheric chemistry involving the dynamics of $CH_4$, $O_3$, and water vapour can lead to both positive and negative climate feedbacks (Fig. 15). The major removal process for $CH_4$ is reaction with the OH radical, but OH is in turn removed by $CH_4$. This self-feedback has the effect of amplifying any changes to $CH_4$ production or removal by a factor $f$ (Prather, 1996). Parameter $f$ depends on the change in the lifetime of $CH_4$ with changing $CH_4$ concentration. It is defined as $1/(1-s)$, where $s = \delta \ln \tau / \delta \ln[CH_4]$ and $\tau$ is the $CH_4$ lifetime. Results from a recent chemistry model intercomparison suggest a value of $f$ of around 1.2–1.3 at present $CH_4$ levels, but rising to around 1.5 by 2100 in the high-methane, low-$NO_x$ RCP8.5 scenario (for representative concentration pathway, RCP, see glossary) (Voulgarakis et al., 2013). Best present-day $f$ values are given by Fiore et al. (2009) with a mean of $133 \pm 0.06$ and a range of 1.25–1.43. Increased temperatures will lead to increased oxidation rates and decreased $CH_4$ (Johnson et al., 2001). Voulgarakis et al. (2013) calculated a change in the $CH_4$ lifetime – due to the combined effects of temperature on the reaction rates and the water vapour concentration – of $-0.34 \pm 0.12$ yr $°C^{-1}$, a percentage change of $-3.6 \pm 1.3 \% °C^{-1}$. Because of the feedback of $CH_4$ to itself, which was not accounted for in these experiments, these values need to be scaled by the factor $f$ above, giving $-4.7 \pm 1.7 \% °C^{-1}$. For a present-day $CH_4$ concentration of 1803 nmol $mol^{-1}$ (Myhre et al., 2013), this translates to $-85 \pm 31$ nmol $mol^{-1} °C^{-1}$ equivalent to a climate feedback of $-0.03 \pm 0.01$ W $m^{-2} °C^{-1}$. Increased water vapour leads to increased $O_3$ destruction. Model simulations show that globally the effect of enhanced $O_3$ loss due to increased water vapour dominates, so that $O_3$ concentrations in the free troposphere decrease in a warmer and wetter climate (Royal Society, 2008; Stevenson et al., 2006). Stevenson et al. (2013) found varying responses to climate change in different chemistry models. They all showed decreased $O_3$ over the oceans, but some showed increased $O_3$ over the tropical continents where $NO_x$ emissions from lightning increased, and some showed increased $O_3$ in the upper troposphere around the subtropical jets due to increased stratosphere–troposphere exchange. The overall feedbacks were $-0.024 \pm 0.027$ W $m^{-2}$ for the years 1850–2000, $-0.025 \pm 0.025$ W $m^{-2}$ for 2000–2030, and $-0.033 \pm 0.042$ W $m^{-2}$ for 2000–2100. Assuming a warming of 4.5 K for RCP8.5 2100, this would give a climate feedback of $-0.007 \pm 0.009$ W $m^{-2} °C^{-1}$.

### 4.6.2 Feedbacks between natural emissions and climate change

Warmer temperatures and wetter climates are likely to increase the natural emissions of wetland $CH_4$, BVOCs, lightning $NO_x$, and wildfires providing positive feedback (Fig. 15). Wetland $CH_4$ emissions were modelled in WETCHIMP (Wetland and Wetland $CH_4$ Intercomparison of Models Project) (Melton et al., 2013). Increasing temperatures will lead to increased microbial generation of $CH_4$, but will also tend to dry out wetland areas. Models disagreed on the sign of the net impact of temperature on emissions. Increased precipitation will always lead to increased $CH_4$ from wetlands. Although not strictly a climate feedback, increased levels of $CO_2$ are expected to have the largest effect on wetland $CH_4$ by increasing the amount of organic carbon in the wetland soils. On longer (centennial to millennial) timescales, emissions from permafrost and $CH_4$ hydrates are likely to be important (O'Connor et al., 2010; see Sect. 4.3). Increases in $CH_4$ emissions also increase $O_3$ production.

BVOCs such as isoprene, terpenes, and other VOCs from land vegetation and ocean plankton are likely to increase in a warmer climate by 48–348 Tg(C) $yr^{-1} °C^{-1}$ (Arneth et al., 2010; Loreto and Schnitzler, 2010). In the free troposphere, enhanced BVOCs and their oxidation products lead to increased $O_3$ concentrations and to increased formation of SOA from these precursor gases (Heald et al., 2009; Lathière et al., 2006; Sanderson et al., 2003). However, the ambient $CO_2$ concentration and plant stress have been identified as another controlling factor, possibly inhibiting the emission of VOCs (Sharkey et al., 1991) and changing their composition. BVOCs will react with OH, but the subsequent reactions can also generate oxidants. Indirect observations of past OH concentrations (Montzka et al., 2011) suggest relatively small changes of less than 10 %, while studies on pre-industrial and glacial chemistry–climate interactions (Levine et al., 2011; Murray et al., 2014; Valdes et al., 2005) indicate that OH could change by around 25 % if BVOC and $NO_x$ emissions are decoupled. Therefore, the overall effect of BVOC emissions on $CH_4$ lifetime is unclear. Chapter 6 of the IPCC 5th Assessment Report (Ciais et al., 2013) assesses the biogeochemical feedbacks from wetland $CH_4$ emissions to be 0.02–0.1 W $m^{-2} °C^{-1}$ and those from BVOC emission to be 0–0.06 W $m^{-2} °C^{-1}$.

Increases in surface temperatures could lead to increased and more intense convection. This in turn would lead to higher flash frequency and $NO_x$ production from lightning (Price and Rind, 1993; Revell et al., 2015). However, current model simulations and lightning parameterizations show rather inconsistent results (Schumann and Huntrieser, 2007). The Atmospheric Chemistry and Climate Model Intercomparison Project (ACCMIP) found lightning $NO_x$ increases of around 20 %–30 % (1–2 Tg(N) $yr^{-1}$) by 2100 in the RCP8.5 scenario (Voulgarakis et al., 2013). Lightning is most prevalent in the tropics where it contributes to enhanced $O_3$ and

OH concentrations. Thus, lightning changes are likely to generate significant but opposing changes in the radiative forcing from $O_3$ and $CH_4$. The net impact will be very sensitive to the vertical profile of the $NO_x$ changes. At higher altitudes, $O_3$ production will be favoured; at lower altitudes, $CH_4$ destruction will be favoured.

Changes in temperature, (soil) moisture, and lightning modify the occurrence of natural wildland fires (Flannigan et al., 2000) (see Sect. 4.3.2). Furthermore, the intensity and extent of wildland fires, and therefore the amount of gases and aerosols released into the atmosphere, is controlled by wind speed and the degree of moisture in the fuel. Wildland fires release $CO_2$, trace gases, and aerosols (Akagi et al., 2011). The $CO_2$ feedbacks of fires are discussed in Sect. 4.3.2. Wildland fires generally lead to enhanced $O_3$ concentrations, predominantly in the lower troposphere (Aghedo et al., 2007; Schultz et al., 1998), whereas the net effect on the $CH_4$ lifetime is unclear and depends upon the relative emission factors of $NO_x$ and hydrocarbons. Fires can also lead to additional VOC emissions and subsequent secondary aerosol formation (see Sect. 4.5).

## 4.7    Stratospheric composition feedbacks

Most weather activity occurs CE13, and strong vertical and latitudinal mixing also prevails in the troposphere. Above the tropopause, in the stratosphere, air movement is less turbulent and marked by less updrafts than in the troposphere, and the temperature generally increases with altitude. The dominant dynamical, radiative, and photochemical processes are quite different in the stratosphere in comparison to the troposphere. For example, stratospheric trace gas changes induced by temperature changes in the troposphere surface system mainly affect the radiatively active trace gases water vapour and ozone. To a lesser degree, methane and nitrous oxide ($N_2O$) are also involved in feedback though the main part of their changes is from forcing (anthropogenic emission increase). Related stratospheric radiative feedbacks are not yet quantified well. Only few simulations with interactively coupled chemistry–climate models (CCMs) which include the ocean exist (Eyring et al., 2013). Future stratosphere radiative cooling, expected mainly from $CO_2$ increase, will also lead to changes in ozone, $CH_4$, and $N_2O$ through reduced chemical gas-phase loss rates (Haigh and Pyle, 1982). This radiative impact is to be regarded as part of the forcing under the effective forcing/feedback concept outlined in Sect. 2. We distinguish two main stratospheric composition climate feedbacks, which involve ozone and water vapour feedback (see feedback diagrams in Fig. 16): the negative dynamically induced lower-stratosphere ozone feedback and the positive stratospheric water vapour feedback via tropical tropopause temperature increase. Ozone and stratospheric water vapour feedbacks are closely coupled. The net effect of adding interactive stratospheric chemistry processes to an Earth system model is a negative feedback, at least for the $CO_2$-driven climate change.

### 4.7.1    Stratospheric ozone feedback

The joint radiative effect from projected ozone changes has been estimated to be negative, ranging between $-0.05$ and $-0.08$ $Wm^{-2}$ until the end of the 21st century (Bekki et al., 2013; Cionni et al., 2011). Ozone depletion since the late 1970s, due to anthropogenic emissions of ozone-depleting substances (ODSs) exclusively, causes an RF of around $-0.05 \pm 0.10$ $Wm^{-2}$ in 2005 (Forster et al., 2007; Hartmann et al., 2013). The stratospheric ozone depletion induces a cooling, both locally in the stratosphere but also to the troposphere surface system below (for the latter effect, both shortwave and longwave effects have to be considered). The ozone-forcing component arising for stratospheric chlorine loading is expected to cease during the course of the 21st century, but other forcing components will persist or even increase, for example, those related to increasing stratospheric $N_2O$ levels (Randeniya et al., 2002; Revell et al., 2012). CCMs consistently simulate a Brewer–Dobson circulation (BDC; see glossary) increase in a warmer climate (Butchart, 2014; Butchart et al., 2010; Eyring et al., 2010b; SPARC CCMVal, 2010). This intensification, mainly triggered by tropical sea surface temperature changes (Butchart, 2014; Deckert and Dameris, 2008; Garny et al., 2011; Zhou et al., 2012), implies an increase in tropical upwelling that leads to faster air transit and less time for ozone production (Bekki et al., 2013; Meul et al., 2014; Oman et al., 2010; WMO, 2011). The effect is illustrated in the feedback diagram of Fig. 16 (dynamically induced lower-stratosphere ozone feedback) and largely controls changes (decrease) in lower stratospheric ozone inside the tropical belt. There are also indications of a compensating effect in the extratropical lower stratosphere, where strengthened subsidence from the BDC increase transports more ozone-rich air downwards (McLandress and Shepherd, 2009). This would imply positive radiative feedback at the tropopause, but it is masked or even reversed (Nowack et al., 2015) by an ozone decrease at extratropical latitudes located in a thin layer around the tropopause. Dietmüller et al. (2014) relate the latter feature to the lifting of the extratropical tropopause in the warmer atmosphere. However, the global mean ozone radiative feedback is clearly dominated by the ozone decrease in the tropical lower stratosphere and is, thus, negative. The effect is strong enough to outweigh positive contributions of those ozone increases at higher altitudes that should be regarded as a rapid adjustment to $CO_2$-induced radiative cooling of the stratosphere (Dietmüller et al., 2014; Jonsson et al., 2004; Nowack et al., 2015). We should add that if radiative forcings originate from other perturbations than $CO_2$ increase, e.g. from changing solar insolation, this may induce distinctly different stratospheric ozone feedbacks (Chiodo and Polvani, 2016), but this has been even less explored. CE14 As for stratospheric-ozone

Earth Syst. Dynam., 10, 1–74, 2019

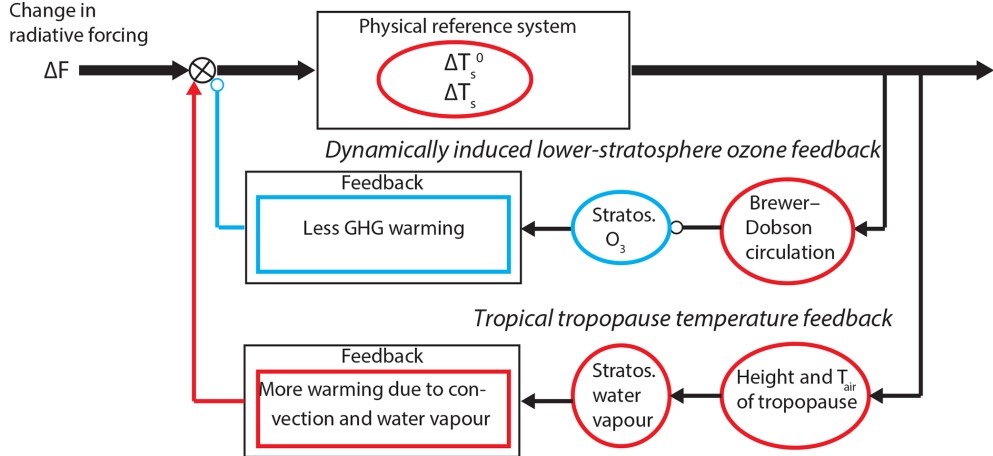

**Figure 16.** Schematic of important stratospheric chemistry feedbacks. Arrows indicate positive coupling; open circles indicate negative coupling. Changes in state variables are indicated by ellipses. Red indicates increasing values of variables, strengthening of processes, or positive feedbacks; blue indicates the opposite. The change in surface air temperature $\Delta T_s$ in the presence of feedback is different from the change $\Delta T_s^0$ without feedback.

feedbacks to $N_2O$, and $CH_4$ increases, only their rapid adjustment component has been addressed so far (Kirner et al., 2015).

### 4.7.2 Stratospheric water vapour feedback

Positive global climate feedback is introduced through an increase in tropopause height and temperature with global warming, leading to an increase in stratospheric water vapour concentrations (see Fig. 16, tropical tropopause temperature feedback). Its main part is triggered by changing sea surface temperature affecting tropospheric convective mixing (Garfinkel et al., 2018; Stuber et al., 2005; Santer et al., 2003), though tropopause dynamics are complex in themselves and there are several other mechanisms involved in the process (Birner and Charlesworth, 2017; Lin et al., 2017; Thuburn and Craig, 2000). CCMs also simulate a decrease (increase) in tropical tropopause pressure (height) over recent decades, largely in agreement with observations (Austin and Reichler, 2008; Garfinkel et al., 2013; Kim et al., 2013; Son et al., 2009; SPARC CCMVal, 2010). Current CCM projections continue this trend into the 21st century, associated with a slow increase in the tropical cold-point tropopause (CPT) temperature of around 0.5–1.0 K per century (Chapter 7 of SPARC CCMVal, 2010). Consistent with the warming tropical tropopause, the models simulate an increase in stratospheric water vapour throughout the 21st century (Gettelman et al., 2010), which increases the respective positive stratospheric water vapour feedback. These projections of gradual stratospheric moistening are, however, not fully consistent with the marked stratospheric water vapour decrease observed after 2001. Current CCMs fail to capture this decrease, except if nudged towards the observed dynamic evolution (Brinkop et al., 2016). Evidently, the interplay of var-

ious processes that force stratospheric water vapour changes (Dessler et al., 2013; Randel et al., 2006; Solomon et al., 2010) is still not fully understood, preventing a clean distinction between adjustment, forcing, and feedback contributions at this stage.

## 5 Feedback evaluation outlook

We have now introduced the various major known climate feedbacks in the Earth system. In reality, the different feedback mechanisms work simultaneously and interact with each other. Because the Earth system is complex, ESMs have been developed that simulate the simultaneous action of feedbacks and interactions between them. How can we determine which feedbacks may be quantitatively more important than others, and how can we constrain them? Can we develop strategies and tools that raise our confidence in the predictive results of the models? We structure the overview of different approaches to evaluate feedbacks into methods that work within the model world (independently of observations) and those that involve observations.

### 5.1 ESM forcing and feedback evaluation within the model world

CE15 Quantifying rapid adjustments for computing effective forcings and determining climate feedback – either aggregated or in isolation – are complex cross-disciplinary tasks. So far, neither a unified method nor standardized metrics for such an evaluation have been developed that consider all Earth system reservoirs and processes, and there is no reason there should ever be a unified and standardized approach that would fit all purposes. This is due to the complex couplings between different feedback loops from the physical

and biogeochemical realms as described above. We summarize in this and the following section the different general approaches in order to assess the importance of climate feedbacks and how well they may be represented in models in a quantitative way. Evaluation approaches aiming at separating and quantifying the strengths of individual feedback processes are often carried out only within the model world, i.e. without comparing the results directly with observations as these may be lacking or of insufficient temporal and spatial coverage.

### 5.1.1 Regression method and fixed SST method

For the determination of the rapid adjustments, the so-called regression method has been applied (Andrews et al., 2012; Forster et al., 2013; Gregory et al., 2004): ESMs (without fixing any variables) are exposed to an abrupt strong forcing (usually 4 times $CO_2$ pre-industrial forcing). The subsequent change in the net radiative balance in relation to the change in surface temperature is then

$$N = F - H = F - \alpha \Delta T,$$

with $N$ being the downward heat flux at the top of the atmosphere, $H$ the radiative response, $F$ the forcing, $\alpha$ the climate response parameter (a measure of the net climate feedback strength), and $\Delta T$ the average surface air temperature change (see Gregory et al., 2004). When plotting $N$ against the evolving $\Delta T$ values after (earlier on, usually with the first 10–20 years of the instantaneous abovementioned $CO_2$ change; meanwhile, longer integration times of ca. 150 years are used), the corresponding data points provide a regression line which can be extrapolated to $N = 0$ for an approximate value of the equilibrium climate sensitivity and which can be extrapolated to $\Delta T = 0$ for a quantification of the effective radiative forcing including rapid adjustments. By using the output of CMIP5 models, Andrews et al. (2012) could further differentiate the contribution of the longwave clear sky, shortwave clear sky, and longwave and shortwave cloud radiative effect components of the overall feedback. In a more recent study, Andrews et al. (2015) identified that the linear relationship between $N$ and $\Delta T$ breaks down for longer model runs (more than 20 years) especially due to evolving changes in SST patterns.

An alternative to the regression method, longer runs with full ESMs, can be used to diagnose the effective radiative forcing through the so-called fixed SST method (Hansen et al., 2002, 2005). Required model runs under perturbed forcing cover only a few weeks (for all ground temperatures fixed; Fig. 4c) or from several decades (typically 30 years) to up to 120 years (only SSTs fixed; Fig. 4d) depending on the nature of the forcing. Where necessary, the forcing of different agents can be computed separately through respective sensitivity experiments. By identifying different efficacies of these agents relative to $CO_2$ forcing, different combinations of forcings can thus be converted into an overall

radiative forcing. The ERF values are associated with uncertainties from the spread in model representations of the rapid adjustments, from the idealized conditions of strong forcings applied rather than a gradual increase, and from statistical uncertainty due to internal climate variability in the underlying multi-year simulations.

### 5.1.2 The partial radiative perturbation method

The fast physical feedbacks of changes in water vapour, lapse rate, clouds, and surface albedo have been specifically determined separately from the output of climate projections with several ESMs. To this end, the output of a model ensemble (see glossary) is used as input to the radiation code of a single ESM. The feedback effect of changes in single climate variables on the top of the radiative forcing can thus be determined when the analysis is at least carried out between two points in time. This partial radiative perturbation method (PRP) going back to Wetherald and Manabe (1988) has been successfully applied to modern ESM output (Bony et al., 2006; Soden and Held, 2006; Vial et al., 2013; Zelinka et al., 2013). An established method to approximate PRP results is the resource-efficient use of pre-calculated radiative kernels (see glossary), which allow the derivation of radiative feedbacks directly from parameter changes without the need for re-calculating the radiative transfer for each case (e.g. Soden et al., 2008). A good example of the results of the PRP method for various classical feedbacks is given in Fig. 17 (corresponding to Fig. 5b in Vial et al., 2013). For these feedbacks, the concept of equilibrium climate sensitivity (ECS) can be applied. Prospects for narrowing bounds on ECS have been provided by Stevens et al. (2016). Observation based estimates (especially those involving palaeoclimate data) suggest an even larger range of equilibrium sensitivities than those shown in Fig. 17 (Knutti et al., 2017).

### 5.1.3 Separation of feedbacks with partially uncoupled ESM runs

Let us now have a look at the Earth system feedbacks, which go beyond the conventional physical feedbacks in the climate system. For estimating feedbacks, ESM experiments are carried out under future scenario forcing (often the idealized scenario with 1 % $CO_2$ yr$^{-1}$ increase in atmospheric $CO_2$ is used as model runs are short, i.e. only 70 years until atmospheric $CO_2$ concentration doubles with respect to the pre-industrial start value; a critical appraisal of the 1 % $CO_2$ yr$^{-1}$ increase scenario is given in MacDougall, 2019). First, a *fully coupled run* with the ESM in question is carried out. Afterwards, the model run is repeated, but with one or more feedback loops switched off. From the difference between this *partially uncoupled run* and the *coupled run*, the strength of the feedback of interest can be estimated. This method has been specifically applied to separate the carbon cycle feedback due to climate change (increasing tempera-

ture) from that due to chemical forcing (rising $CO_2$ emissions to the atmosphere) (Arora et al., 2013; Friedlingstein et al., 2003, 2006). The change in land and ocean carbon storage can be approximated by

$$\Delta C_L^c = \beta_L \cdot \Delta C_A^c + \gamma_L \cdot \Delta T^c,$$
$$\Delta C_O^c = \beta_O \cdot \Delta C_A^c + \gamma_O \cdot \Delta T^c,$$

with $\Delta C_L^c$ and $\Delta C_O^c$ representing the changes in land and ocean carbon storage in the fully coupled ESM simulation due to the physical forcing $\Delta T^c$ and the chemical forcing $\Delta C_A^c$. This is, of course, a strong simplification because all physical climate change is simply projected onto a temperature change. The coefficients $\beta_L$ and $\beta_O$ are the carbon sensitivities to rising $CO_2$. The coefficients $\gamma_L$ and $\gamma_O$ are the carbon sensitivities to rising temperature (the indices L and O stand for land and ocean). While the $\beta$ factor can be determined from an uncoupled run, where the carbon cycle "sees" only the $CO_2$ forcing and not the climate change forcing, the $\gamma$ factor can be determined to first order from the difference between the coupled run and the uncoupled run. In principle an uncoupled run also needs to be added where the carbon cycle does not see any chemical forcing (Arora et al., 2013; Gregory et al., 2009). Large positive values of the $\beta$ and $\gamma$ factors translate into negative climate feedback. Uncertainties arise, as the various feedbacks in reality do not add up linearly due to more complex interactions between the Earth system reservoirs (see Schwinger et al., 2014). This method can be applied to many other climate change processes (as well as shorter-term feedbacks; see Schneider et al., 1999). Its disadvantage is that for each process, additional computationally demanding model runs have to be carried out. In general, the method is not limited to $CO_2$ feedback but can also be applied to other species such as methane or aerosols. Fig. 18 shows the result of a corresponding feedback analysis from CMIP5 models (Arora et al., 2013) for an analysis of nine ESMs including an interactive carbon cycle (land, ocean, atmosphere) under a 1 % yr$^{-1}$ $CO_2$ increase in the atmosphere (until $CO_2$ doubling, which happens after 120 years). Because the atmospheric $CO_2$ concentration has been prescribed in this case, one has to diagnose the feedback in terms of "compatible emissions"; i.e. one asks how much $CO_2$ emissions would be allowed in each model system in order to arrive at the 1 % yr$^{-1}$ $CO_2$ increase in the atmosphere.

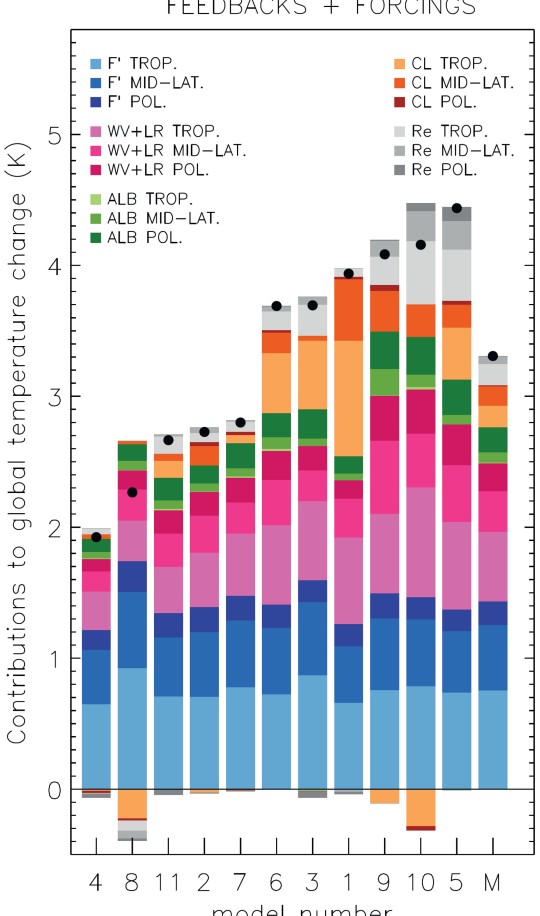

**Figure 17.** Climate sensitivity estimates (for equilibrium at $CO_2$ doubling as given by the black dots) associated with the Planck response to the stratosphere-adjusted forcing and the adjustments ($F'$), feedbacks ("classical feedbacks"), and forcings for 11 different ESMs according to Vial et al. (2013). The colours' hue denotes the regional range (tropical, mid-latitude, and polar). $F'$ denotes the Planck response to stratosphere-adjusted forcing and other adjustments to $CO_2$ forcing and land surface warming, WV+LR denotes the combined water vapour and lapse rate feedback, ALB the albedo feedback, and CK the cloud (CL) feedbacks. Re is the feedback residual term. The numbers denote the following models: (1) IPSL-CM5A-LR; (2) NorESM1-M; (3) MPI-ESM-LR; (4) INMCM4; (5) HadGEM2-ES; (6) CanESM2; (7) MIROC5; (8) CCSM4; (9) BNU-ESM; (10) FGOALS-s2; and (11) MRI-CGCM3. For details, see the original source: Vial et al. (2013) and Fig. 5b therein. Reprinted by permission from: Springer, Climate Dynamics, Vial, J., et al., On the interpretation of inter-model spread in CMIP5 climate sensitivity estimates, 41, 3339–3362, ©Springer (2013).

## 5.2 ESM feedback evaluation involving observations

Here we introduce general feedback evaluation approaches using a combination of models and observations. Specific feedback evaluation options are given in Appendix A. A key task for feedback evaluation is to compare changes in forcing agents and climate state variables with respect to time and space among models and measurements. Due to the model complexity and the increasing number of model systems and

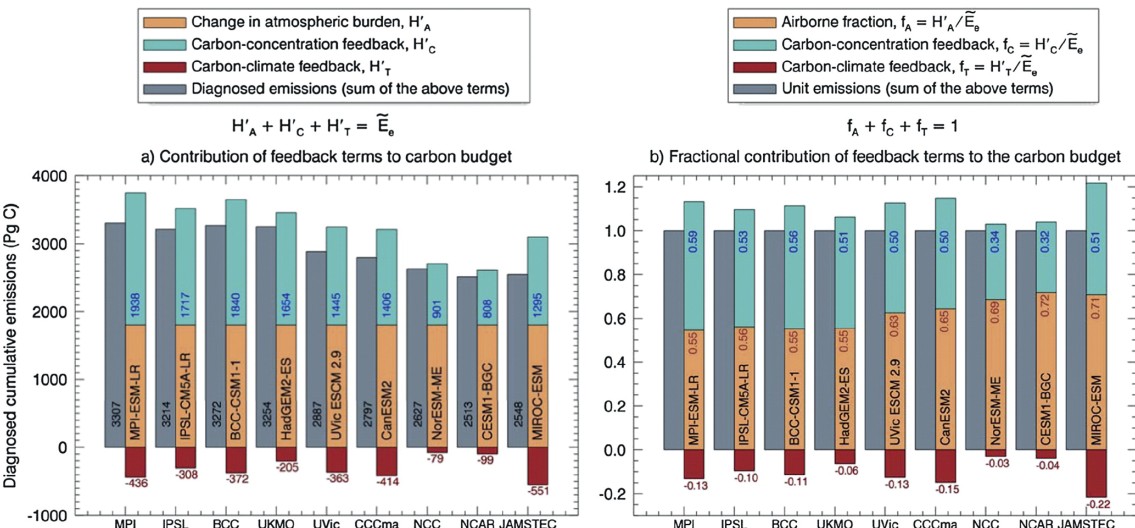

**Figure 18.** Contributions of the carbon-concentration feedback and the simplified carbon cycle–climate feedback to "compatible $CO_2$ emissions" to the atmosphere. Large diagnosed emissions correspond to more negative feedback than low diagnosed emissions. **(a)** Absolute contributions in PgC yr$^{-1}$ and **(b)** fractions of compatible cumulative emissions. The atmosphere contribution in panel **(a)** is constant because the atmospheric $CO_2$ concentrations were prescribed in this experimental set-up. For details, see the original source: Figure 7 in Arora et al. (2013). © American Meteorological Society. Used with permission.

climate state variables, the evaluation of ESMs requires considerable infrastructure resources (Eyring et al., 2016b). Attempts are made to facilitate this task through standardized tools that can be used by the Earth system modelling community (Eyring et al., 2016c; Collier et al., 2018).

### 5.2.1 Observations and signal-to-noise ratio

Assessments of model results (model output variables) are based on evidence from the real world: observational data sets for state variables of the climate system or fluxes of essential properties that are important to climate feedbacks (in situ observations, satellite observations, proxy data); field experiments where natural systems have been purposefully perturbed (free-air carbon dioxide enrichment experiments, mesocosms – see glossary; Liberloo et al., 2009; Riebesell et al., 2007); and case studies (such as volcanic eruptions and their climatic consequences, e.g. Kandlbauer et al., 2013, or extreme heat events, e.g. Reichstein et al., 2007). Direct observations from the instrumental record for the relevant state variables from the various Earth system reservoirs cover only a short period (a few decades at most). Often only sparse observational data sets exist. As a necessary condition in order to attribute a change in climatic state variables to specific radiative forcings and radiative feedbacks, the resulting signal in these variables (relative to a reference state) must be clearly distinguishable from noise. The climate response due to human-caused forcing perturbations must be discernible from natural variability. For future climate change projections on timescales beyond a few decades, internal variability of the climate system is expected to be smaller than the

overall system change in response to anthropogenic forcing. The time of emergence (ToE) – i.e. the time at which the climate change signal emerges from the noise of natural climate variability at the grid-point scale – has been estimated on the basis of CMIP3 climate models to be 30–60 years for surface air temperature depending on the regions considered (Hawkins and Sutton, 2012); however, for selected precipitation change hot spots, shorter ToEs have also been suggested (Giorgi and Bi, 2009). Once changes are aggregated over regions, the anthropogenic signal is much more obvious: changes in temperature, the water cycle, Arctic sea ice, and even extreme events have already been detected and attributed to human influence (Bindoff et al., 2013). Changes in some ocean biogeochemical variables may show shorter ToEs than sea surface temperature (Keller et al., 2014). ToEs for climate-induced changes in land ecosystems are in the same range as for surface temperature, with some shorter ToEs in regional hot spots (Lombardozzi et al., 2014). Careful analysis of requirements for ESM evaluation needs to be considered alongside emerging new observing systems in order to enable improved model evaluation in the decades to come (see Ferraro et al., 2015). There have also been attempts to use global energy budget quantifications in the past 2 decades as a basis for estimating the equilibrium climate sensitivity and the transient climate response (e.g. Otto et al., 2013). Slower feedbacks such as the carbon cycle climate feedback may not be compatible with such estimates due to the short observational period.

### 5.2.2 Control run evaluation with climatological data sets from observations

ESMs are spun-up from initial conditions (either idealized or based on observed climatologies) over a longer period until the output variables approach a quasi-steady state. The initialization and spin-up procedures differ from model system to model system (e.g. Séférian et al., 2016). Differences in simulation results appear when using these diverging procedures. After the models reach quasi-steady state, a longer control run is carried out (Séférian et al., 2013; Zorita et al., 2003). Results for state variables are compared with observations to assess model biases. Control run results represent the pre-industrial situation. However, present-day climatologies from direct measurements are usually employed for comparison, introducing an element of incompatibility. The evaluation of the quasi-steady-state control run is not a feedback analysis per se, but some of the present-day patterns of temperature, precipitation, clouds, etc., are determined by the balance between various feedbacks. Therefore, a good representation of the present-day climatology is an indication that many feedbacks at least co-act in a realistic way. The satisfactory simulation of general spatio-temporal structure of climate state variables (see glossary) is a prerequisite for the subsequent potentially realistic reaction of the respective ESM to anthropogenic forcing. For example, in CMIP5, the ESM with fastest Atlantic Ocean meridional overturning showed corresponding fast ocean uptakes of heat and carbon and a potential overestimation of related negative climate feedbacks (Arora et al., 2013).

### 5.2.3 Evaluation of historical simulations

A first step towards concretely estimating the feedback behaviour in ESMs is running historical simulations from the start of the industrial revolution until the present day ("historical runs") or subsets of this period (Flato et al., 2013). During the 20th century, an increasing amount of direct observations from the instrumental record became available for comparison with the model results. Not only is the climatological mean state of relevant climatic state variables over certain time intervals considered but also the seasonal cycle, interannual variability, the reproduction of variability patterns following the dominant climate variability modes (see glossary: El Niño–Southern Oscillation, ENSO; North Atlantic Oscillation, NAO; Pacific Decadal Oscillation, PDO; Southern Annular Mode, SAM; Madden–Julian Oscillation, MJO; Quasi-Biennial Oscillation, QBO), and changes in these patterns over time (Flato et al., 2013; Kim and Yu, 2012; Lenton et al., 2009; Li et al., 2015; Pascale et al., 2015; Phillips and Bonfils, 2015). As ESMs produce their own internal climate variability and also their own "weather", only the statistical behaviour of the models and the principle sequence of events associated with typical climate variability modes can be compared with observational data. Further, longer-term trends in observed and simulated climatic state variables can be compared (Santer et al., 2008). Flux estimates for matter transport (such as carbon fluxes) among Earth system reservoirs resulting from historical model runs ("forward models", "bottom–up approach") are examined using data-driven approaches ("inverse models", "top–down approach") (Séférian et al., 2013; Sitch et al., 2015; Wanninkhof et al., 2013). State-parameter estimation, systematically combined through data assimilation, where free model parameters such as diffusion coefficients are calibrated through an optimal fit of modelled state variables to observations, is still a novel approach in coupled climate models (e.g. Liu et al., 2014; Zhang et al., 2015). Computational demand and problem complexity are huge. New emerging data assimilation approaches may allow for the feasible implementation of combined state–parameter optimization algorithms for coupled ESMs (Gharamti et al., 2015; Simon et al., 2015).

### 5.2.4 Evaluation with palaeoclimatic data

Palaeoclimatic experiments with ESMs can be useful for assessing the models' ability to account for slow feedbacks and for constraining the sensitivity of models to forcings in general. The general concept is to expose ESMs to reconstructed anomalies in forcing, to diagnose the models' response, and to compare the model results with palaeoclimatic observations. Model forcings for respective experiments are taken from orbital parameter variations of the Earth (eccentricity, axial tilt, precession; Berger and Loutre, 1991), solar activity indices, volcanic eruption records, and different ice sheet topographies. Typical test events for simulations with ESMs include the Last Glacial Maximum (LGM; 21 kyr BP, important for quantifying the positive carbon cycle climate feedback) (Braconnot et al., 2007a, b; Frank et al., 2010; Schmidt et al., 2014) and the last 1000 years including the Maunder Minimum (300 BP; Little Ice Age mechanisms) (Ottera et al., 2010; Zorita et al., 2005). Observational data used in comparison with ESM results are based on the marine and terrestrial palaeoclimate record (such as stable carbon and oxygen isotopes from sediment core analysis, pollen analysis, and bore hole temperatures; see, e.g., Bradley, 1999). Palaeoclimatic observations consist of proxies, i.e. preserved environmental characteristics that replace direct measurements of the instrumental record. These proxy records contain a climate signal but embedded in a suite of other influences of non-climatic origin (Bradley, 1999). Specific links between proxy records and climate state variables rely on respective empirical transfer functions. Proxy data are therefore associated with a considerable uncertainty range. This deficiency is to some degree compensated for by the higher signal-to-noise ratio of the respective variations in climatic state variables during certain time intervals within the Quaternary. On the other hand, modified ice sheet states and sea-level positions for dates older than a few thousand years complicate ESM simulations. Cause–effect links for changes in specific feedback processes

may thus be masked by other processes. We now give a few examples of useful palaeoclimatic studies to assess feedback strengths. Frank et al. (2010) employed climatic forcing data over the past millennium with observations from ice cores in order to constrain the carbon cycle feedback to temperature changes to the lower half of the range than that inferred from projections by ESMs (Friedlingstein et al., 2006). A comparison of simulations with ESMs under forcing conditions for (a) the Last Glacial Maximum and (b) the mid-Holocene provided indications of the strength of the vegetation climate feedback (inducing changes in the evapotranspiration) and the albedo feedback due to changes in snow cover and sea ice (Braconnot et al., 2007b). The various resulting feedback strengths can be weighted through a rigorous comparison of model results and observational palaeoclimatic data following a maximum-likelihood approach. ESMs can also be used for simulating the various palaeoclimatic time windows as given in PALEOSENS Project Members (2012) in order to calibrate their sensitivities.

### 5.2.5 Emergent-constraint approach

A pragmatic way of narrowing down the uncertainties of climate sensitivity to changes in forcing has recently been suggested with the emergent-constraint approach (e.g. Cox et al., 2013, 2018; Hall and Qu, 2006). Emergent constraints "are relationships across an ensemble of models between some aspect of Earth System sensitivity and an observable trend or variation in the contemporary climate" (Flato et al., 2013). The approach thus establishes a relationship between the long-term Earth system sensitivity, which cannot be easily observed, and a short-term (often regionally confined) sensitivity (or a quantity that can be related to such a sensitivity), which is accessible to observations. Examples of such relationships have been established for physical climate components (see, e.g., Allen and Ingram, 2002; Bracegirdle and Stephenson, 2012; Hall and Qu, 2006; Klein and Hall, 2015; Sherwood et al., 2014) as well as (biogeo-)chemical Earth system components (see Bracegirdle and Stephenson, 2013; Cox et al., 2013; Hoffman et al., 2014; Massonnet et al., 2012; Wang et al., 2014; Wenzel et al., 2014, 2016a) on the basis of multi-model ensembles with Earth system models. Figure 19 provides an example for an emergent constraint concerning surface albedo change with warming. Cox et al. (2018) provide a revised estimate for equilibrium climate sensitivity using natural temperature variability as a constraint. Cox et al. (2013) derived a quasi-linear relationship between (a) the climate sensitivity of tropical land carbon storage and (b) the sensitivity of the atmospheric $CO_2$ growth rate to tropical temperature for a subset of carbon cycle climate models as employed in the Coupled Climate–Carbon Cycle Model Intercomparison Project ($C^4MIP$). While (a) is only model derived, (b) can be derived from both models and observations. The observed relationship among variations in the growth rate of atmospheric $CO_2$ and tropical tempera-

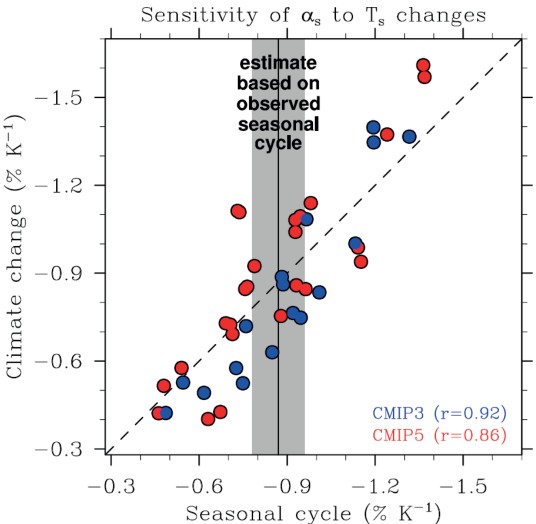

**Figure 19.** Change in the "sensitivity" of surface albedo $\Delta\alpha_s$ over change in surface temperature $\Delta T_s$ of the Northern Hemisphere landmasses ($\Delta\alpha_s/\Delta T_s$) in terms of climate change ($y$ axis) versus in terms of the seasonal cycle ($x$ axis). Blue: results for different models in CMIP3 including the correlation coefficient for best fit. Red: same for CMIP5. Each blue and red dot represents one model run with one specific ESM. The vertical black line indicates the observed value for the seasonal cycle; the grey shading indicates the uncertainty of the observation-derived value. Source: Klein and Hall (2015), who redrew their figure from Hall and Qu (2006) and Qu and Hall (2006). Reprinted by permission from: Springer, Klein, S. A. and A. Hall: Emergent Constraints for Cloud Feedbacks, Curr. Clim. Change Rep., 1, 276–287, ©Springer (2015).

ture was used to put different weights on the various ensemble members (different model systems) when computing the ensemble average. Stronger weighting for models close to the observational constraint was applied. Thus, narrowing of the probability density function (PDF) for the projected climate sensitivity could be achieved. Sherwood et al. (2014) could associate the spread in climate sensitivity of different Earth system models with the strength of convective mixing. By constraining this strength through radiosonde data, they concluded that a climate sensitivity of 3° and higher is more likely than a low sensitivity of around 1.5°. Details in the analysis procedure and the underlying model characteristics influence the results of the emergent-constraint approach (Bracegirdle and Stephenson, 2013; Wang et al., 2014; Wenzel et al., 2014). This also applies to the ensemble size of models, where caution is needed especially when using small ensembles (Caldwell et al., 2018).

## 6 Discussion

In this article, we provided an overview of known climate feedbacks and of methods how to constrain them. It would be desirable to quantify the various feedbacks in the Earth

system in terms of their magnitude and uncertainty. Such quantification could help guide the design of the respective observing and analysis systems. At present, there is, however, no method available that could be used to quantify all climate feedbacks present in state-of-the-art ESMs. Results of analysing feedback strengths are reported using different characteristic numbers and indicators, which refer to different underlying forcing scenarios and time intervals. In addition, there are different concepts concerning the definition of feedbacks in specific contexts and their interdependencies. But even if these technical issues are resolved, the ranking of feedbacks would still depend on the region and quantity of interest: different feedbacks matter for different things such as changes in specific state variables and related impacts. Further, significant feedbacks may still have been overlooked and may be missing in the models. It is expected that the complexity of ESMs will increase further, especially when adding socio-economic feedbacks, other feedbacks associated with the human factor, and long-term geological feedbacks including ice sheets and land–ocean matter redistributions through the weathering–sediment cycles (exogenous dynamics) and tectonic cycles (endogenous dynamics). Any threshold for considering a feedback as important or not will include a high level of ambiguity at this stage. This ambiguity already starts with the differentiation between system response and feedback.

Both fast and slow feedbacks are associated with considerable uncertainties. Almost all biogeochemical feedbacks have high uncertainties. At present, the key evaluation methods, for fast feedbacks, are hindcasts and variability analysis from the time of the instrumental observational record. For slow feedbacks, a greater diversity of evaluation options has been used. Comparison with data sets from climatological and time-dependent observations is carried out. Evaluations against the palaeo-record and against case studies both seem to be viable options for constraining slow feedbacks. The still quite new emergent-constraint approach has so far mainly been applied to surface-related feedbacks (snow, carbon fluxes, and sea ice) which may be ideally suited for this method with regard to timescales and observational records. While fast feedbacks are mainly due to physical processes, biogeochemical processes come more into play on longer timescales and then often in more complex feedback loops involving physical processes as well. The quantitative key climate feedbacks on long timescales are associated with the negative ocean feedbacks of overturning circulation and inorganic carbon buffering and the land carbon feedbacks, which may change sign over time when it comes to integrated carbon uptake or release. Slow Earth system feedbacks often include a partial fast-reacting component as well. For example, the feedbacks due to land $CO_2$ fertilization and inorganic marine carbon chemistry effectively reduce the atmospheric $CO_2$ concentration increase as caused by yearly anthropogenic $CO_2$ emissions, though a full equilibration of

the carbon cycle after the cessation of emissions would require $10^5$ years or longer.

Evaluating Earth system feedbacks of climatic relevance remains a challenge due to the complexity of the problem and the lack of guiding observations. Increasing model process complexity and resolution may increase the difficulty in evaluating ESMs and their feedback simulations. Nevertheless, the systems approach as pursued through Earth system modelling has its own strengths and a number of approaches can be taken to constrain feedback quantifications in the years to come:

1. *Internal consistency – strength of Earth system models.* ESMs are in principle internally consistent under energy and matter conservation. A change in one Earth system reservoir should induce changes in the others through a chain or network of processes reacting to and inducing changes in Earth system state variables. The existence of many stable ESMs simulations without flux adjustment (see glossary) that show many similarities with the real world is a major achievement (Flato, 2011; Flato et al., 2013). The likelihood that all model components are wrong and merely compensate for errors is not zero but is fairly small.

2. *Multi-tracer approach.* The output variables of ESM simulations are compared against a multitude of climate state variables and process tracers as derived from observations. Given that ESMs with further improvements will reproduce a large number of very different types of observations in a satisfactory way with regional and temporal variability patterns, this will add to their credibility also with respect to simulating climate feedback loops.

3. *Consistent tuning to reduce degrees of freedom.* Less arbitrary model tuning is needed. Often a huge number of tunable coefficients are included in models that are not sufficiently constrained by observational evidence. They are adjusted so that the overall reproduction of climatic state variables looks reasonable or "best possible" to the modelling groups or in view of a metric agreed on by the scientific community. A part of this tuning can be done systematically based on cause–effect links of model parameters that are well understood (Mauritsen et al., 2012). Sometimes a more suitable choice of free model parameters can be made, which reduces the fit of the modelled state variables to observational data but can be more correct in the physical or chemical sense. For example, the strength of the ocean overturning circulation – a key variable of climate feedback – needs to be checked against an age tracer such as natural radiocarbon. Other oceanic tracers such as dissolved inorganic carbon, dissolved nutrients, and gases must show similar potential biases to the simulated radiocarbon, otherwise the respective ocean model compo-

nent is physically and chemically inconsistent. Such a "consistently wrong" model would be preferable (and, of course, subject to further improvement) to a model where some state variables are artificially tuned to give results, which may fit the observations better, but which are not correctly based on physical or chemical principles.

4. *Data assimilation and ranking.* Though data assimilation of coupled Earth system models is still in its beginnings (e.g. Zhang et al., 2007), its potential for improving climate feedback representations in ESMs is large. Some methods of combined parameter-state estimation allow a ranking of different process parameters with respect to their determinacy and importance and a ranking on which observations the optimal estimation mainly relies. Thus, insensitive feedback parameters and irrelevant observational constraints can be filtered out and model systems can potentially be simplified.

5. *Purposeful falsification.* Further one should attempt to use ESMs in non-standard conditions in order to see whether ESMs can be falsified. An illustration of this would be the emergent-constraint approach. This approach derives its value from a relationship of a specific simulated climate sensitivity ($A$) to a different climate sensitivity ($B$) that in contrast to $A$ is constrained by observations. The linear relationship of $A$ to $B$ is purely model-derived through multi-model ensembles. Attempts should be made to arrive at a different linear relationship, $A'$, with, for example, an offset with respect to the original solely model-derived sensitivity $A$. This could be achieved in spinning all models up with a slightly modified land sea distribution or with a different total Earth system inventory of carbon or slightly modified solar insolation. Relevant research could identify the robustness of the emergent-constraint approach and the respective results concerning climate sensitivity and feedback strength. Likewise, for narrowing down the range of potential ECS values, Stevens et al. (2016) suggest an approach of "refutational reasoning" by exploring storylines for exceeding or falling below specific bounds for ECS and testing them against evidence.

## 7 Conclusion

Within an Earth system context, many more climatically relevant feedbacks influence climate projections under given forcing scenarios than in previous generations of physical climate models. In addition to the classical physical climate feedbacks, biogeochemical feedbacks are also considered in more complex Earth system models. Often biogeochemical feedback loops are coupled to physical feedback loops. The quantification of these feedbacks and their effect on climate state variables is not straightforward, even if the principle

feedback process is identified. This shows, on the one hand, that ESMs are needed in order to assess the joint action of different feedbacks. On the other hand, it reveals gaps in our understanding with respect to regional characteristics of feedbacks, which are important for regional and impact assessments (Maraun et al., 2017). Various options are available for evaluating the feedbacks and processes in the model world and with the help of measurements (see also Eyring et al., 2018). In order to make progress towards the quantification of Earth system feedback processes, suitable strategies for reducing free model parameters and simplifications of models need to be envisaged in parallel to the ever-increasing addition of new process descriptions in ESMs. One needs a hierarchy of models in addition to observations to make progress in understanding key climate feedbacks.

Opportunities also arise from the experimental design of CMIP6 (Eyring et al., 2016a). The RFMIP (Pincus et al., 2016) is a CMIP6-endorsed MIP (model intercomparison project) that specifically focuses on diagnosing forcings and robust responses and includes "fixed SST" simulations to diagnose the forcing ("RFMIP-lite"). Similarly, the Aerosol and Chemistry MIP (AerChemMIP; Collins et al., 2017) focuses on diagnosing forcings and feedbacks from near-term climate forcers.

In addition to the CMIP6 experimental design, progress in constraining individual feedbacks and in equilibrium climate sensitivity, TCR and the transient climate response to cumulative carbon emissions (TCRE) can be expected from emergent-constraint studies (e.g. Cox et al., 2013; Hall and Qu, 2006; Kessler and Tjiputra, 2016; Wenzel et al., 2014, 2016a). Emergent constraints provide a powerful method to not only constrain feedbacks but also help identify those processes that contribute most to uncertainty in future climate projections. They are thus also important for model development and for prioritizing future observational strategies.

Continuing research efforts will need to clarify what the most important processes for climate feedbacks are and, in general, to retain these in future models used for climate predictions on the decadal scale as well as for climate projections on multi-decadal, centennial, and longer timescales. ESMs have served as useful tools for showing us the range of possible climate system responses and the overall effect of different feedback processes. In order to fully exploit the models, observing systems need to be extended, including palaeoclimatic reconstructions from measurements. For a number of climatic and biogeochemical feedbacks, the sign and/or magnitude are not sufficiently well determined to fully assess their role for our climate projections.

**Data availability.** No data sets were used in this article. TS4

Please note the remarks at the end of the manuscript.

## Appendix A: Observational basis for feedback evaluation

Below, specific options for feedback evaluation for the various feedbacks as described in Sects. 3 and 4 are given. The general strategies for feedback evaluation involving observations are described in Sect. 5.

### A1 Evaluation of atmospheric thermodynamic feedbacks (see also Sect. 3.1)

For assessing the climate sensitivity, changes in the mean TOA radiation balance are of primary importance. Satellite observations are employed to check the ESMs' ability to correctly simulate (a) interannual variability in TOA radiation fluxes that are strongly related to ENSO variability (Loeb et al., 2012) and (b) decadal variability in TOA radiation, cloud, water vapour, and lapse rate responses (Wielicki et al., 2002). Combined use of *reanalysis data* (see glossary, for assessing the atmospheric circulation) and satellite cloud, water vapour, and TOA radiation observations CE16, as well as surface observations such as SST and sea-level pressure help to assess whether circulation and/or thermodynamically induced changes in water vapour, lapse rate, clouds, and TOA radiation fluxes simulated by GCMs both are both CE17 accurate and occur as a result of the correct model processes (Willis et al., 2004; Wong et al., 2006).

Observations and models generally agree on the magnitude of surface warming over the past few decades (of course, the occurrences of transient variability events in reality and in the model world do not usually coincide because each ESM has its own specific internal variability). Transient stalling of warming in reality in specific calendar years needs an explanation, but the occurrence of such stalling events at different points in time and to different extents in models is expected (e.g. Kay et al., 2015). Less agreement exists with respect to temperature and moisture trends in the free troposphere. Models generally show an amplification of surface warming with height in the tropical troposphere, corresponding to a reduced lapse rate (Santer et al., 2005), accompanied by a moistening of the upper tropical troposphere (Soden et al., 2005). However, observations exhibit an entire range of trends, dependent on the type of observation used and the level of post-processing and correction applied to the raw data (Christy, 1995; Mears and Wentz, 2005, 2009; Thorne et al., 2011). Recent comparisons between corrected satellite data and model estimates of tropospheric temperature trends show more consistency but stress the need for a long time series analysis ($> 17$ years) before a robust trend signal arises from the noise of internal climate variability (Santer et al., 2011).

Model evaluation of the water vapour and lapse rate feedback is generally based on the models' ability to reproduce long-term trends seen in satellite and radiosonde data. Due to the small signal-to-noise ratio of a human-induced trend compared to natural variability, the implied accuracy demand on observations is often higher than achievable, particularly for the UTT. Simulated column-integrated water vapour increases with surface temperature at $\sim 6\,\%$–$7.5\,\%\,°C^{-1}$, consistent with Clausius–Clapeyron under constant relative humidity (O'Gorman and Muller, 2010; Schneider et al., 2010) (see also CMIP3 results in Vecchi and Soden, 2007). While there are significant regional variations, such an increase is consistent with observations (McCarthy et al., 2009; Soden et al., 2005; Trenberth et al., 2005) including satellite observations from the Special Sensor Microwave Imager (SSM/I) (Wentz et al., 2007).

Another important option for evaluating short-term GCM process responses to time-varying forcing is to utilize natural forced modes in the climate system, such as the diurnal cycle (Love et al., 2011; Smith et al., 2008; Stratton and Stirling, 2012; Yang and Slingo, 2001) and the annual cycle (Hall and Qu, 2006; Klocke et al., 2011; Knutti et al., 2006). Large-scale internal modes of variability that induce time-limited, anomalous regional forcing of the climate system can also be used to evaluate GCM processes and feedback responses. Such modes include ENSO (Guilyardi et al., 2009; Neale et al., 2008), the MJO (Deng and Wu, 2010; Fu and Wang, 2009), and convectively coupled waves (Straub et al., 2010; Tulich et al., 2011). The methodology of using variability from climate modes as a proxy for climate change is not without problems, however, as the changes are subjected to different regional forcings rather than to global forcings. Therefore, these observational constraints are maybe more necessary as tests for ESMs rather than being used as proxies for global climate sensitivities.

### A2 Evaluation of cloud feedbacks (see also Sect. 3.2)

Until recently, cloud feedbacks were diagnosed through the change in CRE between a doubled $CO_2$ and a control climate run. This method has the drawback of it also including radiative effects due to changes in water vapour in the cloud-free atmosphere leading to a downward shift in the cloud feedback strength that can even make positive cloud feedbacks look negative (Soden and Held, 2006). New analysis methods like the radiative-kernel method (Zelinka et al., 2012b) do not have this drawback and additionally allow making cloud feedbacks further attributable to clouds belonging to a certain height range (low, middle, high) or to a specific cloud property (amount, height, opacity). This breakdown by cloud property and cloud level has been useful in helping to assess which cloud response mechanisms are robustly reproduced by GCMs.

Evidence for the FAT mechanism has been provided by several CRM simulation studies reproducing the increase in cloud top height with increasing surface temperature (Harrop and Hartmann, 2012; Kuang and Hartmann, 2007; Romps, 2011; Tompkins and Craig, 1998). Observational studies roughly confirm the expected cloud height changes as a re-

sponse to regional, seasonal, and interannual changes in near-troposphere temperature (Chae and Sherwood, 2010; Eitzen et al., 2009; Xu et al., 2007; Zelinka and Hartmann, 2011). However, cloud responses as deduced from regional forcings on shorter timescales may be incapable of being used as a proxy for overall cloud feedbacks as a result of global warming.

In GCMs, the large-scale mixing between the boundary layer and free troposphere occurs at the small scale through transport by parameterized shallow cumulus clouds and at the larger scale through shallow atmospheric circulations. The strength of this so-called lower-tropospheric mixing appears to vary substantially among GCMs, and differences explain about half of the variance in climate sensitivity across climate models (Sherwood et al., 2014). This suggests that (a) low-tropospheric mixing transports moisture vertically and dehydrates the low-cloud layer at a rate that increases as the climate warms and (b) that this rate of increase scales with the initial mixing strength. This allows the use of low-tropospheric mixing as an emergent constraint. The diagnosed mixing strength from reanalysis data appears to be sufficiently strong to imply a climate sensitivity of more than 3 K for a doubling of carbon dioxide (Sherwood et al., 2014).

## A3 Evaluation of fast land surface feedbacks (see also Sect. 3.3)

One option is to evaluate the overall strength of the albedo feedback using satellite observations. This approach relies on an apparent emergent constraint: snow albedo feedback strength in the context of the seasonal cycle, a measurable quantity, is highly predictive of its strength in climate change in both CMIP3 and CMIP5 ensembles. The high correlation between these two examples of the feedback is likely due to the fact that they each have a very similar geographical footprint, with similar vegetation snow-masking effects (Hall and Qu, 2006; Qu and Hall, 2007, 2014). A complementary approach is to compare the radiative forcing associated with the observed trend in albedo of Northern Hemisphere landmasses over the past decades to that simulated by climate models over the same period (Flanner et al., 2011). Any comprehensive validation of the snow albedo feedback would have to also include an analysis of model performance in the feedback's two components (snow area and snow metamorphosis, the latter being the transformation of snow grains after deposition due to various processes). Though the snow cover component is typically expected to be an order of magnitude larger than the snow metamorphosis component in CMIP3 and CMIP5 models (Qu and Hall, 2007, 2014), this may not be the case in reality and both components may also interact with each other. Accurate and long-term surface albedo data records with high temporal frequency such as those provided by the ESA GlobAlbedo project (http://www.globalbedo.org, last access: 11 June 2019) would be required for that purpose. These would need to cover the full range of vegetation types associated with seasonal snow cover, through all phases of the snow season.

The $CO_2$–stomata–water feedback has mostly been deduced from FACEs CE18 (free air $CO_2$-enrichment experiments) (Ainsworth and Rogers, 2007). Remotely sensed data on LAI can help to assess the performance of respective vegetation models (Gutman and Ignatov, 1998).

## A4 Evaluation of fast ocean feedbacks (see also Sect. 3.4)

In the last decade, significant progress has been made in establishing a network of high-quality air–sea flux buoys, such as the WHOI (Woods Hole Oceanographic Institution) Stratus buoy (Colbo and Weller, 2007), the Kuroshio Earth Observatory buoys (Cronin et al., 2008), and moorings in the Gulf Stream (Bigorre et al., 2013) and the Southern Ocean (Schulz et al., 2012). These in situ observations have provided significant insights into the air–sea interaction processes as well as having contributed to the development of improved quality flux products, for example, the NOC (UK National Oceanography Centre) in situ data set (Berry and Kent, 2011), COREv2 (Coordinated Ocean Research Experiments version 2) (Large and Yeager, 2009) and Objectively Analyzed Air-Sea heat flux (OAFlux; Yu et al., 2008) blended data sets, GSSTF (Goddard Satellite-Based Surface Turbulent Fluxes; Chou et al., 2003), and HOAPS (Hamburg Ocean Atmosphere Parameters and Fluxes from Satellite Data; Andersson et al., 2010) remote-sensing data sets. (For a more detailed overview of the flux data sets, see Josey et al., 2013.)

The last decades of the 20th century were marked by record strength ENSO events in 1982–1983 and 1997–1998. The need for a better understanding of the mechanisms involved drove the launch of the Tropical Ocean–Global Atmosphere (TOGA) programme which led to the establishment of a continuous ENSO monitoring system. An array of moorings (TAO/TRITON) in the equatorial Pacific was deployed to measure the ocean temperature down to 500 m, air temperature, humidity, winds, and ocean currents (McPhaden et al., 2010). These data, together with measurements from drifting buoys, Argo floats, tide gauges, ships, and satellite observations, provided essential insights into the ENSO phenomena (McPhaden, 1999) and helped develop ENSO theory and forecasting capabilities (Zebiak and Cane, 1987). The advancing of the ocean observational network made possible the development of ocean reanalysis products (Giese and Ray, 2011; Lübbecke and McPhaden, 2014), which have the advantage of full spatial and temporal coverage of the climate models and at the same time are constrained by the observations.

## A5 Evaluation of the sea ice feedbacks (see also Sect. 3.5)

At the present stage of research, non-linear sea ice processes such as open formation water efficiency and growth–thickness relationship can be diagnosed, but understanding how the latter processes contribute to actual climate or sea ice feedbacks requires more work and dedicated model experiments. Key variables to evaluate the ability of ESMs to simulate sea ice feedbacks are the sea ice concentration, surface albedo, and thickness. Sea ice concentration is very well derived from satellite-based passive microwave retrievals (e.g. Ivanova et al., 2014) and has been monitored on a daily basis for more than 35 years. Retrievals are insensitive to cloud cover and are thoroughly validated. Sea ice surface albedo can also be reliably derived from space-borne shortwave (Pistone et al., 2014) and infrared radiometry (Laine, 2004).

By contrast, accurate ice thickness large-scale retrievals are still challenging. There is much promise in the development of satellite-derived ice thickness products, based on several retrieval techniques and sensor types (laser or radar altimetry and microwave radar interferometry). Yet uncertainties due to the lack of reliable snow data and uncertainties in ice density (Kern et al., 2015, 2016; Zygmuntowska et al., 2014) have slowed down the scientific use of such products until recently (Kwok and Cunningham, 2016). Because of these uncertainties, the ice draft observations from upward-looking sonars on board submarines still constitute a reliable and multi-decadal source of information on Arctic ice thickness, despite uneven spatial coverage (Rothrock et al., 2008). The sole large-scale Antarctic sea ice observation data set is the ASPeCt climatology (Antarctic Sea-ice Processes and Climate, 1980–2005), based on visual observations from ships (Worby et al., 2008).

## A6 Evaluation of slow land surface physics feedbacks (see also Sect. 4.1)

Some evidence for the vegetation snow-masking feedback comes from climate simulations of the middle Holocene, ca. 6000 years before present (Ganopolski et al., 1998; Otto et al., 2011). Due to subtle changes in the Earth's orbit, northern high-latitude regions received an increased amount of radiation (Berger, 1978), which led to a northward shift in the northern tree line as shown in pollen records (Prentice et al., 1998). There are indications that last glacial inception 115 000 years ago would not have happened without feedback due to the dynamics of terrestrial ecosystems strongly enhancing the snow albedo feedback (de Noblet et al., 1996; Kageyama et al., 2004). To better understand the dynamics of snow cover in forested areas and enhance the capability of simulating the vegetation snow-masking effect, the Snow Model Intercomparison Project (SnowMIP) has been initiated (Essery et al., 2009). Loranty et al. (2014) provide an evaluation of the relationship between vegetation cover and snow surface albedo in CMIP5 models.

Evaluating the vegetation–albedo–evapotranspiration feedback requires consistent time series of land cover, rainfall, and albedo over the same time periods. Long-term and climate-quality (consistent homogeneous time series) satellite records are required for this purpose. The METEOSAT surface albedo data (Pinty et al., 2000) have been shown to be very useful for monitoring the Sahelian drought and associated rainfall feedbacks during the second half of the 20th century (Govaerts and Lattanzio, 2008; Löw and Govaerts, 2010). Loew et al. (2013) have shown the potential of using long-term records of precipitation, surface albedo, and vegetation cover to study the dynamics of the vegetation–albedo–rainfall interactions in the Sahel but also emphasized limitations with current observational data sets. CMIP5 simulations revealed a large variety in simulating the vegetation–soil-moisture–rainfall feedback and projections of droughts are still highly uncertain (Orlowsky and Seneviratne, 2013). Novel, multi-decadal satellite information allows for the monitoring of droughts and precipitation anomalies at the regional to global scale (Dorigo et al., 2014; Löw et al., 2009) and for new perspectives in quantifying the role of soil moisture changes. Another possible slow land surface albedo feedback involves the formation of very bright desert soils in North Africa that may lead to a much more pronounced reduction in rainfall than removal of low-albedo vegetation alone (Knorr and Schnitzler, 2006; Knorr et al., 2001).

## A7 Evaluation of the slow ocean feedbacks (see also Sect. 4.2)

Since 2004, the RAPID (Rapid Climate Change/Meridional Overturning Circulation and Heatflux Array) programme has been collecting records of temperature and velocity used to estimate meridional mass and heat transport along a basin-wide meridional section at 26.5° N (Cunningham et al., 2007) for monitoring AMOC changes. Based on an extended time series 2004–2012, Smeed et al. (2014) reported a decline in the AMOC strength since 2008 by 2.7 Sv. Another moored array was deployed by Meridional Overturning Variability Experiment (MOVE) in the western Atlantic at 16° N to observe the deep-water flow of the western boundary current as part of the lower limb of AMOC (Kanzow et al., 2006). Willis (2010) used satellite altimetry and Argo floats to derive the AMOC at 41° N. The author found a mean value of AMOC strength of about $\sim 15$ Sv for the 2004–2006 period and no significant trend for the 2002–2009 period. The large variability and the insufficient length of the time series limit the certainty in the observational estimates of the AMOC long-term variability. An alternative could be to use recently developed ocean reanalysis products (Balmaseda et al., 2007). Munoz et al. (2011) compared six ocean reanalysis data sets and found large differences across the products

in the mean strength of MOC (meridional overturning circulation) and meridional heat transport (MHT). The uncertainty due to different assimilation techniques and ocean models is still large.

## A8 Evaluation of land biogeochemistry feedbacks (see also Sect. 4.3)

Spatial evidence from the $CO_2$ concentration network favours the land biosphere as the origin of the greater part of the interannual atmospheric $CO_2$ variability. Key aspects that need to be better quantified are the GPP sensitivity to climate, and in particular the climate sensitivity of phenology (the timing of leaf onset and fall, which is a key control on GPP). On the data side, there are excellent decadal-scale sources available, e.g. ground-based $CO_2$ flux measurements (FLUXNET; Baldocchi et al., 2001), $CO_2$ atmospheric concentration measurements from the NOAA Earth System Research Laboratory (ESRL), and remotely sensed "greenness" measures, such as the normalized difference vegetation index (NDVI) (Reed et al., 1994), based on AVHRR (Advanced Very High Resolution Radiometer) data. The seasonal cycles of the ground-based $CO_2$ flux data can be used to differentiate between CE19 simulations of GPP in response to changes in light, temperature, and water availability (Morales et al., 2005). The seasonal cycle and interannual variability of atmospheric $CO_2$ concentration allow us to test climatic responses of respectively the temperate and tropical terrestrial carbon balance (e.g. Cadule et al., 2009). Careful evaluation of modelled atmospheric $CO_2$ variability on ENSO timescales against observations should help to better constrain the NEP response to climate anomalies and hence give more confidence in its response over longer timescales. An analysis of the $C^4MIP$ models shows a correlation between NEP sensitivity to ENSO and centennial timescales (Cox and Jones, 2008). Remote sensing provides various types of rather direct information on terrestrial carbon cycling. NDVI from AVHRR has limitations that some more recent fraction of absorbed photosynthetic active radiation (FAPAR) indices avoid, but it has the advantage of a long (> 20-year) time series. Numerous data–model comparisons have used NDVI data to evaluate model simulations of vegetation phenology and its interannual variability and trends (Piao et al., 2007, 2008). Specific extreme events recorded in these remote-sensing data, such as the 2003 European heat wave or the 2005 and 2010 Amazon drought, can also be used to test the dynamic global vegetation models (DGVMs) when driven by observed time series of climate (Reichstein et al., 2007). The emergent-constraint approach (see Sect. 3.2) can help to narrow down the bandwidth of possible climate sensitivities of land carbon $CO_2$ flux under climatic warming (Cox et al., 2013; Wenzel et al., 2014) and the fertilization effect (Wenzel et al., 2016b).

For permafrost and wetland dynamics, different tundra types (hydrological conditions) have been observed in dedicated field studies which can be used for model parameterizations (Schuur et al., 2015). Ice thickness in model soil layers can be compared with ground ice data from the Circum-Arctic Map of Permafrost and Ground-Ice Conditions (http://nsidc.org/data/ggd318.html, last access: 11 June 2019; Brown et al., 1997). Wetland coverage can be deduced from satellite measurements such as MODIS (Moderate Resolution Imaging Spectroradiometer) (Friedl et al., 2010) and the Global Lakes and Wetlands Database (Lehner and Doll, 2004). Melting permafrost induces perturbations of the local micro-topography through subsiding thermokarst formations. Meanwhile, respective changes in fine-scale topography are accessible through remote sensing (West and Plug, 2008). Recently, an emergent constraint was found for permafrost loss in relation to climate warming (Chadburn, 2017).

Fire evaluation and correlation with climatic conditions can be estimated with the active-fires data from MODIS as synthesized in the multi-annual burned area product (Tansey et al., 2008; van der Werf et al., 2006). Together with the carbon monoxide (CO) column retrievals from MOPITT (Measurements of Pollution in the Troposphere) (Yin et al., 2015), this will provide global information on fire incidence, spread, and emissions. Further CO data are available from IASI (Infrared Atmospheric Sounding Interferometer) (George et al., 2009). The Global Fire Assimilation System (GFAS) (Kaiser et al., 2012) uses fire radiative power from MODIS and has been shown to yield the best performance with respect to fire emissions estimates (period covered 2003–2016). Little direct comparison has been done between Earth system models and these fire-related products. Given the key importance of vegetation fires for atmospheric chemistry and aerosols, as well as for the carbon cycle, it is imperative that the model results be critically evaluated, exploiting the availability of remote-sensing products, though uncertainties still exist in these products (Ito and Penner, 2005; see Sect. 4.6).

## A9 Evaluation of marine biogeochemical feedbacks (see also Sect. 4.4)

The backbone of ocean biogeochemical model development and evaluation is the measurement of dissolved tracers in the ocean water column, such as dissolved inorganic carbon, alkalinity, oxygen, nutrients (primarily nitrate, phosphate, and silicic acid), as well as tracers of the ocean circulation (radiocarbon, chlorofluorocarbons). New high-quality data syntheses of three-dimensional tracer concentrations such as GLODAP (GLobal Ocean Data Analysis Project) (Key et al., 2004; Olsen et al., 2016) and CARINA (CARbon IN the Atlantic) (Key et al., 2010), surface ocean $pCO_2$ data syntheses (Bakker et al., 2016; Pfeil et al., 2013; Takahashi et al., 2009), and oceanic time series measurements (e.g. Bates et al., 2014) have provided the foundation for model validation. For marine $N_2O$ and $CH_4$ assessments, the growing MEMENTO (MarinE MethanE and NiTrous Oxide) database

is available (Bange et al., 2009; Zamora et al., 2012). Remotely sensed data sets are employed for process-based evaluation of biological carbon cycling next to three-dimensional tracer data syntheses. These include ocean colour and derived products, such as chlorophyll concentration, primary production, and plankton distribution (e.g. Sea-viewing Wide Field-of-view Sensor, SeaWiFS, and MODIS; Alvain et al., 2005; Henson et al., 2012). For a time-dependent analysis of oceanic particle fluxes through the water column and details of flux changes with depth (Berelson et al., 2007; Martin et al., 1987), sediment trap data (e.g. Honjo et al., 2008) are of high value though their accuracy is associated with a larger uncertainty range due to potential systematic measurement errors, especially for shallow traps. Feedback-relevant modifications of ecosystem functioning with climate change and increasing $CO_2$ concentrations have also been investigated in laboratory and mesocosm experiments (Engel et al., 2014; Iglesias-Rodriguez et al., 2008; Riebesell et al., 2007). Results on feedback process can in principle be transferred to Earth system models, but due to the short duration, the specific set-ups of the measurements (including local effects), and partially disagreeing outcomes, these results cannot as yet be extrapolated to the global climate system with confidence.

The excess of carbon in the oceans in comparison to preindustrial oceans without major anthropogenic perturbations of the carbon cycle generally needs to be reconstructed from modern ocean data. Respective determinations of this $C_{ant}$ are associated with method-dependent biases (e.g. Fletcher et al., 2006; Tanhua et al., 2007).

Ocean models could possibly be calibrated through case studies on the large $CO_2$ variations observed during glaciations by employing palaeoclimatic data (Archer et al., 2000; Heinze et al., 1991; Kohfeld et al., 2005; Sigman and Boyle, 2000; Watson et al., 2015), but complications due to changes in ice sheet volume and sea level should also be taken into account. In addition, more recent ocean carbon cycle variations such as the transiently reduced $CO_2$ uptake of the Southern Ocean (Landschutzer et al., 2015; Le Quéré et al., 2007; Lenton et al., 2009) and the North Atlantic (Metzl et al., 2010; Watson et al., 2009) can be used for validating ocean model components through respective hindcast simulations.

The emergent-constraint approach has been applied to link air–sea flux changes in the tropics (Wenzel et al., 2014) and the Southern Ocean (Kessler and Tjiputra, 2016) as simulated by different ESMs to observational signals (tropical temperatures) or potentially observable flux changes (Southern Ocean $CO_2$ uptake strength). Kwiatkowski (2017) found an emergent constraint on narrowing down the uncertainties in declining primary production at low latitudes. Due to the long timescales involved in oceanic processes, the emergent constraint appears to still be a challenge in narrowing down oceanic climate sensitivities.

## A10 Evaluation of aerosol feedbacks (see also Sect. 4.5)

Model evaluation of DMS necessitates a strong focus on process understanding in terms of both how marine biology responds to climate factors and how cloud properties respond to DMS emissions. The model evaluation is complicated by the fact that multiple timescales are involved. For instance, it is not just important to represent how phytoplankton responds to rapid changes in temperature or solar radiation but also how it may respond and adapt in the longer term to changes in climatic conditions and/or the availability of nutrients. It has been shown that because of the atmospheric lifetime of DMS and $SO_2$, changes in cloud properties should be expected to occur far away from the location of DMS emissions (Woodhouse et al., 2008). This means that previous attempts to evaluate models by looking at their ability to reproduce observed correlation of co-located indicators of the marine biological activity with cloud properties are fundamentally flawed. A more subtle evaluation of modelled gas–aerosol–cloud interactions is therefore required. Recent attempts to determine oceanic DMS fluxes from satellites may help to constrain the respective ocean biogeochemical models (Land et al., 2014).

For an evaluation of changes in dust mobilization, long-term data sets are available for atmospheric dust in Barbados, from satellites such as Meteosat and TOMS (Total Ozone Mapping Spectrometer), for visibility in the WMO (World Meteorological Organization) network as well as oceanic, lake, and coral palaeo-data, which can be used to assess the model ability to simulate the right level of interannual and decadal variability (Chiapello and Moulin, 2002; Mahowald et al., 2010; Prospero and Lamb, 2003; Shao et al., 2013). There is also some quantitative understanding of how dust levels from the Sahara and Sahel regions respond to climate drivers (e.g. drought and NAO) against which climate models can be compared (Chiapello et al., 2005; Ginoux et al., 2004).

Model evaluation concerning the secondary aerosol feedbacks will have to rely on process understanding and evaluation (e.g. the response of BVOC emissions to short-term meteorological factors and slow plant dynamics; see Sect. 4.6), addressing the $CO_2$ inhibition effect (Arneth et al., 2007), shift in plant functional types, or acclimation of vegetation to climate change. Carslaw et al. (2010) have estimated that the increase in SOA by 2010 may have resulted in a direct radiative forcing of $-0.04$ to $-0.24\,\mathrm{W\,m^{-2}}$, thus dampening climate warming effects from greenhouse gases. The indirect effect of SOA burden changes through aerosol–cloud interactions, and other associated changes in the atmospheric composition (ozone, $NO_x$) are highly uncertain and may alter the feedback strength and sign.

Diagnostics to evaluate aerosol cloud perturbations include cloud albedo and cloud cover, trends in surface radiation, and heating rates, which can be derived for examples of

cloud statistics, satellite aerosol fields, the Baseline Surface Radiation Network (BSRN; Ohmura et al., 1998), Aeronet (Holben et al., 1998), the Earth Radiation Budget Experiment (ERBE; Barkstrom, 1984), BC concentrations, radiative fluxes, and albedo retrievals (available, among others, from MODIS).

## A11 Evaluation of tropospheric gas-phase chemistry feedbacks (see also Sect. 4.6)

The response of OH changes to climate and respective reaction rates could be constrained from the lifetimes of halogenated gases (Montzka et al., 2011). These data seem to indicate that OH variability was less than 3 % during the last decade. The ENSO cycle could be used to correlate $O_3$ and $CH_4$ burdens with temperature (e.g. Wang et al., 2011).

The evaluation of the natural emissions feedback processes is difficult because the available observational data are either of insufficient quality to unambiguously constrain ESM simulations, or they do not provide global coverage. Fortems-Cheiney et al. (2012) attempted to constrain BVOC emissions through multi-species data assimilation, including formaldehyde retrievals from SCanning Imaging Absorption spectroMeter for Atmospheric CHartographY (SCIA-MACHY) as a potential marker substance. Unfortunately, the errors in the satellite data are too large to obtain unambiguous results. With the advent of new satellite instruments, it may eventually become possible to achieve such constraints (if simultaneous data of $CH_2O$, CO, and glyoxal with sufficiently low uncertainty become available).

There have been attempts to directly infer lightning $NO_x$ production from satellite retrievals of $NO_2$ (Beirle et al., 2006), but these need to be further developed to provide useful constrains for evaluating models (Beirle et al., 2006; Christian et al., 2003; Schumann and Huntrieser, 2007).

Airborne field campaign measurements together with chemical box models have been used to infer ozone formation and destruction rates under different pollution and weather conditions (Davis et al., 2003; DiNunno et al., 2003). One example of how such semi-empirical analyses may be used for the evaluation of global models is shown in Auvray et al. (2007).

## A12 Evaluation of stratospheric composition feedbacks (see also Sect. 4.7)

In order to simulate stratospheric ozone and water vapour distributions correctly in a model, a variety of key processes have to be represented (Eyring et al., 2005). The SPARC-CCMVal report (Stratosphere-troposphere Processes And their Role in Climate; SPARC CCMVal, 2010) summarizes a detailed process-oriented evaluation of the current generation of CCMs with observations. In addition to meteorological reanalysis data from different sources, satellite remote sensing now provides long-term data sets for model evaluation. Verti-

cal profiles and total and partial columns for various chemical species are available from a variety of satellite instruments (e.g. Atmospheric Chemistry Experiment – Fourier Transform Spectrometer, ACE-FTS; Michelson Interferometer for Passive Atmospheric Sounding, MIPAS; Aura Microwave Limb Sounder, MLS; Upper Atmosphere Research Satellite Halogen Occultation Experiment, UARS HALOE; SCIA-MACHY; Global Ozone Monitoring Experiment phases 1 and 2, GOME; Solar Backscatter Ultraviolet Instrument retrievals, SBUV; TOMS; and Ozone Monitoring Instrument, OMI). For the more specific task of evaluating new radiative feedbacks involving chemistry in CCMs, it is essential to check if and to what extent the stratospheric ozone change pattern simulated by these models for the recent past is realistic (WMO, 2014). Some of the characteristic features of this pattern, especially those related to $CO_2$-induced cooling and to SST increase (Garny et al., 2011), evidently persist into the future (Bekki et al., 2013; Lin et al., 2009; Meul et al., 2014). An important point is that radiative feedbacks strongly depend on details in the three-dimensional patterns, in particular its vertical structure, because the crucial longwave radiative feedback component is dominated by ozone changes in the vicinity of the tropopause (Dietmüller et al., 2014; Marsh et al., 2016; Nowack et al., 2015). The first attempts to derive trends in the latitudinal and vertical distribution of observed ozone change were made from aircraft and radiosonde data (Poberaj et al., 2009) and by combining satellite data with a regression model (Cionni et al., 2011; Hassler et al., 2013, 2018). As the number and the length of these observed time series increase, the combination of analysis methods will become more appealing (Ball et al., 2017) and advanced analysis methods (e.g. fingerprint techniques) may soon be designed to enable the separation of forcing- and feedback-related pattern components.

Considering the dominating influence of BDC changes on the chemically induced radiative feedback in CCM simulations, the evaluation of the consistent BDC intensification simulated by current climate models under global warming (Butchart et al., 2006; Garcia and Randel, 2008; McLandress and Shepherd, 2009; Oberlander et al., 2013) is of utmost importance. Available observations (Bönisch et al., 2011; Engel et al., 2009; Hegglin and Shepherd, 2009; Stiller et al., 2012) are not in agreement with each other. Yet, it has been argued that the perceived disagreement may be inconclusive as the observed time series are too short or too inhomogeneous, and because analysis techniques are partly inconsistent (Bunzel and Schmidt, 2013; Butchart, 2014; Garcia et al., 2011). BDC and its changes cannot be directly measured but have to be diagnosed indirectly through measuring the age of stratospheric air by analysing appropriate tracers. The age of air, however, is determined both by transport and mixing (Garny et al., 2014; Ploeger et al., 2015) and thus is difficult to quantify.

Next in importance for global radiative feedback analysis in CCMs is the evaluation of the interaction among changes

in tropical upwelling, ozone in the lower tropical stratosphere, tropical CPT temperature, and stratospheric water vapour. Respective coupling is obvious in CCMs, but as long as these models have problems in reproducing the variability of stratospheric water vapour changes over the last 50 years (including the drop after 2001), the validity of model projections in stratospheric water vapour will remain questionable. Trends in the dehydration of air entering the stratosphere via the tropical pipe are largely controlled by temperature changes at the tropical CPT (Dessler et al., 2013). Links to SST changes (Rosenlof and Reid, 2008) and to BDC changes (Randel et al., 2006) have been revealed by observation analysis. CCMs qualitatively capture these effects (Austin and Reichler, 2008; Kim et al., 2013), but inter-model deviations remain quantitatively considerable. An essential requirement to make them easier to interpret is the continuous evaluation and improvement of radiation parameterizations in the CCMs, particularly their quality to simulate the temperature response to ozone changes and stratospheric water vapour itself (Forster et al., 2011; Maycock and Shine, 2012). Joint evaluation of stratospheric water vapour and temperature is a specific point of importance, as stratospheric temperature adjustment forms a decisive part of stratospheric water vapour radiative feedback (Banerjee et al., 2019; Maycock et al., 2014).

Separation between slow feedbacks and rapid adjustments, which is a common method for physical feedbacks (Geoffroy et al., 2014; Vial et al., 2013; Zelinka et al., 2013) is not easily applicable to stratospheric water vapour and ozone changes, due to difficulties in establishing a sufficient statistically significant signal from the natural variability (Forster et al., 2016). Tentative results from Nowack et al. (2015) suggest that at least the clear-sky component of the ozone radiative feedback is dominated by slow (SST-driven) processes.

# Appendix B: Glossary

*Biological carbon pump*. Biota extract inorganically dissolved carbon together with nutrients from the ocean surface through photosynthesis to produce particulate organic carbon (living organic biomass). Thus, the surface ocean $CO_2$ partial pressure is reduced. After the death of organisms, particulate matter sinks through the water column and gets remineralized back to inorganically dissolved carbon and nutrients (while a fraction reaches the sediment surface). This vertical redistribution is called the biological carbon pump. Upwelling and mixing bring carbon and nutrients back to the surface. The production of calcareous shell material acts in opposition to organic carbon and somewhat diminishes the effect of the organic carbon pump. In a steady-state ocean, continuous plankton growth acts as a partial "lid" for $CO_2$ on the sea surface. Without the action of marine biota, the pre-industrial atmospheric $CO_2$ partial pressure would have been considerably higher (by ca. 100 %).

*Brewer–Dobson circulation (BDC)*. The BDC is the meridional circulation pattern between stratosphere and troposphere, which transports ozone away from the tropics towards higher latitudes.

*Clausius–Clapeyron relation*. This relation describes the non-linear increase in saturation water vapour pressure as a function of rising air temperature for the equilibrium case.

*Climate state variables*. The variables that describe the condition of the climate system and its variability such as temperature, velocity, pressure, humidity, salinity, and greenhouse gas concentrations as functions of space and time. They are the output variables of climate models and Earth system models. In contrast, climate parameters include the coefficients of turbulent mixing, diffusion coefficients, and solubilities of gases in seawater.

*Climate variability modes*. The climate system changes with some primary spatio-temporal patterns according to system-inherent properties (such as basin length, density of water and air, and gravity acceleration) when stimulated in a stochastic or quasi-stochastic way and through interaction among its components. It is, therefore, often not possible to identify the cause and effect of a certain variability characteristic or occurrence. Mathematically, one can identify those spatial patterns of changes in climate state variables (e.g. the sea surface temperature) that contribute most to a given variability signal (often through analysis using empirical orthogonal functions, EOFs) and a respective time series (principal component). These dominant modes are the climate variability modes. Important examples are the El Niño–Southern Oscillation, ENSO; the North Atlantic Oscillation, NAO; the Pacific Decadal Oscillation, PDO; the Southern Annular Mode, SAM; the Madden–Julian Oscillation, MJO; and the Quasi-Biennial Oscillation, QBO.

*Cold-point tropopause*. The cold-point tropopause (CPT) is defined as the coldest altitude level, which represents the thermal boundary between the stratosphere and troposphere in the tropics.

*Convection*. Vertical movement of air or water (often in contrast to advection, which is horizontal movement).

*El Niño–Southern Oscillation (ENSO)*. ENSO is the most important climate variability mode. It involves the equatorial Pacific Ocean and changes in the Walker circulation. In a positive ENSO event, the trade winds weaken, thus reducing upwelling and affecting the equatorial current system in the ocean. The results are anomalously high sea surface temperatures in the equatorial Pacific Ocean. ENSO has regional effects on climate and links with extratropical climate variability through teleconnections. El Niño is the positive phase of ENSO; La Niña is its negative phase.

*Flux adjustment*. Correction term applied to component models in a coupled model framework to prevent unrealistic model drift. When using the flux adjustment method, the component models essentially run as if in uncoupled mode and only anomalies are exchanged between the components. Most current Earth system models and coupled atmosphere–

ocean general circulation models no longer employ flux adjustments.

*General circulation model (GCM)*. A GCM is a global numerical model of the atmosphere (AGCM) or ocean (OGCM) based on a discretization in space (grid with grid points) as well as time (time stepping) and the prognostic Navier–Stokes equations, which represent the hydrodynamic expression of Newton's second law on the rotating Earth. Coupled GCMs of the ocean and atmosphere are called AOGCMs; ocean GCMs that include biogeochemical components such as a representation of the inorganic and organic carbon cycles are termed biogeochemical ocean GCMs or BOGCMs.

*Jet (or jet stream)*. Jets are relatively narrow bands of very strong winds just below the tropopause (9–16 km altitude) due to the temperature difference between warm tropical air masses and cold polar air masses. Jets have a strong effect on the movement of weather systems.

*La Niña*. The negative phase of El Niño Southern–Oscillation (ENSO; see dedicated glossary entry) with anomalously cold temperatures in the equatorial Pacific Ocean and strong upwelling along the Peruvian coast.

*Mesocosms*. Experimental devices which enable the measurement of ecosystem variables in natural seawater volumes and are exposed to different forcing boundary conditions such as changing temperature and $CO_2$ partial pressure. In a mesocosm, usually natural water volumes are closed off within a large translucent plastic container or tube. Mesocosms are often fixed at a particular point where several experiments can be run in parallel. Recently, floating mesocosms have been designed which allow for deployment from research vessels.

*Model ensemble*. This is a group of models employed in identical experiments. Climate projections differ from model to model and even for one specific model if initial conditions or model formulations are slightly changed. In order to estimate the uncertainties in projections, several model systems (possibly with several realizations for each model) are employed. The resulting spread in output variables gives an indication of the uncertainty of projections with respect to model formulation and initial conditions. This is, however, not a rigorous uncertainty analysis based on mathematical theory.

*Ocean overturning circulation*. This is the large-scale pattern of oceanic circulation with downward motion at high latitudes and upward motion in upwelling areas. The concept of the global ocean conveyor belt – with young waters (with respect to their last contact with the atmosphere) descending in the northern North Atlantic, being upwelled and re-cooled in the Southern Ocean, spreading through the deep Indian and Pacific oceans, and upwelling slowly at the north Pacific before returning at the ocean surface to the North Atlantic – is reflected in oceanic tracers (such as nutrients) but does not apply to the motion of real single water parcels.

*Oxidizing capacity*. This refers to the atmosphere's rate of removing trace substances through oxidizing chemical reactions (such as the conversion of DMS to $SO_2$, $CH_4$ to CO, or CO to $CO_2$). Many of these reactions involve ozone and the hydroxyl free radical (OH).

*Plant stomata*. These are tiny openings in the leaves of plants. They act as valves for water, oxygen, and $CO_2$ exchange. Depending on the $CO_2$ requirements for assimilation, plants regulate their stomata for maximizing water use efficiency.

*Polar amplification*. Perturbations of the Earth's radiative balance lead to greater warming/cooling at high latitudes (especially in the Arctic) than at low latitudes. Among the sources of this mechanism are ice albedo feedbacks, changes in snow and sea ice cover, variations in high-latitude cloud cover, and the modes of oceanic and atmospheric poleward heat transport.

*Radiative kernel*. Algorithms in the atmospheric model component of Earth system models (ESMs) that quantify the radiative transfer within the atmosphere. The radiative kernel of one ESM can be fed with climate state variable data from different ESMs (such as humidity, lapse rate, and cloud type and position) to determine feedback strengths.

*Reanalysis data*. These are gridded databases of climate state variables with global coverage for the past decades. They are produced through data assimilation of observed data into climate models. The underlying data assimilation procedures aim to bring the time-dependent modelled values as close as possible to the time-dependent observations. In the ideal case, reanalysis data would result in similar values as the observations (at the locations and sampling times of the measurements and within the uncertainty range of the measurements) while dynamically interpolating the values at times and locations, where no observations are available.

*Representative concentration pathway (RCP)*. For the 5th IPCC assessment report four key trajectories of greenhouse gas concentration (and emission) trajectories or "pathways" were compiled as scenarios for driving model projections. These RCPs were based on high, medium, and low future emission possibilities. In the year 2100, radiative forcing values are assumed to stabilize at +2.6, +4.5, +6.0, and +8.5 Wm$^{-2}$ respectively for RCP2.6 (low emissions), RCP4.5 and RCP6.0 (medium emissions), and RCP8.5 (high emissions, "business as usual").

*Top of the atmosphere (TOA)*. This is the reference level for the comparison and computation of radiative imbalances of the atmosphere and the Earth system. The choice of the correct reference level (e.g. 30 km) is also important for consistent comparison between model and satellite data.

*Tropopause*. Boundary between troposphere and stratosphere (changes from 16 km at low latitudes to 9 km at the poles).

*Walker circulation*. This is the zonal atmospheric overturning circulation over the tropical Pacific Ocean; it includes low-level winds blowing westward across the tropical Pacific, rising air mass motion over the warm western Pacific,

returning eastward flow in the upper troposphere, and finally sinking motion over the cold eastern central Pacific Ocean.

## Appendix C

Abbreviations

| | |
|---|---|
| ACCMIP | Atmospheric Chemistry and Climate Model Intercomparison Project |
| ACE-FTS | Atmospheric Chemistry Experiment – Fourier Transform Spectrometer |
| AerChemMIP | Atmospheric Chemistry and Climate Model Intercomparison Project |
| AMOC | Atlantic meridional overturning circulation |
| AOGCM | Atmosphere–ocean general circulation model |
| ASPeCT | Antarctic Sea-ice Processes and Climate |
| AVHRR | Advanced Very High Resolution Radiometer |
| BC | Black carbon |
| BDC | Brewer–Dobson Circulation (see glossary) |
| BP | Years before present, or "before present" |
| BSRN | Baseline Surface Radiation Network |
| BVOC | Biogenic volatile organic compound |
| $C^4MIP$ | Coupled Climate–Carbon Cycle Model Intercomparison Project |
| $CaCO_3$ | Calcium carbonate |
| $C_{ant}$ | Anthropogenic carbon |
| CARINA | Carbon in the Atlantic Ocean Region |
| CCM | Chemistry–climate model |
| CCN | Cloud condensation nuclei |
| CFC | Chlorofluorocarbon |
| $CH_2O$ | Formaldehyde |
| $CH_4$ | Methane |
| CMIP | Coupled Model Intercomparison Project |
| CMIP3 | Coupled Model Intercomparison project phase 3 |
| CMIP5 | Coupled Model Intercomparison project phase 5 |
| CMIP6 | Coupled Model Intercomparison project phase 6 |
| CO | Carbon monoxide |
| $CO_2$ | Carbon dioxide |
| $CO_3^{2-}$ | Carbonate ion |
| COADS | Comprehensive Ocean–Atmosphere Data Set |
| COREv2 | Coordinated Ocean Research Experiments version 2 |
| CPT | Cold point tropopause |
| CRE | Cloud radiative effect |
| CRM | Cloud-resolving model |
| DGVM | Dynamic global vegetation model |
| DMS | Dimethylsulfide $(CH_3)_2S$ |
| ECS | Equilibrium climate sensitivity |
| EECRA | Extended Edited Cloud Report Archive |
| ENES | European Network for Earth System Modelling |
| ENSO | El Niño–Southern Oscillation (see glossary) |
| EOF | Empirical orthogonal function |
| ERBE | Earth Radiation Budget Experiment |
| ERF | Effective radiative forcing |
| ESM | Earth system model |
| ESRL | Earth System Research Laboratory |
| FAPAR | Fraction of absorbed photosynthetic active radiation |
| FAT | Fixed anvil temperature |
| FLUXNET | International network measuring terrestrial carbon, water and energy fluxes |
| GCM | General circulation model (see glossary) |
| GFAS | Global Fire Assimilation System |

Abbreviations

| | |
|---|---|
| GHG | Greenhouse gas |
| GLODAP | Global Ocean Data Analysis Project |
| GOME | Global Ozone Monitoring Experiment |
| GPP | Gross primary production |
| GSSTF | Goddard Satellite-Based Surface Turbulent Fluxes |
| $HCO_3^-$ | Bicarbonate ion |
| $HNO_3$ | Nitric acid |
| $H_2O_2$ | Hydrogen peroxide |
| HOAPS | Hamburg Ocean Atmosphere Parameters and Fluxes from Satellite Data |
| IASI | Infrared Atmospheric Sounding Interferometer |
| IPCC | Intergovernmental Panel on Climate Change |
| ISCCP | International Satellite Cloud Climatology Project |
| kyr | Thousand years |
| LAI | Leaf area index |
| LES | Large eddy simulation |
| LWCRE | Longwave cloud radiative effect |
| MEMENTO | MarinE MethanE and NiTrous Oxide database |
| METEOSAT | Geostationary meteorological satellite |
| MHT | Meridional heat transport |
| MIP | Model intercomparison project |
| MIPAS | Michelson Interferometer for Passive Atmospheric Sounding |
| MJO | Madden–Julian oscillation |
| MLS | Aura Microwave Limb Sounder |
| MOC | Meridional overturning circulation |
| MODIS | Moderate Resolution Imaging Spectroradiometer |
| MOPITT | Measurements Of Pollution In The Troposphere |
| MOVE | Meridional Overturning Variability Experiment |
| NAO | North Atlantic Oscillation |
| NCAR | National Center for Atmospheric Research |
| NDVI | Normalized difference vegetation index |
| NEP | Net ecosystem production |
| $NH_x$ | Nitrogen hydrogen compounds |
| $N_2O$ | Nitrous oxide |
| NOAA | National Oceanic and Atmospheric Administration |
| $NO_x$ | Nitrogen oxides |
| NRC | National Research Council (of the National Academies), USA |
| $O_3$ | Ozone |
| OAflux | Objectively Analyzed Air-sea Fluxes |
| OH | Hydroxyl free radical, neutral form of the hydroxide ion ($OH^-$) |
| OMI | Ozone Monitoring Instrument |
| $pCO_2$ | Carbon dioxide partial pressure |
| PDO | Pacific Decadal Oscillation |
| QBO | Quasi-Biennial Oscillation |
| PDF | Probability density function |
| PRP | Partial radiative perturbation method |
| RAPID | Rapid climate change project |
| RCP | Representative concentration pathway (see glossary) |
| RF | (Adjusted) radiative forcing |
| RFMIP | Radiative Forcing Model Intercomparison Project |
| SAM | Southern Annular Mode |
| SBUV | Solar Backscatter Ultraviolet instrument retrievals |

Abbreviations

| | |
|---|---|
| SCIAMACHY | Scanning Imaging Absorption Spectrometer for Atmospheric Chartography/Chemistry |
| SeaWiFS | Sea-viewing Wide Field-of-view Sensor |
| SnowMIP | Snow Model Intercomparison Project |
| $SO_2$ | Sulfur dioxide |
| SOA | Secondary organic aerosol |
| SPARC CCMVAL | Stratosphere-troposphere Processes And their Role in Climate, Chemistry-Climate Model Validation activity |
| SST | Sea surface temperature |
| SWCRE | Shortwave cloud radiative effect |
| TAO/TRITON | Tropical Atmosphere Ocean project / Triangle Trans-Ocean Buoy Network |
| TCR | Transient climate response |
| TCRE | Transient climate response to cumulative carbon emissions |
| THC | Global ocean thermohaline circulation |
| TOA | Top of the atmosphere (see glossary) |
| ToE | Time of emergence |
| TOGA | Tropical Ocean Global Atmosphere project |
| TOMS | Total Ozone Mapping Spectrometer |
| UARS HALOE | Upper Atmosphere Research Satellite Halogen Occultation Experiment |
| UNFCCC | United Nations Framework Convention on Climate Change |
| UTT | Upper tropical troposphere |
| VOC | Volatile organic compound |
| WETCHIMP | Wetland and Wetland $CH_4$ Intercomparison of Models Project |
| WHOI | Woods Hole Oceanographic Institution |
| WMO | World Meteorological Organization |

**Author contributions.** CH was responsible for the overall realization and structure including Table 1, Figs. 1–4, and the final design and drafting of all figures except Figs. 17–19. CH, VE, PF, and CJ are the core author group for the article as a whole. All authors contributed to specific sections corresponding to their expertise. Earlier drafts of the feedback diagrams were provided by PF and SG with input from all authors. RK, MP, and MGS provided important input across the various sections. CE20

**Competing interests.** The authors declare that they have no conflict of interest.

**Disclaimer.** This article reflects only the authors' view – the funding agencies as well as their executive agencies are not responsible for any use that may be made of the information that the article contains.

**Acknowledgements.** This work was supported through project IS-ENES2 (Infrastructure for the European Network for Earth System modelling – Phase 2; EU 7th Framework Programme, grant no. 312979, European Commission), CRESCENDO (Coordinated Research in Earth Systems and Climate: Experiments, kNowledge, Dissemination and Outreach; Horizon 2020 European Union's Framework Programme for Research and Innovation, grant no. 641816, European Commission), and EVA (Earth system modelling of climate Variations in the Anthropocene; grant no. 229771, KLIMAFORSK Programme, The Research Council of Norway). Thanks are due to the following colleagues for constructive comments and discussions: Olivier Boucher (LMD), Greg Flato (Canadian Centre for Climate Modelling and Analysis), Eric Guilyardi (IPSL/LOCEAN), Chris Jones (UK MetOffice), Darryn Waugh (John Hopkins University), and Thomas Toniazzo (NORCE Norwegian Research Centre). Language editing was carried out by Seth Pyenson. We would like to thank the three anonymous referees, the handling editor, and the editorial support team for their thorough work on this paper.

**Financial support.** This research has been supported by the European Commission (Infrastructure for the European Network for Earth System modelling – Phase 2, grant no. 312979, IS-ENES2), the European Commission, Horizon 2020 European Union's Framework Programme for Research and Innovation (CRESCENDO (grant no. 641816)), and the Research Council of Norway, KLIMAFORSK Programme (KLIMAFORSK Programme, Earth system modelling of climate Variations in the Anthropocene; grant no. 229771, EVA). Publication costs were supported by "Open Access funding – University of Bergen".

**Review statement.** This paper was edited by Somnath Baidya Roy and reviewed by three anonymous referees.

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

## Remarks from the language copy-editor

**CE1** Please note that only those copy-editing changes have been inserted that were necessary and grammatically correct. A large number could not be inserted because the alternative text provided was ungrammatical or because the requested change conflicted with our house standards. For your convenience, I will provide a short explanation of more general grammatical issues in this comment (numbered below) rather than inserting a comment at each instance, but please let me know if you have questions on matters I may not have addressed. Where more detailed explanations or clarifications were necessary in relation to specific sentences, I have inserted comments at the relevant places. Please also see our proofreading guidelines at https://www.earth-system-dynamics.net/for_authors/proofreading_guidelines.html and our house standards at https://www.earth-system-dynamics.net/for_authors/manuscript_preparation.html for more information. (1) In English, in noun+noun compounds, it is usually only the second noun that is pluralized. For this reason, expressions like "cloud tops feedback" are grammatically incorrect and have not been inserted into the text. (2) With regard to hyphenation, it is standard punctuation in English to hyphenate adjective+noun modifiers (e.g. "a long-distance flight"). This is why expression such as "low-cloud feedback" were hyphenated. Removing the hyphen would imply that you were referring to cloud feedback that was low rather than feedback related to low clouds. If this is the case, please let us know and the change will be inserted. The same applies to expressions such as "lower-stratosphere ozone feedback". I took this to refer to the lower stratosphere, rather than lower feedback. (3) It is our house standard to write out units when they are used in running text without a preceding value. (4) In English, we distinguish between two different types of relative clauses. Defining (or restrictive) relative clauses provide information that is essential and defines the antecedent. These are not separated from the main clause by a comma and can be introduced by "which" or "that". Non-defining (or non-restrictive) relative clauses provide additional information on the antecedent, but this information is not essential for the reader/listener to understand the main clause. These clauses are separated from the main clause by commas and the appropriate relative pronoun is "which" ("that" is not used for this type of clause). More information and some examples can be found here: https://www.lexico.com/en/grammar/relative-clauses. (5) The word "yet" is not equivalent to "but". Usage examples are given here https://www.lexico.com/en/definition/yet, but any standard dictionary should provide some helpful examples. (6) In English, the phrase "not only ... but" often requires a particular kind of syntax. When it introduces the subject of the clause, it is either preceded by "it is" or the sentence undergoes inversion. (7) The verb "to define" is not used with an infinitive. (8) Adverbs such as "also" usually go in mid-position when it is used within the sentence, that is to say before simple verbs but after forms of "to be" and between auxiliaries and the infinitive when that construction is used. (9) The correct preposition to be used with the noun "shift" is "in". (10) We do not generally redefine abbreviations within the main body of the text or within the Appendix.

**CE11** The original sentence is not correct as the "that" clause is incomplete. Is "that" perhaps a typo? If so, it could be removed and an additional comma inserted, resulting in the following: "The major feedback associated with fires is the release of additional CO2 to the atmosphere, though the effect of non-deforestation fires due to climate change may be small relative to purposeful biomass burnings, because most biomass burned in wildfires is dead plant material, so that fire simply provides an accelerated decomposition route compared to microbial decomposition." Is this what you mean? If not, please double-check and revise your original sentence.

**CE16**  Does "observations" refer to all three items (satellite cloud, water vapour and TOA radiation)? If so, the "and" is required – alternatively, semi-colons could be used to separate the list items: "*reanalysis data* (see glossary, for assessing the atmospheric circulation); satellite cloud, water vapour, and TOA radiation observations; and surface observations". If "observations" does not refer to all three things, "satellite cloud" either requires an article or should be made plural. Related to this, is "Combined use" the subject of the sentence or do "Combined use" and "surface observations" fulfil this function together? I took the latter to be the case, but your new sentence suggests this may not be correct, and in that case, "help" would also have to be changed to "helps".

## Remarks from the typesetter