# Peer review of "Climate feedbacks in the Earth system and prospects for their evaluation"

_Earth System Dynamics, 2018_

## Referee Comment (RC1) · Anonymous Referee #1 · 19 Jan 2019

General comments:

Heinze et al. provide a comprehensive review of various climate feedback processes in the Earth System Models (ESMs). Overall, this is a timely review paper, as we continue to add complexity to the ESMs which in turn make it difficult for us to understand feedback processes and quantification of uncertainty in climate projections by these models. I don't see any significant issues with the paper. At the same time, this paper discusses a broad spectrum of the feedback processes, and I don't have the expertise to evaluate some portions of the manuscript, especially those dealing with chemical and biogeochemical feedbacks. I have a couple of minor comments that the authors may consider in revising the paper. Otherwise, I recommend the paper be accepted.

Specific comments:

[Figure]

P19, section 3.2.3 Mid-latitude cloud amount feedback: This is a critical feedback process in the ESMs, and a more detailed discussion is desirable. The misrepresentation of the extratropical low-level clouds is a significant problem in most of the climate models in CMIP5. The biases in the simulation of these clouds cause increased absorption of shortwave radiation by the southern ocean which is attributed as the reason for the double ITCZ problem in those models (Hwang and Frierson, 2013).

P23, section 3.4.2 Tropical circulation response to a warming climate. The discussion of Pacific Walker Circulation (PWC) response to global warming is incomplete, as the authors present one viewpoint about this widely debated problem. The argument by Vecchi et al. (2006) that the PWC weakens in a warming climate in response to a differential rate of increase in precipitation and atmospheric humidity was later contradicted (e.g., Tokinaga et al., 2012; Sandeep et al., 2014). Tokinaga et al. (2012) have shown that the PWC variability in the 20th century was related to the changes in the east-west gradient of equatorial Pacific sea surface temperature. Sandeep et al. (2014) have shown that the PWC can even strengthen while the global convective mass flux weakens, contradicting the arguments of Vecchi et al. (2006).

References:

Hwang, Y-T and Frierson, D. M. W. (2013) Link between the double-Intertropical Convergence Zone problem and cloud biases over the Southern Ocean, PNAS, 110 (13) 4935-4940

Sandeep, S., Stordal, F., Sardeshmukh, P.D. et al. (2014) Pacific Walker Circulation variability in coupled and uncoupled climate models, Clim Dyn, 43, 103-117

Tokinaga, H, Xie S-P, Deser, C, Kosaka Y, Okumura YM (2012) Slowdown of the Walker circulation driven by tropical Indo-Pacific warming. Nature, 491(7424), 439–443

---

## Referee Comment (RC2) · Anonymous Referee #2 · 13 Mar 2019

Review of Heinze et al

This is a very well written and useful overview of the most important climate feedback processes that govern the Earth system response to an external forcing. The concept of feedbacks and its analysis via the electric circuit analogue is well explained, and all feedbacks discussed are presented in a similar schematic, which makes the discussion about the feedback mechanism good to follow. Some generic discussions were eye-opening to me (such as the notion that the choice of a reference is not straightforward at ESM timescales, where the entire system is always in transition).

In some occassions I had the impression that forcings and feedbacks work in another direction than was suggested by the authors, and have indicated so in the list of minor comments below.

[Figure]

The paper is quite long, but it is very comprehensive and therefore reads as elementary textbook material for every beginning or mature ESM developer or climate system analyst. Therefore, apart from a number of minor comments, I do support publication of this manuscript in ESD.

Minor comments ===============

- p3,l23: "prognostic": it is rarely the purpose of an ESM to make a prognosis (in the sense, an expected evolution of the climate system). There is always a lot of conditionality involved, which makes the term "projection" more appropriate

- fig 1: we do miss some elements discussed later in this paper (e.g. the vegetation feedbacks on CO2 levels)

- p8,l5: "...are not included in the concept of ECS." Is this for principal or for practical reasons?

- Eq 10: it was a bit confusing to interpret E not to be a flux but a cumulative emission in mass units, maybe explain this explicitly

- p9,l10: not sure I understand what d* is

- p9,l15: what do you mean with "respective"?

- Fig 3: unclear what the black dashed arrow on the left of the figure means, it is not explained in the caption

- p12,l8: "The lower the compatible emissions, the stronger the underlying positive carbon cycle climate feedback." This is not straightforward to me. Can you explain?

- Fig 4: I suggest to make a distinction between arrows that represent a flux and arrows that point at elements in the figure
- p13,l12: "upper": do you mean the long or the short time scale here?

- p13,l19: "the feedbacks considered" in this physical subsection, is what you mean. Other feedbacks later in the manuscript do not fall in one of these categories

- p16,l25: "As the moist adiabatic lapse rate decreases with increasing surface temperature": this is also no straightforward to me. Why is this?

- p19,l2-3: I always understood that the positive feedback only occurs when cloud top is moved to a cooler layer reducing outgoing LW. When upward motion does not lead to reaching a cooler temperatures (due to warming the entire system), this positive feedback vanishes I would say. Why is it still present?

- p19, l25: "with weaker shortwave radiation": I'm missing the essential step in this sentence, which involves reducing the ability to reflect sunlight on a bright surface

- p19, l31: "increase" is not a proper, term. Probably you mean elevation, or an increase in the thickness of the layer exceeding 0oC

- p19,l34: why do we have a negative feedback here?

- fig 7/section 3.3: I would expect to also see physical feedback of evaporation increase due to higher temperature (modulated by soil moisture availability)

- p21,l13: use K or oC throughout the paper

- p25,l2: how does thinner sea ice and its reduced insulation lead to a negative feedback?

- p25,l25: the statement on effects of roughness on turbulent fluxes could deserve a reference, as this conclusion is not without controversy

- p27,l1: "thermal effects of evaporation": Normally E increases with T, but a negative feedback via available soil water applies. Vegetation feedbacks may occur in cases of episodic droughts

- p27, l29: "in" -> "on"

- p28,l6: This is the first time that bias is mentioned. Should other feedback considerations be assessed using knowledge about the impact of bias?

- p29, l23: how do wetlands modulate the amount of precipitation received?

- fig 12+13: this is quite a busy picture. I suggest to group all boxes saying "CO2 (or CH4) warming" and reroute the feedbacks loops via that single box

- fig 12: increased biomass leading to more forest fires is a negative feedback on enhanced plant growth; I don't understand this one very well

- p31, l9: "radioactively"?

- p34, l8: "physical downward transport of surface waters": do you refer to freshwater suppression of convection? A bit unclear

- p37, l14: "as a consequence": does dust production depend on global temperature? That is not stated explicitly

- p42, l32: the negative RF of stratospheric ozone is a surprise to me: I always thought ozone is an absorber in the UV spectrum and so heats up the climate system

- p45, l20: insert "and" before "which"

- p47, l3: larger what?
- p47, l34: "which happens after 120 years": please make explicit that this time you include the carbon uptake processes

- p49, l11: the time of emergence of 30-60 yrs, does that apply to temperature?

- p49, l30: delete "among"

- section 5.2.4: not entire clear how these paleo runs can add insights on feedbacks

- p53,l25: "similarities with the real world": this statement does deserve a citation

- p54, l14: "projection parameters (e.g. resolution matrices)": unclear to me

---

## Referee Comment (RC3) · Anonymous Referee #3 · 14 Mar 2019

This paper is a comprehensive review of the state of knowledge of climate feedbacks in the Earth System. I read the document from the perspective of someone who wasn't necessarily familiar with all the details of climate feedbacks, which is the intended audience. Frankly, since there are so many feedbacks, each with their own nuances and levels of understanding, this is the appropriate perspective to have since no individual scientist can fully understand them all. So, from that perspective, I felt that the level of presentation and discussion was appropriate. I have a few minor suggestions included below. Overall, I thought that the feedback discussion section was better than the feedback evaluation section, though this likely mainly reflects the big challenges that our community faces with respect to evaluating feedbacks with limited and short record observational data.

[Figure]

The paper is well-written and is structured appropriately. I don't really have too many criticisms or suggestions as the paper was clearly put together in a thoughtful and careful way.

Minor points:

1. P. 3, line 27: replace "climatological timescales" with "climate timescales"

2. P. 5, line 18: I would add 'surface energy budget' into the list of things that LULCC affects. The changes in surface roughness between forests and grasslands/croplands is often one of the most important factors affecting the LULCC impact on climate through the impact on surface energy budget partitioning.

3. P.5, line 24: The comment about "Anthropogenic driving factors such as albedo changes from deforestation, agriculture ..." seems to be mainly repetitive to the discussion of land use change higher up in the paragraph, but it is introduced as an additional 'factor'

4. P. 13: I found the beginning of Section 3 to be a little bit confusing. Section 3 is meant to be about fast feedbacks, but then there is some discussion in the introductory material of fast versus slow feedbacks and which overall feedbacks are considered. There is also a listing of the four basic feedback types, which are a mixture of fast and slow feedbacks. Maybe there needs to be a separate introductory section about what feedbacks will be considered and not considered. I would advise the authors to consider ways to clarify the text.

5. P. 13: Similarly, the list of the four basic types of feedbacks doesn't transition cleanly/clearly into the detailed descriptions in the following sections. For example, the detailed descriptions don't start with the first basic feedback type (thermodynamic shortwave radiation feedbacks). And, the detailed feedback descriptions tend to bounce around across basic feedback types as well, with even a transition to another section to describe some of the basic feedback types. I'm not sure if it would be

straightforward to rework ordering or text to guide the reader along a little better (and also not clear how important it is), but thought I would highlight the possible issue to the authors.

6. P. 17: Line 17: "Clouds belong to the prime sources for uncertainty in". This is strange wording and could likely be improved.

7. P. 24, line 24: I believe that an increase in LAI can lead to an increase or a decrease in albedo. The direction of change depends on the underlying soil albedo. Note that the amplitude of these feedbacks described in this paragraph are highly uncertain, which could be stated, though maybe that is true of many of the feedbacks and therefore would be repetitive to state that there is high uncertainty for many feedbacks.

8. P. 20, Section 4.3.1: From my perspective, there is too much emphasis in this paragraph on the impact of permafrost thaw in increasing methane emissions. Schuur et al. (2015) emphasize that the biggest feedback from permafrost thaw is expected to be from carbon dioxide release as organic material currently frozen or nearly frozen in permafrost soils thaws and decomposes. Increased methane emissions associated with warmer and potentially wetter soils is also a permafrost carbon feedback, but it is not expected to be as large as that associated with CO2 emissions. Note also that all current estimates of the permafrost climate-carbon feedback have neglected the potentially significant emissions from abrupt permafrost thaw processes. The literature on this is essentially negligible, though, so hard to cite.

9. P. 30, line 32: CO2 fertilization is not due only to improved water use efficiency of plants. Increased CO2 uptake by plants under high CO2 conditions is due to the impacts of CO2 concentration on plant photosynthetic processes.

10. P. 31, line 5: True, but the models with CN representation in CMIP5 have been shown to have unrealistic behavior with respect to N-limitation impacts on the carbon-concentration feedback (e.g., Bonan and Levis, 2010).

[Figure]

11. P. 47, Line 3: "Suggest an even larger RANGE of equilibrium"

12. P. 45, Section 5.1.3: It would be worth citing this recent paper (McDougall et al., 2019) that discusses the limitations of 1% experiments to assess feedbacks in ESMs.

13. P. 48, line 10: I would suggest citing the recently published ILAMB paper (Collier et al., 2018) in addition to Eyring et al. (2016c) to indicate the breadth of efforts in this arena of model assessment.

14. P. 49, line 10: The text as written implies at the beginning that the ToE has a 30-60 year timescale. Clearly, as the authors note further down in the text, the ToE depends strongly on which variable and on what spatial scale is being considered. And, another paper on ToE related to carbon is Lombardozzi et al. (2014).

15. P. 51, Section 5.2.5: Seems like this paper that highlights some of the potential limitations associated with emergent constraints should be cited (Caldwell et al., 2018).

16. P. 54, line 1: Perhaps should replace the term "individual modeler" with "modeling groups". Obviously, ESMs are not developed by individuals and decisions are not made about the quality of simulations by individual modelers either.

Lombardozzi, Danica, Gordon B. Bonan, and Douglas W. Nychka. "The emerging anthropogenic signal in land–atmosphere carbon-cycle coupling." Nature Climate Change 4.9 (2014): 796.

Bonan, Gordon B., and Samuel Levis. "Quantifying carbon–nitrogen feedbacks in the Community Land Model (CLM4)." Geophysical Research Letters 37.7 (2010).

MacDougall, Andrew Hugh. "Limitations of the 1% experiment as the benchmark idealized experiment for carbon cycle intercomparison in C 4 MIP." Geoscientific Model Development 12.2 (2019): 597-611.

Collier, Nathan, et al. "The International Land Model Benchmarking (ILAMB) system: design, theory, and implementation." Journal of Advances in Modeling Earth Systems

10.11 (2018): 2731-2754.

Caldwell, Peter M., Mark D. Zelinka, and Stephen A. Klein. "Evaluating emergent constraints on equilibrium climate sensitivity." Journal of Climate 31.10 (2018): 3921-3942.
* * *

---

## Author Comment (AC2) · 10 Apr 2019

**Author response to referee comments for manuscript ESD-2018-84**
**TITLE: *Climate feedbacks in the Earth system and prospects for their evaluation**

BY AUTHORS: Christoph Heinze, Veronika Eyring, Pierre Friedlingstein, Colin Jones, Yves Balkanski, Williams Collins, Thierry Fichefet, Shuang Gao, Alex Hall, Detelina Ivanova, Wolfgang Knorr, Reto Knutti, Alexander Löw, Michael Ponater, Martin G. Schultz, Michael Schulz, Pier Siebesma, Joao Teixeira, George Tselioudis, and Martin Vancoppenolle

**RESPONSE TO REVIEWER #2**

!! ALL PAGE/LINE NUMBERS REFER TO THE DISCUSSION PAPER AS PUBLISHED ON THE ESD DISCUSSIONS WEBSITE !!

*REVIEWER General comment:*
*This is a very well written and useful overview of the most important climate feedback processes that govern the Earth system response to an external forcing. The concept of feedbacks and its analysis via the electric circuit analogue is well explained, and all feedbacks discussed are presented in a similar schematic, which makes the discussion about the feedback mechanism good to follow. Some generic discussions were eye opening to me (such as the notion that the choice of a reference is not straightforward at ESM timescales, where the entire system is always in transition). In some occasions I had the impression that forcings and feedbacks work in another direction than was suggested by the authors, and have indicated so in the list of minor comments below.*
*The paper is quite long, but it is very comprehensive and therefore reads as elementary textbook material for every beginning or mature ESM developer or climate system analyst. Therefore, apart from a number of minor comments, I do support publication of this manuscript in ESD.*

OUR RESPONSE:
We would like to thank reviewer#2 for the constructive comments.

*REVIEWER Minor comment:*
*p3, l23: "prognostic": it is rarely the purpose of an ESM to make a prognosis (in the sense, an expected evolution of the climate system). There is always a lot of conditionality involved, which makes the term "projection" more appropriate*

OUR RESPONSE:
We changed the sentence:
"Such complex model simulations reveal prevailing deficiencies in our prognostic capability of the full Earth system that need to be overcome."
To:
"Such complex model simulations reveal prevailing deficiencies in our capability to project the evolution of the full Earth system. These deficiencies need to be overcome."

*REVIEWER Minor comment:*
*fig 1: we do miss some elements discussed later in this paper (e.g. the vegetation feedbacks on $CO_2$ levels)*

OUR RESPONSE:
We will add vegetation feedbacks to increasing $CO_2$ levels, the fire feedback, and the soil moisture evapotranspiration feedback to Figure 1.

*REVIEWER Minor comment:*
*p8, l5: "...are not included in the concept of ECS." Is this for principal or for practical reasons?*

OUR RESPONSE:
This is for principal reasons. A world with perpetual constant twofold atmospheric $CO_2$ concentration as compared to the pre-industrial situation cannot be realised when including processes that alter the atmospheric $CO_2$ concentration themselves.
We will change:
"…and slow feedback-like changes in vegetation types and ice sheets are not included in the concept of equilibrium climate sensitivity (Knutti and Hegerl, 2008; Knutti and Rugenstein, 2015), which was mainly developed for physical climate models…"
To:
"…and slow feedback-like changes in vegetation types and ice sheets are underlined deliberately not included in the concept of equilibrium climate sensitivity (Knutti and Hegerl, 2008; Knutti and Rugenstein, 2015), which was developed mainly to inter-compare the performance of physical climate models…"

*REVIEWER Minor comment:*
*Eq 10: it was a bit confusing to interpret E not to be a flux but a cumulative emission in mass units, maybe explain this explicitly*

OUR RESPONSE:
We will add the text "(cumulated emissions of $CO_2$ since beginning of industrialisation)" in the description of E.

*REVIEWER Minor comment:*
*p9, l10: not sure I understand what d\* is*

OUR RESPONSE:
The coefficient d\* is explained in line 15 (page 9). We will add a "(see below)" after the first time d\* gets mentioned in line 10. Line: 15: "includes" will be replaced by "represents". We will add the text "(d\* is the combination of modules in ESMs that convert greenhouse gas concentration changes into surface air temperature changes)" at line 16.

*REVIEWER Minor comment:*
*p9, l15: what do you mean with "respective"?*

OUR RESPONSE:
We will delete the word "respective".

*REVIEWER Minor comment:*
*Fig 3: unclear what the black dashed arrow on the left of the figure means, it is not explained in the caption*

OUR RESPONSE:
We will change the black dashed arrow so that it goes from the lower right corner (*CO2 emissions*) to the left upper corner (*Radiative forcing*). We will add the following text to the caption of Figure 3: "The dashed black arrow illustrates that the $CO_2$ emissions are initially the same for the chemical and the radiative forcing."

*REVIEWER Minor comment:*
*p12, l8: "The lower the compatible emissions, the stronger the underlying positive carbon cycle climate feedback." This is not straightforward to me. Can you explain?*

OUR RESPONSE:
We will add the following text:
"This is illustrated by the following definition of compatible emissions in a model projection framework using prescribed atmospheric $CO_2$ (see also Box 6.4 in Cias et al., 2013):

$$Emissions_{compatible} = \left(\frac{dCO_2}{dt}\right)^{prescribed}_{atmosphere} + (carbon\ uptake)_{land} + (carbon\ uptake)_{ocean}$$

For a projection with increasing carbon uptake by land and ocean under rising atmospheric $CO_2$ concentrations, high compatible emissions would result. In contrast, for a projection with decreasing carbon uptake by land and ocean under rising atmospheric $CO_2$ concentrations, the compatible emissions would be smaller."

*REVIEWER Minor comment:*
*Fig 4: I suggest to make a distinction between arrows that represent a flux and arrows that point at elements in the figure*

OUR RESPONSE:
We will change the (now) black arrows in Figure 4 in order to make the distinction between fluxes and description of elements.

*REVIEWER Minor comment:*
*p13, l12: "upper": do you mean the long or the short time scale here?*

OUR RESPONSE:
We will change the following text:
"… where the upper end of the timescale spectrum …"
To:
"… where the upper end of the timescale (few years) …"

*REVIEWER Minor comment:*
*p13, l19: "the feedbacks considered" in this physical subsection, is what you mean. Other feedbacks later in the manuscript do not fall in one of these categories*

OUR RESPONSE:
We will change the following text:
"The feedbacks considered (regardless of whether they are fast or slow) can be grouped into four basic types:…"
To:
"The physical feedbacks considered (regardless of whether they are fast or slow) can be grouped into four basic types: …"

*REVIEWER Minor comment:*
*p16, l25: "As the moist adiabatic lapse rate decreases with increasing surface temperature": this is also no straightforward to me. Why is this?*

OUR RESPONSE:
We will add the following text on page 16, line 29, in order to clarify this point:
"Especially in tropical regions, a stronger warming of the troposphere as compared to the surface occurs under increased greenhouse gas concentrations in the atmosphere. This effect results in a negative feedback to climate due to an increase in thermal emission to space (Boucher et al., 2013; Bony et al., 2006)."

*REVIEWER Minor comment:*
*p19, l2-3: I always understood that the positive feedback only occurs when cloud top is moved to a cooler layer reducing outgoing LW. When upward motion does not lead to reaching a cooler temperatures (due to warming the entire system), this positive feedback vanishes I would say. Why is it still present?*

OUR RESPONSE:

We will change on page 19, lines 2-3 the text "positive longwave radiative feedback" to "positive cloud longwave feedback".

We will add the following text on page 19, line 3: "The clouds are not warming synchronously with the surface temperature. Therefore, the warming Tropics become less efficient at radiating away heat. As a consequence, the clouds induce a positive feedback to climate (Zelinka and Hartmann, 2011)."

(The reference is already included in the discussion paper.)

*REVIEWER Minor comment:*

*p19, l25: "with weaker shortwave radiation": I'm missing the essential step in this sentence, which involves reducing the ability to reflect sunlight on a bright surface*

OUR RESPONSE:

We will change the phrase (page 19, line 25): "These shifts of clouds to higher latitudes with weaker shortwave radiation induce a positive feedback of an uncertain amount…"

To:

"These shifts of optically thick storm clouds to higher latitudes with weaker incoming solar radiation makes them less efficient radiation reflectors and thus induces a positive feedback of an uncertain amount…".

Further we will change the section heading "3.2.3 Mid-latitude cloud amount feedback" to "3.2.3 Mid-latitude cloud amount feedback" to "3.2.3 Mid-latitude cloud reflectance feedback". This change will also be made in Figure 6. In Figure 6, we will in addition change "polar cloud amount" to "mid-latitude cloud reflectance".

*REVIEWER Minor comment:*

*p19, l31: "increase" is not a proper, term. Probably you mean elevation, or an increase in the thickness of the layer exceeding 0°C*

OUR RESPONSE:

We will change "increase in" to "elevation of".

*REVIEWER Minor comment:*

*p19,l34: why do we have a negative feedback here?*

OUR RESPONSE:

We will add the following text to explain this better:  "Due to the larger reflectivity of liquid water clouds over ice clouds (cloud cover and water mass unchanged) a change from ice to liquid clouds must induce a negative (shortwave) cloud radiative feedback (Tan et al., 2016)."

We will add the reference:

Tan, I., Storelvmo, T., Zelinka, M. D. : Observational constraints on mixed phase clouds imply higher climate sensitivity, Science 352, 224-227, doi:10.1126/science.aad5300, 2016.

*REVIEWER Minor comment:*

*fig 7/section 3.3: I would expect to also see physical feedback of evaporation increase due to higher temperature (modulated by soil moisture availability)*

OUR RESPONSE:

We will include the positive soil moisture evapotranspiration feedback. Respective additions will be made to Figure 1, Figure 7, and Table 1.

The text on page 21, lines 1-5, will be changed to: "The most important fast land surface feedbacks are that of snow albedo, the positive soil moisture evapotranspiration feedback and the positive $CO_2$–stomata–water feedback (see feedback diagrams in Figure 7)."

The headline section 3.3.2 will be changed to: "Soil moisture evapotranspiration feedback and $CO_2$-stomata-water feedback".

Before the sentence on page 21, line 19, the following text will be inserted: "Warming leads to an increase of evaporation from soils. This negative soil moisture anomaly leads to a positive surface temperature anomaly through the reduction in latent heat flux (Senerivatne et al., 2010). The result is a positive feedback. Next to this physical feedback a chemically forced feedback exists."
We will add the following reference:
Seneviratne, S. I., Corti, T., Davin, E. L., Hirschi, M., Jaeger, E. B., Lehner, I., Orlowsky, B., and Teuling, A. J.: Investigating soil moisture-climate interactions in a changing climate: A review, Earth-Sci Rev, 99, 125-161, 10.1016/j.earscirev.2010.02.004, 2010.

*REVIEWER Minor comment:*
*p21, l13: use K or °C throughout the paper*

OUR RESPONSE:
We will convert in all unit descriptions °C$^{-1}$ instead of K$^{-1}$. For general descriptions of temperatures we will use °C, but for temperature differences K. A respective homogenisation of the manuscript will be carried out.

*REVIEWER Minor comment:*
*p25, l2: how does thinner sea ice and its reduced insulation lead to a negative feedback?*

OUR RESPONSE:
This is due to the increased heat loss of ocean water (which can lead to buoyancy driven convection and water column overturning), see also Notz and Marotzke (2012) (cited on page 24, line 19). We will add the following text on page 35, line 5: "The increased open water fracture in the thinning ice promotes new ice growth during the winter season which increases the ice covered area insulating the ocean below, and thus acting as a negative feedback. The negative feedback may be counteracted, however, by the effect of brine release. The ice brine released during the ice formation destabilizes the strongly stratified Arctic ocean water column below and induces convection which then may entrain warmer waters from below contradicting the new ice production. Please see and add a new reference (Goosse et al., 2018)."
We will add the reference:
Goosse, H., Kay, J. E., Armour, K. C., Bodas-Salcedo, A., Chepfer, H., Docquier, D., Jonko, A., Kushner, P. J., Lecomte, O., Massonnet, F., Park, H. S., Pithan, F., Svensson, G., and Vancoppenolle, M.: Quantifying climate feedbacks in polar regions, Nat Commun, 9, ARTN 1919, 10.1038/s41467-018-04173-0, 2018.

*REVIEWER Minor comment:*
*p25, l25: the statement on effects of roughness on turbulent fluxes could deserve a reference, as this conclusion is not without controversy*

OUR RESPONSE:
We will add the following reference on page 25, line 26:
Gustafsson, D., Lewan, E., and Jansson, P. E.: Modeling water and heat balance of the boreal landscape - comparison of forest and arable land in Scandinavia, J Appl Meteorol, 43, 1750-1767, Doi 10.1175/Jam2163.1, 2004.

*REVIEWER Minor comment:*
*p27, l1: "thermal effects of evaporation": Normally E increases with T, but a negative feedback via available soil water applies. Vegetation feedbacks may occur in cases of episodic droughts*

OUR RESPONSE:
We will add on page 27, line 27, after "thermal effects induced by evapotranspiration": "(changes in latent and sensible heat fluxes modulated by soil moisture availability and prevailing vegetation)".

*REVIEWER Minor comment:*
*p27, l29: "in" -> "on"*

OUR RESPONSE:
This will be corrected.

*REVIEWER Minor comment:*
*p28, l6: This is the first time that bias is mentioned. Should other feedback considerations be assessed using knowledge about the impact of bias?*

OUR RESPONSE:
We think, that the implications for other feedbacks due to this bias cannot as yet be assessed due to lacking knowledge. We will introduce a "potentially" before "complex implications" on page 28, line 7.

*REVIEWER Minor comment:*
*p29, l23: how do wetlands modulate the amount of precipitation received?*

OUR RESPONSE:
It was not our intention to state that wetlands modulate precipitation. In order to avoid such a misunderstanding, we will change the sentence on page 29, lines 22-24, from:
"Further, in a warmer world, permafrost areas may shrink and more wetlands may appear due to this process or develop in already humid regions, which may receive additional precipitation."
To:
"Further, areas that experience additional precipitation in response to warming, permafrost areas may shrink and more wetlands may appear due to this process or develop in already humid regions. This additional precipitation in response to warming is projected for many regions (Collins et al., 2013)."
The reference is already included in the discussion paper.

*REVIEWER Minor comment:*
*fig 12+13: this is quite a busy picture. I suggest to group all boxes saying "$CO_2$ (or $CH_4$) warming" and reroute the feedbacks loops via that single box*

OUR RESPONSE:
We prefer to have each single feedback be represented separately (also due to sometimes different sign) and think that the simplification with such a single box would make the figures more difficult to understand.

*REVIEWER Minor comment:*
*fig 12: increased biomass leading to more forest fires is a negative feedback on enhanced plant growth; I don't understand this one very well*

OUR RESPONSE:
We do not understand the comment. In Figure 12, the biomass limited fire feedback is leading to more $CO_2$ release and hence provides a positive feedback to the atmospheric $CO_2$ concentration.

*REVIEWER Minor comment:*
*p31, l9: "radioactively"?*

OUR RESPONSE:
We will change "radioactively" to "radiatively".

*REVIEWER Minor comment:*

*p34, l8: "physical downward transport of surface waters": do you refer to freshwater suppression of convection? A bit unclear*

OUR RESPONSE:
We will change the text passage:
"This negative feedback (Figure 13) is expected to be considerably smaller than the feedback due to reduced physical downward transport of surface waters with high anthropogenic carbon loadings (Broecker, 1991; Maier-Reimer et al., 1996; Plattner et al., 2001). The stronger partial retention of waters with high anthropogenic $CO_2$ burdens at the sea surface will thus dominate over the more efficient biogenic downward particle flux in a more slowly overturning ocean."
To:
"This negative feedback (Figure 13) is expected to be considerably smaller than the feedback due to reduced physical downward transport of surface waters with high anthropogenic carbon loadings (Broecker, 1991; Maier-Reimer et al., 1996; Plattner et al., 2001)."

*REVIEWER Minor comment:*
*p37, l14: "as a consequence": does dust production depend on global temperature? That is not stated explicitly*

OUR RESPONSE:
We will delete: "As a consequence".

*REVIEWER Minor comment:*
*p42, l32: the negative RF of stratospheric ozone is a surprise to me: I always thought ozone is an absorber in the UV spectrum and so heats up the climate system*

OUR RESPONSE:
The referee is right that the radiative impact of ozone is a local heating. The text, however, is addressing stratospheric ozone depletion, i.e., the radiative impact *change* of an ozone decrease. This induces a cooling, both locally in the stratosphere, but also to the troposphere surface system below (for the latter effect both shortwave and longwave effects have to be considered).
We will add the following text on page 42, line 33:
"The stratospheric ozone depletion induces a cooling, both locally in the stratosphere, but also to the troposphere surface system below. For the latter, a negative radiative forcing at the tropopause originates from the longwave radiative effect induced by a lower stratosphere ozone decrease (Hansen et al., 1997)."
(The reference is already included in the discussion paper.)

*REVIEWER Minor comment:*
*p45, l20: insert "and" before "which"*

OUR RESPONSE:
We will carry out this change.

*REVIEWER Minor comment:*
*p47, l3: larger what?*

OUR RESPONSE:
We will correct this ("larger range").

*REVIEWER Minor comment:*
*p47, l34: "which happens after 120 years": please make explicit that this time you include the carbon uptake processes*

OUR RESPONSE:
We think that this should be clear already (because we write "including interactive carbon cycle").

*REVIEWER Minor comment:*
*p49, l11: the time of emergence of 30-60 yrs, does that apply to temperature?*

OUR RESPONSE:
We will insert "for surface air temperature" after "30–60 years".

*REVIEWER Minor comment:*
*p49, l30: delete "among"*

OUR RESPONSE:
We will correct this.

*REVIEWER Minor comment:*
*section 5.2.4: not entire clear how these paleo runs can add insights on feedbacks*

OUR RESPONSE:
We will revise section 5.2.4. We will include a remark on the general idea on how one can use past changes in forcing to assess the response of ESMs through comparison of model results with palaeo-climatic time series data. We will give a few more examples (next to the ones cited already in the discussion paper) to illustrate how palaoe-climatic model runs can help in quantifying feedbacks and climate sensitivity. We will change section 5.2.4 by rearranging existing text and adding text:
"Palaeoclimatic experiments with ESMs can be useful for assessing the models' ability to account for slow feedbacks and for constraining sensitivity of models to forcings in general. The general concept is to expose ESMs to reconstructed anomalies in forcing, to diagnose the models' response, and to compare the model results with palaeoclimatic observations. Model forcings for respective experiments are taken from orbital parameter variations of the Earth (eccentricity, axial tilt, precession; Berger and Loutre (1991)), solar activity indices, volcanic eruption records, and different ice sheet topographies. Typical test events for simulations with ESMs include the last glacial maximum (LGM, 21 kyr BP, important for quantifying the positive carbon cycle climate feedback) (Braconnot et al., 2007a; Braconnot et al., 2007b; Frank et al., 2010;Schmidt et al., 2014) and the last 1000 years including the Maunder minimum (300 yr BP, "little ice age" mechanisms) (Ottera et al., 2010; Zorita et al., 2005). Observational data used in comparison with ESM results are based on the marine and terrestrial palaeoclimate record (such as stable carbon and oxygen isotopes from sediment core analysis, pollen analysis, bore hole temperatures etc.; see, e.g. Bradley (1999)). Palaeo-climatic observations consist of proxies, i.e. preserved environmental characteristics that replace direct measurements of the instrumental record. These proxy records contain a climate signal, but embedded in a suite of other influences of non-climatic origin (Bradley, 1999). Specific links between proxy records and climate state variables rely on respective empirical transfer functions. Proxy data are therefore associated with a considerable uncertainty range. This deficiency is to some degree compensated for by the higher signal-to-noise ratio of the respective variations in climatic state variables during certain time intervals within the Quaternary. On the other hand, modified ice sheet states and sea-level positions for dates older than a few thousand years complicate ESM simulations. Cause–effect links for changes in specific feedback processes may thus be masked by other processes. We give now a few examples for useful palaeo-climatic studies to assess feedback strengths. Frank et al. (2010) employed climatic forcing data over the past millennium with observations from ice cores in order to constrain the carbon cycle feedback to temperature changes to the lower half of the range than inferred from projections by ESMs (Friedlingstein et al., 2006). A comparison of simulations with ESMs under forcing conditions for (a) the last glacial maximum and (b) the mid-holocene provided indications for the strength of the vegetation climate feedback (inducing changes in the evapotranspiration) and the albedo feedback due to changes in snow-cover and sea ice (Braconnot et al., 2007b). The various resulting feedback strengths can be weighted through a rigorous comparison of model results and observational palaoe-climatic data following a

maximum likelihood approach. ESMs can also be used for simulating the various palaeo-climatic time windows as given in (PALEOSENS Project Members, 2012) in order to calibrate their sensitivities."
We will add the reference:
PALEOSENS Project Members: Making sense of paleoclimatic sensitivity, Nature, 491, 683-691, 10.1038/nature11574, 2012.

*REVIEWER Minor comment:*
*p53,l25: "similarities with the real world": this statement does deserve a citation*

OUR RESPONSE:
We will add the following references: Flato (2011), and Flato et al. (2013) (both are already in the reference list of the discussion paper).

*REVIEWER Minor comment:*
*p54, l14: "projection parameters (e.g. resolution matrices)": unclear to me*

OUR RESPONSE:
We will remove the text passage "include projection parameters (e.g. resolution matrices) which". It is not relevant for the non-expert in this field.

---

## Author Comment (AC3) · 10 Apr 2019

**Author response to referee comments for manuscript ESD-2018-84**
**TITLE: *Climate feedbacks in the Earth system and prospects for their evaluation**

BY AUTHORS: Christoph Heinze, Veronika Eyring, Pierre Friedlingstein, Colin Jones, Yves Balkanski, Williams Collins, Thierry Fichefet, Shuang Gao, Alex Hall, Detelina Ivanova, Wolfgang Knorr, Reto Knutti, Alexander Löw, Michael Ponater, Martin G. Schultz, Michael Schulz, Pier Siebesma, Joao Teixeira, George Tselioudis, and Martin Vancoppenolle

**RESPONSE TO REVIEWER #3**

!! ALL PAGE/LINE NUMBERS REFER TO THE DISCUSSION PAPER AS PUBLISHED ON THE ESD DISCUSSIONS WEBSITE !!

*REVIEWER: General comments:*
*This paper is a comprehensive review of the state of knowledge of climate feedbacks in the Earth System. I read the document from the perspective of someone who wasn't necessarily familiar with all the details of climate feedbacks, which is the intended audience. Frankly, since there are so many feedbacks, each with their own nuances and levels of understanding, this is the appropriate perspective to have since no individual scientist can fully understand them all. So, from that perspective, I felt that the level of presentation and discussion was appropriate. I have a few minor suggestions included below. Overall, I thought that the feedback discussion section was better than the feedback evaluation section, though this likely mainly reflects the big challenges that our community faces with respect to evaluating feedbacks with limited and short record observational data.*
*The paper is well-written and is structured appropriately. I don't really have too many criticisms or suggestions as the paper was clearly put together in a thoughtful and careful way.*

OUR RESPONSE:
We would like to thank the reviewer #3 for the constructive comments.

*REVIEWER Minor point 1.:*
*P. 3, line 27: replace "climatological timescales" with "climate timescales"*

OUR RESPONSE:
We will make this replacement.

*REVIEWER Minor point 2.:*
*P. 5, line 18: I would add 'surface energy budget' into the list of things that LULCC affects. The changes in surface roughness between forests and grasslands/croplands is often one of the most important factors affecting the LULCC impact on climate through the impact on surface energy budget partitioning.*

OUR RESPONSE:
We will include "surface energy budget" in this list.

*REVIEWER Minor point 3.:*
*P.5, line 24: The comment about "Anthropogenic driving factors such as albedo changes from deforestation, agriculture . . ." seems to be mainly repetitive to the discussion of land use change higher up in the paragraph, but it is introduced as an additional 'factor'*

OUR RESPONSE:
We will delete the "Further" at the beginning of this sentence.

*REVIEWER Minor point 4.:*
*P. 13: I found the beginning of Section 3 to be a little bit confusing. Section 3 is meant to be about fast feedbacks, but then there is some discussion in the introductory material of fast versus slow feedbacks and*

*which overall feedbacks are considered. There is also a listing of the four basic feedback types, which are a mixture of fast and slow feedbacks. Maybe there needs to be a separate introductory section about what feedbacks will be considered and not considered. I would advise the authors to consider ways to clarify the text.*

OUR RESPONSE:
We will restructure the text on page 13 lines 9-27, see the following point, and make clear which part is a general comment and which part refers to fast physical climate feedbacks only.

*REVIEWER Minor point 5.:*
*P. 13: Similarly, the list of the four basic types of feedbacks doesn't transition cleanly/clearly into the detailed descriptions in the following sections. For example, the detailed descriptions don't start with the first basic feedback type (thermodynamic shortwave radiation feedbacks). And, the detailed feedback descriptions tend to bounce around across basic feedback types as well, with even a transition to another section to describe some of the basic feedback types. I'm not sure if it would be straightforward to rework ordering or text to guide the reader along a little better (and also not clear how important it is), but thought I would highlight the possible issue to the authors.*

OUR RESPONSE:
We will restructure the text on page 13 lines 9-27. The four basic types of feedbacks refer to the columns of the right hand side of Table 1. We will make this clearer.
The text on page 13, lines 9-27 will be changed to:
"We will now describe the major feedback processes and the options that currently exist to evaluate them. The following general note may serve as a guide through this section. We first briefly summarise the fast physical feedbacks that are already part of conventional physical AOGCMs and then discuss the fast and slow Earth system feedbacks (Section 4) which have been included in climate simulations through the increasing model complexity of ESMs. The following feedbacks will not be considered in detail: (a) ice sheet feedbacks, due to their long timescale (though we will mention the freshwater release from melting glaciers and its impact on ocean circulation) and (b) socio-economic feedbacks (see van Vuuren et al. (2012)), as rigorous mechanisms to interpret these are still under development. Table 1 provides a general overview of the most important feedbacks (both short and long term). The feedbacks considered can be grouped into four basic types (see Table 1, right hand side): (1) thermodynamic shortwave radiation feedbacks (to a large degree these are the albedo feedbacks), (2) thermodynamic longwave (LW) radiation feedbacks (including dynamics of water vapour and heat redistribution through circulation, though these can also affect shortwave radiation), (3) atmospheric composition altering feedbacks due to GHGs (in addition to water vapour which is already mentioned in (2), such as $CO_2$, $CH_4$, $N_2O$, and $O_3$), and (4) atmospheric-composition-altering feedbacks involving non-GHGs and particles/droplets (such as $NO_x$ and aerosols). For each family of feedbacks described in the following sections, we provide more details on the respective observational constraints in Appendix 1.

Fast feedbacks cover a timescale of months to a few years, where the upper end of the timescale spectrum (few years) would be defined by the mixing timescale of the upper ocean down to the thermocline (of course, equilibration times with the entire deep ocean also would be longer, up to several thousand years). Fast feedbacks are key to decadal climate prediction efforts, while slow feedbacks mainly come into play after a few decades."

*REVIEWER Minor point 6.:*
*P. 17: Line 17 (should be line 18): "Clouds belong to the prime sources for uncertainty in". This is strange wording and could likely be improved.*

OUR RESPONSE:
We will change the text to:
"Limited understanding of cloud processes and difficulties in simulating cloud feedbacks belong …".

*REVIEWER Minor point 7.:*
*P. 24 (should be page 21), line 24: I believe that an increase in LAI can lead to an increase or a decrease in albedo. The direction of change depends on the underlying soil albedo. Note that the amplitude of these feedbacks described in this paragraph are highly uncertain, which could be stated, though maybe that is true of many of the feedbacks and therefore would be repetitive to state that there is high uncertainty for many feedbacks.*

OUR RESPONSE:
We will add the following sentence: "One uncertainty source associated with this feedback is the original underlying surface albedo (if the underlying albedo was is high, then the feedback would even be reversed)."

*REVIEWER Minor point 8.:*
*P. 20, Section 4.3.1: From my perspective, there is too much emphasis in this paragraph on the impact of permafrost thaw in increasing methane emissions. Schuur et al. (2015) emphasize that the biggest feedback from permafrost thaw is expected to be from carbon dioxide release as organic material currently frozen or nearly frozen in permafrost soils thaws and decomposes. Increased methane emissions associated with warmer and potentially wetter soils is also a permafrost carbon feedback, but it is not expected to be as large as that associated with $CO_2$ emissions. Note also that all current estimates of the permafrost climate-carbon feedback have neglected the potentially significant emissions from abrupt permafrost thaw processes. The literature on this is essentially negligible, though, so hard to cite.*

OUR RESPONSE:
We will insert the following sentence on page 29, line 27:" The overall quantitative partitioning of permafrost carbon release into $CO_2$ and $CH_4$ is uncertain (Ciais et al., 2013)." (The reference is already included in the discussion paper.)
We will further change the subsequent sentence to: "See the discussion in Section 4.6 for implications of increased $CH_4$ emissions to the tropospheric gas-phase chemistry."

*REVIEWER Minor point 9.:*
*P. 30 (should be page 29), line 32 (should be line 33): $CO_2$ fertilization is not due only to improved water use efficiency of plants. Increased $CO_2$ uptake by plants under high $CO_2$ conditions is due to the impacts of $CO_2$ concentration on plant photosynthetic processes.*

OUR RESPONSE:
We will add "photosynthetic processes" to Figure 5 (combine it with the text string "water use efficiency").
We will add to the sentence on page 29, line 34, the text passage: "and enhanced photosynthetic processes (Liberloo et al., 2009; Norby et al., 2005)." The reference Liberloo et al. (2009) is already included in the discussion paper. We will add the following reference:
Norby, R. J., DeLucia, E. H., Gielen, B., Calfapietra, C., Giardina, C. P., King, J. S., Ledford, J., McCarthy, H. R., Moore, D. J. P., Ceulemans, R., De Angelis, P., Finzi, A. C., Karnosky, D. F., Kubiske, M. E., Lukac, M., Pregitzer, K. S., Scarascia-Mugnozza, G. E., Schlesinger, W. H., and Oren, R.: Forest response to elevated $CO_2$ is conserved across a broad range of productivity, P Natl Acad Sci USA, 102, 18052-18056, 10.1073/pnas.0509478102, 2005.

*REVIEWER Minor point 10.:*
*P. 31, line 5: True, but the models with CN representation in CMIP5 have been shown to have unrealistic behavior with respect to N-limitation impacts on the carbon concentration feedback (e.g., Bonan and Levis, 2010).*
*Reference:*
*Bonan, Gordon B., and Samuel Levis. "Quantifying carbon-nitrogen feedbacks in the Community Land Model (CLM4)." Geophysical Research Letters 37.7 (2010).*

OUR RESPONSE:

We will add the following sentence: "The approach for C-N-coupling as applied in these models may need improvements (Bonan and Levis, 2010)." The following reference will be added:

Bonan, G. B., and Levis, S.: Quantifying carbon-nitrogen feedbacks in the Community Land Model (CLM4), Geophys Res Lett, 37, Artn L07401, 10.1029/2010gl042430, 2010.

*REVIEWER Minor point 11.:*

*P. 47, Line 3: "Suggest an even larger RANGE of equilibrium"*

OUR RESPONSE:

We will correct this.

*REVIEWER Minor point 12.:*

*P. 45 (should be page 47), Section 5.1.3: It would be worth citing this recent paper (McDougall et al., 2019) that discusses the limitations of 1% experiments to assess feedbacks in ESMs.*
*Reference:*
*MacDougall, Andrew Hugh. "Limitations of the 1% experiment as the benchmark idealized experiment for carbon cycle intercomparison in C 4 MIP." Geoscientific Model Development 12.2 (2019): 597-611.*

OUR RESPONSE:

We will change the sentence:

"For estimating feedbacks, ESM experiments are carried out under future scenario forcing (often the idealised scenario with 1% $CO_2$ $yr^{-1}$ increase in atmospheric $CO_2$ is used as model runs are short, i.e. only 70 years until atmospheric $CO_2$ concentration doubles with respect to the pre-industrial start value)."
To:
"For estimating feedbacks, ESM experiments are carried out under future scenario forcing (often the idealised scenario with 1% $CO_2$ $yr^{-1}$ increase in atmospheric $CO_2$ is used as model runs are short, i.e. only 70 years until atmospheric $CO_2$ concentration doubles with respect to the pre-industrial start value; a critical appraisal of the 1% $CO_2$ $yr^{-1}$ increase scenario is given in MacDougall, 2019)."
We will add the reference:
MacDougall, A. H.: Limitations of the 1 % experiment as the benchmark idealized experiment for carbon cycle intercomparison in (CMIP)-M-4, Geosci Model Dev, 12, 597-611, 10.5194/gmd-12-597-2019, 2019.

*REVIEWER Minor point 13.:*

*P. 48, line 10: I would suggest citing the recently published ILAMB paper (Collier et al., 2018) in addition to Eyring et al. (2016c) to indicate the breadth of efforts in this arena of model assessment.*
*Reference:*
*Collier, Nathan, et al. "The International Land Model Benchmarking (ILAMB) system: design, theory, and implementation." Journal of Advances in Modeling Earth Systems 10.11 (2018): 2731-2754.*

OUR RESPONSE:

We will add this citation and add the reference: Collier, N., Hoffman, F. M., Lawrence, D. M., Keppel-Aleks, G., Koven, C. D., Riley, W. J., Mu, M. Q., and Randerson, J. T.: The International Land Model Benchmarking (ILAMB) System: Design, Theory, and Implementation, J Adv Model Earth Sy, 10, 2731-2754, 10.1029/2018ms001354, 2018.

*REVIEWER Minor point 14.:*

*P. 49, line 10: The text as written implies at the beginning that the ToE has a 30-60 year timescale. Clearly, as the authors note further down in the text, the ToE depends strongly on which variable and on what spatial scale is being considered. And, another paper on ToE related to carbon is Lombardozzi et al. (2014).*
*Reference:*
*Lombardozzi, Danica, Gordon B. Bonan, and Douglas W. Nychka. "The emerging anthropogenic signal in land–atmosphere carbon-cycle coupling." Nature Climate Change 4.9 (2014): 796.*

OUR RESPONSE:
We will add that the ToE of 30-60 years refers to surface temperature (see also the respective comment of reviewer #2 and our respective response). We will add the following text on page 49, line 16: "ToEs for climate induced changes in land ecosystems are in the same range as for surface temperature, with some shorter ToEs in regional hot spots (Lombardozzi et al., 2014)." We will add the following reference: Lombardozzi, D., Bonan, G. B., and Nychka, D. W.: The emerging anthropogenic signal in land-atmosphere carbon-cycle coupling, Nat Clim Change, 4, 796-800, 10.1038/Nclimate2323, 2014.

*REVIEWER Minor point 15.: P. 51, Section 5.2.5: Seems like this paper that highlights some of the potential limitations associated with emergent constraints should be cited (Caldwell et al., 2018).*
*Reference:*
*Caldwell, Peter M., Mark D. Zelinka, and Stephen A. Klein. "Evaluating emergent constraints on equilibrium climate sensitivity." Journal of Climate 31.10 (2018): 3921-3942.*

OUR RESPONSE:
We will add the following sentence on page 51, line 32: "This also applies to the ensemble size of models, where caution is needed especially when using small ensembles (Caldwell et al., 2018)."
We will add the reference:
Caldwell, P. M., Zelinka, M. D., and Klein, S. A.: Evaluating Emergent Constraints on Equilibrium Climate Sensitivity, Journal of Climate, 31, 3921-3942, 10.1175/Jcli-D-17-0631.1, 2018.

*16. P. 54, line 1: Perhaps should replace the term "individual modeler" with "modelling groups". Obviously, ESMs are not developed by individuals and decisions are not made about the quality of simulations by individual modelers either.*

OUR RESPONSE:
We will carry out this change.

---

## Author Response (AR1)

**Author response to referee comments for manuscript ESD-2018-84**
**TITLE: *Climate feedbacks in the Earth system and prospects for their evaluation**

BY AUTHORS: Christoph Heinze, Veronika Eyring, Pierre Friedlingstein, Colin Jones, Yves Balkanski, Williams Collins, Thierry Fichefet, Shuang Gao, Alex Hall, Detelina Ivanova, Wolfgang Knorr, Reto Knutti, Alexander Löw, Michael Ponater, Martin G. Schultz, Michael Schulz, Pier Siebesma, Joao Teixeira, George Tselioudis, and Martin Vancoppenolle

**All page/line numbers refer to the discussion paper.**

**The revised manuscript with tracked changes marked is attached.**

**RESPONSE TO REVIEWER #1**

*REVIEWER: General comments:*
*Heinze et al. provide a comprehensive review of various climate feedback processes in the Earth System Models (ESMs). Overall, this is a timely review paper, as we continue to add complexity to the ESMs which in turn make it difficult for us to understand feedback processes and quantification of uncertainty in climate projections by these models. I don't see any significant issues with the paper. At the same time, this paper discusses a broad spectrum of the feedback processes, and I don't have the expertise to evaluate some portions of the manuscript, especially those dealing with chemical and biogeochemical feedbacks. I have a couple of minor comments that the authors may consider in revising the paper. Otherwise, I recommend the paper be accepted.*

OUR RESPONSE:
We would like to thank the reviewer #1 for the constructive comments.

*REVIEWER Specific comment:*
*P19, section 3.2.3 Mid-latitude cloud amount feedback: This is a critical feedback process in the ESMs, and a more detailed discussion is desirable. The misrepresentation of the extratropical low-level clouds is a significant problem in most of the climate models in CMIP5. The biases in the simulation of these clouds cause increased absorption of shortwave radiation by the southern ocean which is attributed as the reason for the double ITCZ problem in those models (Hwang and Frierson, 2013).*
*Reference:*
*Hwang, Y-T and Frierson, D. M. W. (2013) Link between the double-Intertropical Convergence Zone problem and cloud biases over the Southern Ocean, PNAS, 110 (13), 4935-4940.*

OUR RESPONSE:
 (a) We have changed the phrase (page 19, line 25): "These shifts of clouds to higher latitudes with weaker shortwave radiation induce a positive feedback of an uncertain amount…"
To:
"These shifts of optically thick storm clouds to higher latitudes with weaker incoming solar radiation makes them less efficient radiation reflectors and thus induces a positive feedback of an uncertain amount…". Further we have changed the section heading "3.2.3 Mid-latitude cloud amount feedback" to "3.2.3 Mid-latitude cloud reflectance feedback". This change has also been made in Figure 6. In Figure 6, we further have changed "polar cloud amount" to "mid-latitude cloud reflectance". We have added after page 19, line 29: "The misrepresentation of extratropical low-level clouds in models has also implications for the cloud water phase feedback (see following section)."
(b) We have added in section 3.2.4 (page 19, line 34):
 "Extratropical low-level clouds are often misrepresented in Earth system models contributing to the double-intertropical convergence zone problem and to short-wave radiation biases (too much heating of the Southern Ocean) (Hwang and Frierson, 2013). A correction of this bias is likely to decrease the negative cloud water phase feedback (and introduces a positive low cloud feedback that is similar in mechanism to

our tropical low cloud feedback in section 3.2.2) (Frey and Kay, 2018). This misrepresentation of extratropical low-level clouds reduces the confidence in the magnitude of the feedback."
We have added the following references:
Frey, W. R., and Kay, J. E.: The influence of extratropical cloud phase and amount feedbacks on climate sensitivity, Climate Dynamics, 50, 3097-3116, 10.1007/s00382-017-3796-5, 2018.
Hwang, Y. T., and Frierson, D. M. W.: Link between the double-Intertropical Convergence Zone problem and cloud biases over the Southern Ocean, P Natl Acad Sci USA, 110, 4935-4940, 10.1073/pnas.1213302110, 2013.

*REVIEWER Specific comment:*
*P23, section 3.4.2 Tropical circulation response to a warming climate. The discussion of Pacific Walker Circulation (PWC) response to global warming is incomplete, as the authors present one viewpoint about this widely debated problem. The argument by Vecchi et al. (2006) that the PWC weakens in a warming climate in response to a differential rate of increase in precipitation and atmospheric humidity was later contradicted (e.g., Tokinaga et al., 2012; Sandeep et al., 2014). Tokinaga et al. (2012) have shown that the PWC variability in the 20th century was related to the changes in the east-west gradient of equatorial Pacific sea surface temperature. Sandeep et al. (2014) have shown that the PWC can even strengthen while the global convective mass flux weakens, contradicting the arguments of Vecchi et al. (2006).*
*References:*
*Sandeep, S., Stordal, F., Sardeshmukh, P.D. et al. (2014) Pacific Walker Circulation variability in coupled and uncoupled climate models, Clim Dyn, 43, 103-117.*
*Tokinaga, H, Xie S-P, Deser, C, Kosaka Y, Okumura YM (2012) Slowdown of the Walker circulation driven by tropical Indo-Pacific warming. Nature, 491(7424), 439–443.*

OUR RESPONSE:
We have included these additions to the debate of tropical climate feedbacks. We have added the following text on page 23, line 23:
"Tokinaga et al. (2012), however, attribute the weakening of the Walker circulation with climate warming mainly to the ocean (SST changes)."
We have further added the following text on page 23, line 27:
"Sandeep et al. (2014) argued that SST changes during the 20$^{th}$ century warming even led to an overall strengthening of the Pacific Walker circulation (while this strengthening was to some degree compensated by variability induced by the El Niño Southern Oscillation climate variability mode)."
We have added the following references:
Sandeep, S., Stordal, F., Sardeshmukh, P. D., and Compo, G. P.: Pacific Walker Circulation variability in coupled and uncoupled climate models, Climate Dynamics, 43, 103-117, 10.1007/s00382-014-2135-3, 2014.
Tokinaga, H., Xie, S. P., Deser, C., Kosaka, Y., and Okumura, Y. M.: Slowdown of the Walker circulation driven by tropical Indo-Pacific warming, Nature, 491, 439-443, 10.1038/nature11576, 2012.

**RESPONSE TO REVIEWER #2**

*REVIEWER General comment:*
*This is a very well written and useful overview of the most important climate feedback processes that govern the Earth system response to an external forcing. The concept of feedbacks and its analysis via the electric circuit analogue is well explained, and all feedbacks discussed are presented in a similar schematic, which makes the discussion about the feedback mechanism good to follow. Some generic discussions were eye opening to me (such as the notion that the choice of a reference is not straightforward at ESM timescales, where the entire system is always in transition). In some occasions I had the impression that forcings and feedbacks work in another direction than was suggested by the authors, and have indicated so in the list of minor comments below.*
*The paper is quite long, but it is very comprehensive and therefore reads as elementary textbook material for every beginning or mature ESM developer or climate system analyst. Therefore, apart from a number of minor comments, I do support publication of this manuscript in ESD.*

OUR RESPONSE:
We would like to thank reviewer#2 for the constructive comments.

*REVIEWER Minor comment:*
*p3, l23: "prognostic": it is rarely the purpose of an ESM to make a prognosis (in the sense, an expected evolution of the climate system). There is always a lot of conditionality involved, which makes the term "projection" more appropriate*

OUR RESPONSE:
We have changed the sentence:
"Such complex model simulations reveal prevailing deficiencies in our prognostic capability of the full Earth system that need to be overcome."
To:
"Such complex model simulations reveal prevailing deficiencies in our capability to project the evolution of the full Earth system. These deficiencies need to be overcome."

*REVIEWER Minor comment:*
*fig 1: we do miss some elements discussed later in this paper (e.g. the vegetation feedbacks on $CO_2$ levels)*

OUR RESPONSE:
We have added vegetation feedbacks to increasing $CO_2$ levels, the fire feedback, and the soil moisture evapotranspiration feedback to Figure 1.

*REVIEWER Minor comment:*
*p8, l5: "...are not included in the concept of ECS." Is this for principal or for practical reasons?*

OUR RESPONSE:
This is for principal reasons. A world with perpetual constant twofold atmospheric $CO_2$ concentration as compared to the pre-industrial situation cannot be realised when including processes that alter the atmospheric $CO_2$ concentration themselves.
We have changed:
"…and slow feedback-like changes in vegetation types and ice sheets are not included in the concept of equilibrium climate sensitivity (Knutti and Hegerl, 2008; Knutti and Rugenstein, 2015), which was mainly developed for physical climate models…"
To:
"…and slow feedback-like changes in vegetation types and ice sheets are underlined{deliberately} not included in the concept of equilibrium climate sensitivity (Knutti and Hegerl, 2008; Knutti and Rugenstein, 2015), which underlined{was developed mainly to inter-compare the performance of physical climate models}…"

*REVIEWER Minor comment:*
*Eq 10: it was a bit confusing to interpret E not to be a flux but a cumulative emission in mass units, maybe explain this explicitly*

OUR RESPONSE:
We have added the text "(cumulated emissions of $CO_2$ since beginning of industrialisation)" in the description of E.

*REVIEWER Minor comment:*
*p9, l10: not sure I understand what d\* is*

OUR RESPONSE:

The coefficient d* is explained in line 15 (page 9). We have added a "(see below)" after the first time d* gets mentioned in line 10. Line: 15: "includes" has been replaced by "represents". We have added the text "(d* is the combination of modules in ESMs that convert greenhouse gas concentration changes into surface air temperature changes)" at line 16.

*REVIEWER Minor comment:*
*p9, l15: what do you mean with "respective"?*

OUR RESPONSE:
We have deleted the word "respective".

*REVIEWER Minor comment:*
*Fig 3: unclear what the black dashed arrow on the left of the figure means, it is not explained in the caption*

OUR RESPONSE:
We have changed the black dashed arrow so that it goes from the lower right corner (*CO₂ emissions*) to the left upper corner (*Radiative forcing*). We have added the following text to the caption of Figure 3: "The dashed black arrow illustrates that the $CO_2$ emissions are initially the same for the chemical and the radiative forcing."

*REVIEWER Minor comment:*
*p12, l8: "The lower the compatible emissions, the stronger the underlying positive carbon cycle climate feedback." This is not straightforward to me. Can you explain?*

OUR RESPONSE:
We have added the following text:
"This is illustrated by the following definition of compatible emissions in a model projection framework using prescribed atmospheric $CO_2$ (see also Box 6.4 in Ciais et al., 2013):
$$Emissions_{compatible} = \left(\frac{dCO_2}{dt}\right)^{prescribed}_{atmosphere} + (carbon\ uptake)_{land} + (carbon\ uptake)_{ocean}$$
For a projection with increasing carbon uptake by land and ocean under rising atmospheric $CO_2$ concentrations, high compatible emissions would result. In contrast, for a projection with decreasing carbon uptake by land and ocean under rising atmospheric $CO_2$ concentrations, the compatible emissions would be smaller."

*REVIEWER Minor comment:*
*Fig 4: I suggest to make a distinction between arrows that represent a flux and arrows that point at elements in the figure*

OUR RESPONSE:
We have changed the arrow heads of the arrows denoting the description of elements in order to discriminate them form arrows denoting fluxes.

*REVIEWER Minor comment:*
*p13, l12: "upper": do you mean the long or the short time scale here?*

OUR RESPONSE:
We have changed the following text:
"… where the upper end of the timescale spectrum …"
To:
"… where the upper end of the timescale spectrum (few years) …"

*REVIEWER Minor comment:*

*p13, l19: "the feedbacks considered" in this physical subsection, is what you mean. Other feedbacks later in the manuscript do not fall in one of these categories*

OUR RESPONSE:
We have rearranged the entire paragraph and made clear that this is a general note not only applying to the physical feedbacks.

*REVIEWER Minor comment:*
*p16, l25: "As the moist adiabatic lapse rate decreases with increasing surface temperature": this is also no straightforward to me. Why is this?*

OUR RESPONSE:
We have added the following text on page 16, line 29, in order to clarify this point:
"Especially in tropical regions, a stronger warming of the troposphere as compared to the surface occurs under increased greenhouse gas concentrations in the atmosphere. This effect results in a negative feedback to climate due to an increase in thermal emission to space (Boucher et al., 2013; Bony et al., 2006)."

*REVIEWER Minor comment:*
*p19, l2-3: I always understood that the positive feedback only occurs when cloud top is moved to a cooler layer reducing outgoing LW. When upward motion does not lead to reaching a cooler temperatures (due to warming the entire system), this positive feedback vanishes I would say. Why is it still present?*

OUR RESPONSE:
On page 19, lines 2-3, we have changed the text "positive longwave radiative feedback" to "positive cloud longwave feedback".
We have added the following text on page 19, line 3: "The clouds are not warming synchronously with the surface temperature. Therefore, the warming Tropics become less efficient at radiating away heat. As a consequence, the clouds induce a positive feedback to climate (Zelinka and Hartmann, 2011)."
(The reference is already included in the discussion paper.)

*REVIEWER Minor comment:*
*p19, l25: "with weaker shortwave radiation": I'm missing the essential step in this sentence, which involves reducing the ability to reflect sunlight on a bright surface*

OUR RESPONSE:
We have changed the phrase (page 19, line 25): "These shifts of clouds to higher latitudes with weaker shortwave radiation induce a positive feedback of an uncertain amount…"
To:
"These shifts of optically thick storm clouds to higher latitudes with weaker incoming solar radiation makes them less efficient radiation reflectors and thus induces a positive feedback of an uncertain amount…".
Further we have changed the section heading "3.2.3 Mid-latitude cloud amount feedback" to "3.2.3 Mid-latitude cloud reflectance feedback". This change has also be made in Figure 6. In Figure 6, we further have changed "polar cloud amount" to "mid-latitude cloud reflectance".

*REVIEWER Minor comment:*
*p19, l31: "increase" is not a proper, term. Probably you mean elevation, or an increase in the thickness of the layer exceeding 0°C*

OUR RESPONSE:
We have changed "increase in" to "elevation of".

*REVIEWER Minor comment:*

*p19,l34: why do we have a negative feedback here?*

OUR RESPONSE:
We have added the following text to explain this better:  "Due to the larger reflectivity of liquid water clouds over ice clouds (cloud cover and water mass unchanged) a change from ice to liquid clouds must induce a negative (shortwave) cloud radiative feedback (Tan et al., 2016)."
We have added the reference:
Tan, I., Storelvmo, T., Zelinka, M. D. : Observational constraints on mixed phase clouds imply higher climate sensitivity, Science 352, 224-227, doi:10.1126/science.aad5300, 2016.

*REVIEWER Minor comment:*
*fig 7/section 3.3: I would expect to also see physical feedback of evaporation increase due to higher temperature (modulated by soil moisture availability)*

OUR RESPONSE:
We have included the positive soil moisture evapotranspiration feedback. Respective additions have been made to Figure 1, Figure 7, and Table 1.
The text on page 21, lines 1-5, has been changed to: "The most important fast land surface feedbacks are that of snow albedo, the positive soil moisture evapotranspiration feedback and the positive $CO_2$–stomata–water feedback (see feedback diagrams in Figure 7)."
The headline section 3.3.2 has been changed to: "Soil moisture evapotranspiration feedback and $CO_2$-stomata-water feedback".
Before the sentence on page 21, line 19, the following text has been inserted: "Warming leads to an increase of evaporation from soils. This negative soil moisture anomaly leads to a positive surface temperature anomaly through the reduction in latent heat flux (Senerivatne et al., 2010). The result is a positive feedback. Next to this physical feedback a chemically forced feedback exists."
We have added the following reference:
Seneviratne, S. I., Corti, T., Davin, E. L., Hirschi, M., Jaeger, E. B., Lehner, I., Orlowsky, B., and Teuling, A. J.: Investigating soil moisture-climate interactions in a changing climate: A review, Earth-Sci Rev, 99, 125-161, 10.1016/j.earscirev.2010.02.004, 2010.

*REVIEWER Minor comment:*
*p21, l13: use K or °C throughout the paper*

OUR RESPONSE:
In all unit descriptions, we have converted $K^{-1}$ to $°C^{-1}$. For general descriptions of temperatures we use °C, but for temperature differences K. A respective homogenisation of the manuscript has been carried out.

*REVIEWER Minor comment:*
*p25, l2: how does thinner sea ice and its reduced insulation lead to a negative feedback?*

OUR RESPONSE:
This is due to the increased heat loss of ocean water (which can lead to buoyancy driven convection and water column overturning), see also Notz and Marotzke (2012) (cited on page 24, line 19).
We changed the text on page 25, lines 2-5, from:
"Third, thinner ice has less snow (Hezel et al., 2012), further decreasing its insulation power. Overall, these three mechanisms drastically (and non-linearly) increase the growth rate for thin ice (Bitz and Roe, 2004) contributing to bring sea ice back to its equilibrium thickness, in balance with radiative forcing (Tietsche et al., 2011). "
To:
"Third, thinner ice has less snow (Hezel et al., 2012), further decreasing the insulation power of the sea ice cover. Overall, these three mechanisms drastically (and non-linearly) increase the growth rate for thin ice (Bitz and Roe, 2004) contributing to rapidly bringing sea ice back to its equilibrium thickness in response to

a perturbation (Tietsche et al., 2011). In the Southern Ocean, where the stratification of the water column is weaker than in the Arctic, two competing ice-ocean feedbacks have been documented (Goosse et al., 2018). The first feedback is negative and termed ice production—entrainment feedback. It arises because brine rejection during freezing deepens the ocean mixed layer, bringing to the surface warmer water from deeper levels, melting a part of the ice initially formed and inhibiting ice production. The second feedback is positive and termed ice production—ocean heat storage feedback. It stems from the fact that anomalous sea ice production induces vertical exchanges of salt, a higher stratification, storage of heat at depth and finally lower oceanic heat fluxes that favour further ice production. "

We have added the reference:

Goosse, H., Kay, J. E., Armour, K. C., Bodas-Salcedo, A., Chepfer, H., Docquier, D., Jonko, A., Kushner, P. J., Lecomte, O., Massonnet, F., Park, H. S., Pithan, F., Svensson, G., and Vancoppenolle, M.: Quantifying climate feedbacks in polar regions, Nat Commun, 9, ARTN 1919, 10.1038/s41467-018-04173-0, 2018.

*REVIEWER Minor comment:*
*p25, l25: the statement on effects of roughness on turbulent fluxes could deserve a reference, as this conclusion is not without controversy*

OUR RESPONSE:
We have added the following reference on page 25, line 26:
Gustafsson, D., Lewan, E., and Jansson, P. E.: Modeling water and heat balance of the boreal landscape - comparison of forest and arable land in Scandinavia, J Appl Meteorol, 43, 1750-1767, Doi 10.1175/Jam2163.1, 2004.

*REVIEWER Minor comment:*
*p27, l1: "thermal effects of evaporation": Normally E increases with T, but a negative feedback via available soil water applies. Vegetation feedbacks may occur in cases of episodic droughts*

OUR RESPONSE:
We have added on page 27, line 27, after "thermal effects induced by evapotranspiration": "(changes in latent and sensible heat fluxes modulated by soil moisture availability and prevailing vegetation)".

*REVIEWER Minor comment:*
*p27, l29: "in" -> "on"*

OUR RESPONSE:
This has been corrected.

*REVIEWER Minor comment:*
*p28, l6: This is the first time that bias is mentioned. Should other feedback considerations be assessed using knowledge about the impact of bias?*

OUR RESPONSE:
We think, that the implications for other feedbacks due to this bias cannot as yet be assessed due to lacking knowledge. We have inserted "potentially" before "complex implications" on page 28, line 7.

*REVIEWER Minor comment:*
*p29, l23: how do wetlands modulate the amount of precipitation received?*

OUR RESPONSE:
It was not our intention to state that wetlands modulate precipitation. In order to avoid such a misunderstanding, we have changed the sentence on page 29, lines 22-24, from:
"Further, in a warmer world, permafrost areas may shrink and more wetlands may appear due to this process or develop in already humid regions, which may receive additional precipitation."

To:
"Further, in a warmer world, permafrost areas may shrink and more wetlands may appear due to this process or develop in already humid regions, in case that these receive additional precipitation. The latter is projected for many regions (the contrast between dry and wet regions is likely to increase under global warming; Collins et al., 2013)."
The reference is already included in the discussion paper.

*REVIEWER Minor comment:*
*fig 12+13: this is quite a busy picture. I suggest to group all boxes saying "$CO_2$ (or $CH_4$) warming" and reroute the feedbacks loops via that single box*

OUR RESPONSE:
We prefer to have each single feedback be represented separately (also due to sometimes different sign) and think that the simplification with such a single box would make the figures more difficult to understand.

*REVIEWER Minor comment:*
*fig 12: increased biomass leading to more forest fires is a negative feedback on enhanced plant growth; I don't understand this one very well*

OUR RESPONSE:
We do not understand the comment. In Figure 12, the biomass limited fire feedback is leading to more $CO_2$ release and hence provides a positive feedback to the atmospheric $CO_2$ concentration.

*REVIEWER Minor comment:*
*p31, l9: "radioactively"?*

OUR RESPONSE:
We have changed "radioactively" to "radiatively".

*REVIEWER Minor comment:*
*p34, l8: "physical downward transport of surface waters": do you refer to freshwater suppression of convection? A bit unclear*

OUR RESPONSE:
We have changed the text passage:
"This negative feedback (Figure 13) is expected to be considerably smaller than the feedback due to reduced physical downward transport of surface waters with high anthropogenic carbon loadings (Broecker, 1991; Maier-Reimer et al., 1996; Plattner et al., 2001)."
To:
"This negative feedback (Figure 13) is expected to be considerably smaller than the feedback due to reduced physical downward transport of surface waters with high anthropogenic carbon loadings (Broecker, 1991; Maier-Reimer et al., 1996; Plattner et al., 2001). The stronger partial retention of waters with high anthropogenic $CO_2$ burdens at the sea surface will thus dominate over the more efficient biogenic downward particle flux in a more slowly overturning ocean."

*REVIEWER Minor comment:*
*p37, l14: "as a consequence": does dust production depend on global temperature? That is not stated explicitly*

OUR RESPONSE:
We have deleted: "As a consequence".

*REVIEWER Minor comment:*
*p42, l32: the negative RF of stratospheric ozone is a surprise to me: I always thought ozone is an absorber in the UV spectrum and so heats up the climate system*

OUR RESPONSE:
The referee is right that the radiative impact of ozone is a local heating. The text, however, is addressing stratospheric ozone depletion, i.e., the radiative impact *change* of an ozone decrease. This induces a cooling, both locally in the stratosphere, but also to the troposphere surface system below (for the latter effect both shortwave and longwave effects have to be considered).
We have added the following text on page 42, line 33:
"The stratospheric ozone depletion induces a cooling, both locally in the stratosphere, but also to the troposphere surface system below. For the latter, a negative radiative forcing at the tropopause originates from the longwave radiative effect induced by a lower stratosphere ozone decrease (Hansen et al., 1997)."
(The reference is already included in the discussion paper.)

*REVIEWER Minor comment:*
*p45, l20: insert "and" before "which"*

OUR RESPONSE:
We have carried out this change.

*REVIEWER Minor comment:*
*p47, l3: larger what?*

OUR RESPONSE:
We have corrected this ("larger range").

*REVIEWER Minor comment:*
*p47, l34: "which happens after 120 years": please make explicit that this time you include the carbon uptake processes*

OUR RESPONSE:
We think that this should be clear already (because we write "including interactive carbon cycle").

*REVIEWER Minor comment:*
*p49, l11: the time of emergence of 30-60 yrs, does that apply to temperature?*

OUR RESPONSE:
We have inserted "for surface air temperature" after "30–60 years".

*REVIEWER Minor comment:*
*p49, l30: delete "among"*

OUR RESPONSE:
We have corrected this.

*REVIEWER Minor comment:*
*section 5.2.4: not entire clear how these paleo runs can add insights on feedbacks*

OUR RESPONSE:
We have revised section 5.2.4. We have included a remark on the general idea on how one can use past changes in forcing to assess the response of ESMs through comparison of model results with palaeo-climatic time series data. We have discussed a few more examples to illustrate how paleao-climatic model

runs can help in quantifying feedbacks and climate sensitivity. We have changed section 5.2.4 by rearranging existing text and adding text:

"Palaeoclimatic experiments with ESMs can be useful for assessing the models' ability to account for slow feedbacks and for constraining sensitivity of models to forcings in general. The general concept is to expose ESMs to reconstructed anomalies in forcing, to diagnose the models' response, and to compare the model results with palaeoclimatic observations. Model forcings for respective experiments are taken from orbital parameter variations of the Earth (eccentricity, axial tilt, precession; Berger and Loutre (1991)), solar activity indices, volcanic eruption records, and different ice sheet topographies. Typical test events for simulations with ESMs include the last glacial maximum (LGM, 21 kyr BP, important for quantifying the positive carbon cycle climate feedback) (Braconnot et al., 2007a; Braconnot et al., 2007b; Frank et al., 2010; Schmidt et al., 2014) and the last 1000 years including the Maunder minimum (300 yr BP, "little ice age" mechanisms) (Ottera et al., 2010; Zorita et al., 2005). Observational data used in comparison with ESM results are based on the marine and terrestrial palaeoclimate record (such as stable carbon and oxygen isotopes from sediment core analysis, pollen analysis, bore hole temperatures etc.; see, e.g. Bradley (1999)). Palaeo-climatic observations consist of proxies, i.e. preserved environmental characteristics that replace direct measurements of the instrumental record. These proxy records contain a climate signal, but embedded in a suite of other influences of non-climatic origin (Bradley, 1999). Specific links between proxy records and climate state variables rely on respective empirical transfer functions. Proxy data are therefore associated with a considerable uncertainty range. This deficiency is to some degree compensated for by the higher signal-to-noise ratio of the respective variations in climatic state variables during certain time intervals within the Quaternary. On the other hand, modified ice sheet states and sea-level positions for dates older than a few thousand years complicate ESM simulations. Cause–effect links for changes in specific feedback processes may thus be masked by other processes. We give now a few examples for useful palaeo-climatic studies to assess feedback strengths. Frank et al. (2010) employed climatic forcing data over the past millennium with observations from ice cores in order to constrain the carbon cycle feedback to temperature changes to the lower half of the range than inferred from projections by ESMs (Friedlingstein et al., 2006). A comparison of simulations with ESMs under forcing conditions for (a) the last glacial maximum and (b) the mid-Holocene provided indications for the strength of the vegetation climate feedback (inducing changes in the evapotranspiration) and the albedo feedback due to changes in snow-cover and sea ice (Braconnot et al., 2007b). The various resulting feedback strengths can be weighted through a rigorous comparison of model results and observational palaeoe-climatic data following a maximum likelihood approach. ESMs can also be used for simulating the various palaeo-climatic time windows as given in (PALEOSENS Project Members, 2012) in order to calibrate their sensitivities."

We have added the reference:

PALEOSENS Project Members: Making sense of paleoclimatic sensitivity, Nature, 491, 683-691, 10.1038/nature11574, 2012.

*REVIEWER Minor comment:*
*p53,l25: "similarities with the real world": this statement does deserve a citation*

OUR RESPONSE:
We have added the following references: Flato (2011), and Flato et al. (2013) (both are already in the reference list of the discussion paper).

*REVIEWER Minor comment:*
*p54, l14: "projection parameters (e.g. resolution matrices)": unclear to me*

OUR RESPONSE:
We have removed the text passage "include projection parameters (e.g. resolution matrices) which". It is not relevant for the non-expert in this field.

**RESPONSE TO REVIEWER #3**

*REVIEWER: General comments:*
*This paper is a comprehensive review of the state of knowledge of climate feedbacks in the Earth System. I read the document from the perspective of someone who wasn't necessarily familiar with all the details of climate feedbacks, which is the intended audience. Frankly, since there are so many feedbacks, each with their own nuances and levels of understanding, this is the appropriate perspective to have since no individual scientist can fully understand them all. So, from that perspective, I felt that the level of presentation and discussion was appropriate. I have a few minor suggestions included below. Overall, I thought that the feedback discussion section was better than the feedback evaluation section, though this likely mainly reflects the big challenges that our community faces with respect to evaluating feedbacks with limited and short record observational data.*
*The paper is well-written and is structured appropriately. I don't really have too many criticisms or suggestions as the paper was clearly put together in a thoughtful and careful way.*

OUR RESPONSE:
We would like to thank the reviewer #3 for the constructive comments.

*REVIEWER Minor point 1.:*
*P. 3, line 27: replace "climatological timescales" with "climate timescales"*

OUR RESPONSE:
We have made this replacement.

*REVIEWER Minor point 2.:*
*P. 5, line 18: I would add 'surface energy budget' into the list of things that LULCC affects. The changes in surface roughness between forests and grasslands/croplands is often one of the most important factors affecting the LULCC impact on climate through the impact on surface energy budget partitioning.*

OUR RESPONSE:
We have included "surface energy budget" in this list.

*REVIEWER Minor point 3.:*
*P.5, line 24: The comment about "Anthropogenic driving factors such as albedo changes from deforestation, agriculture . . ." seems to be mainly repetitive to the discussion of land use change higher up in the paragraph, but it is introduced as an additional 'factor'*

OUR RESPONSE:
We have deleted the "Further" at the beginning of this sentence.

*REVIEWER Minor point 4.:*
*P. 13: I found the beginning of Section 3 to be a little bit confusing. Section 3 is meant to be about fast feedbacks, but then there is some discussion in the introductory material of fast versus slow feedbacks and which overall feedbacks are considered. There is also a listing of the four basic feedback types, which are a mixture of fast and slow feedbacks. Maybe there needs to be a separate introductory section about what feedbacks will be considered and not considered. I would advise the authors to consider ways to clarify the text.*

OUR RESPONSE:
We have restructured the text on page 13 lines 9-27, see the following point, and have made clear which part is a general comment and which part refers to fast physical climate feedbacks only.

*REVIEWER Minor point 5.:*

*P. 13: Similarly, the list of the four basic types of feedbacks doesn't transition cleanly/clearly into the detailed descriptions in the following sections. For example, the detailed descriptions don't start with the first basic feedback type (thermodynamic shortwave radiation feedbacks). And, the detailed feedback descriptions tend to bounce around across basic feedback types as well, with even a transition to another section to describe some of the basic feedback types. I'm not sure if it would be straightforward to rework ordering or text to guide the reader along a little better (and also not clear how important it is), but thought I would highlight the possible issue to the authors.*

OUR RESPONSE:
We have restructured the text on page 13 lines 9-27. The four basic types of feedbacks refer to the columns of the right hand side of Table 1. We have made this clearer.
The text on page 13, lines 9-27 has been changed to:
"We will now describe the major feedback processes and the options that currently exist to evaluate them. The following general note may serve as a guide through this section. We first briefly summarise the fast physical feedbacks that are already part of conventional physical AOGCMs and then discuss the fast and slow Earth system feedbacks (Section 4) which have been included in climate simulations through the increasing model complexity of ESMs. The following feedbacks will not be considered in detail: (a) ice sheet feedbacks, due to their long timescale (though we will mention the freshwater release from melting glaciers and its impact on ocean circulation) and (b) socio-economic feedbacks (see van Vuuren et al. (2012)), as rigorous mechanisms to interpret these are still under development. Table 1 provides a general overview of the most important feedbacks (both short and long term). The feedbacks considered can be grouped into four basic types (see Table 1, right hand side): (1) thermodynamic shortwave radiation feedbacks (to a large degree these are the albedo feedbacks), (2) thermodynamic longwave (LW) radiation feedbacks (including dynamics of water vapour and heat redistribution through circulation, though these can also affect shortwave radiation), (3) atmospheric composition altering feedbacks due to GHGs (in addition to water vapour which is already mentioned in (2), such as $CO_2$, $CH_4$, $N_2O$, and $O_3$), and (4) atmospheric-composition-altering feedbacks involving non-GHGs and particles/droplets (such as $NO_x$ and aerosols). For each family of feedbacks described in the following sections, we provide more details on the respective observational constraints in Appendix 1.

Fast feedbacks cover a timescale of months to a few years, where the upper end of the timescale spectrum (few years) would be defined by the mixing timescale of the upper ocean down to the thermocline (of course, equilibration times with the entire deep ocean also would be longer, up to several thousand years). Fast feedbacks are key to decadal climate prediction efforts, while slow feedbacks mainly come into play after a few decades."

*REVIEWER Minor point 6.:*
*P. 17: Line 17 (should be line 18): "Clouds belong to the prime sources for uncertainty in". This is strange wording and could likely be improved.*

OUR RESPONSE:
We have changed the text to:
"Limited understanding of cloud processes and difficulties in simulating cloud feedbacks belong …".

*REVIEWER Minor point 7.:*
*P. 24 (should be page 21), line 24: I believe that an increase in LAI can lead to an increase or a decrease in albedo. The direction of change depends on the underlying soil albedo. Note that the amplitude of these feedbacks described in this paragraph are highly uncertain, which could be stated, though maybe that is true of many of the feedbacks and therefore would be repetitive to state that there is high uncertainty for many feedbacks.*

OUR RESPONSE:

We have added the following sentence: "One uncertainty source associated with this feedback is the original underlying surface albedo (if the underlying albedo was is high, then the feedback would even be reversed)."

*REVIEWER Minor point 8.:*
*P. 20, Section 4.3.1: From my perspective, there is too much emphasis in this paragraph on the impact of permafrost thaw in increasing methane emissions. Schuur et al. (2015) emphasize that the biggest feedback from permafrost thaw is expected to be from carbon dioxide release as organic material currently frozen or nearly frozen in permafrost soils thaws and decomposes. Increased methane emissions associated with warmer and potentially wetter soils is also a permafrost carbon feedback, but it is not expected to be as large as that associated with $CO_2$ emissions. Note also that all current estimates of the permafrost climate-carbon feedback have neglected the potentially significant emissions from abrupt permafrost thaw processes. The literature on this is essentially negligible, though, so hard to cite.*

OUR RESPONSE:
We have inserted the following sentence on page 29, line 27:" The overall quantitative partitioning of permafrost carbon release into $CO_2$ and $CH_4$ is uncertain (Ciais et al., 2013)." (The reference is already included in the discussion paper.)
We further have changed the subsequent sentence to: "See the discussion in Section 4.6 for implications of increased $CH_4$ emissions for the tropospheric gas-phase chemistry."

*REVIEWER Minor point 9.:*
*P. 30 (should be page 29), line 32 (should be line 33): $CO_2$ fertilization is not due only to improved water use efficiency of plants. Increased $CO_2$ uptake by plants under high $CO_2$ conditions is due to the impacts of $CO_2$ concentration on plant photosynthetic processes.*

OUR RESPONSE:
We have added "enhanced photosynthetic processes" to Figure 12 (combine it with the text string "water use efficiency").
We have added the following text to the sentence on page 29, line 34: "and enhanced photosynthetic processes (Liberloo et al., 2009; Norby et al., 2005)." The reference Liberloo et al. (2009) is already included in the discussion paper. We have added the following reference:
Norby, R. J., DeLucia, E. H., Gielen, B., Calfapietra, C., Giardina, C. P., King, J. S., Ledford, J., McCarthy, H. R., Moore, D. J. P., Ceulemans, R., De Angelis, P., Finzi, A. C., Karnosky, D. F., Kubiske, M. E., Lukac, M., Pregitzer, K. S., Scarascia-Mugnozza, G. E., Schlesinger, W. H., and Oren, R.: Forest response to elevated $CO_2$ is conserved across a broad range of productivity, P Natl Acad Sci USA, 102, 18052-18056, 10.1073/pnas.0509478102, 2005.

*REVIEWER Minor point 10.:*
*P. 31, line 5: True, but the models with CN representation in CMIP5 have been shown to have unrealistic behavior with respect to N-limitation impacts on the carbon concentration feedback (e.g., Bonan and Levis, 2010).*

C4: In several references/citations the author name "Loew, A." has been corrected to "Löw, A."

C5: We have added to the acknowledgements: "We would like to thank the three anonymous referees, the handling editor, and the editorial support team for their thorough work with on manuscript."

C6: We have asked the editorial support team (editorial@copernicus.org) to correct a small error on the manuscript web-site:
"[16]Norwegian Meteorological Ins titute, Oslo, Norway"
Needs to be corrected to
"[16]Norwegian Meteorological Institute, Oslo, Norway".

C7: In contrast to the discussion paper, we have moved Table 1 and all figures to the end of the manuscript.

C8: Page numbers have been updated in the table of contents (so that the numbers match the manuscript file without showing tracked changes; many numbers are smaller than before because the table and figures have been moved to the end of the manuscript).

C9: On page 23, line 30, the text ", however," has been deleted. On page 23, line 31, the text "However," has been deleted. The reference Goosse et al. (2018) has been added on page 23, line 33.

C10: The reference "Holland and Raphael, 2006" on page 23, line 10, has been corrected to "Holland et al., 2006" (and respective changes of the reference list have been carried out).

C11: The sentence on page 16, lines 15-17 was optimised from:
"For example, when the tropical ocean warms as a result of a $CO_2$-induced increase in downwelling LW radiation, the Clausius–Clapeyron relationship (see glossary) leads to an increase in the evaporation of water into the atmosphere (Bohren and Albrecht, 1998)."
To:
"For example, when the tropical ocean warms as a result of a $CO_2$-induced increase in downwelling LW radiation, the Clausius–Clapeyron relationship (see glossary) leads to an increased ability of the atmosphere to carry water vapour that evaporated from the ocean (Bohren and Albrecht, 1998)."

C12: On page 41, line 31, the reference "Revell et al., 2015" has been added.
This reference has been added to the reference list:
Revell, L. E., Tummon, F., Stenke, A., Sukhodolov, T., Coulon, A., Rozanov, E., Garny, H., Grewe, V., and Peter, T.: Drivers of the tropospheric ozone budget throughout the 21st century under the medium-high climate scenario RCP 6.0, Atmospheric Chemistry and Physics, 15, 5887-5902, 10.5194/acp-15-5887-2015, 2015.

C13: On page 42, line 30, we have changed the text:
"The joint radiative effect from ozone changes"
To:
"The joint radiative effect from projected ozone changes".

C14: On page 43, line 8, we have changed the text:
"the Brewer–Dobson circulation"
To:
"a Brewer–Dobson circulation".

C15: On Page 44, line, 2 the following text has been changed:
"rapid concentration adjustment"
To:
"rapid adjustment component".

C16: On page 44, line 6, the following text had been changed from:
"temperature with warming"
To:
"temperature with global warming".

C17: On page 44, line 13, the text "small and" has been deleted.

C18: Page 44, line 21 was has been changed from:
"clean distinction among, adjustment, forcing and feedback contributions at this stage."

To:
"a clean distinction between adjustment, forcing, and feedback contributions at this stage."

C19: On page 65, line 34, the reference "Banerjee et al., 2018" (that has been a manuscript under review) has been updated to the actual version "Banerjee et al., 2019".
The reference has been changed in the reference list to:
Banerjee, A., Chiodo, G., Previdi, M., Ponater, M., Conley, A.J., and Polvani, M.: Stratospheric water vapor: an important climate feedback, Climate Dynamics, in press, Doi 10.1007/s00382-019-04721-4, 2019.

C20: The name "Dietmuller" has been changed to "Dietmüller" in the respective reference (Dietmüller et al., 2014).

C21: Affiliation no. 2 for the main author has been changed from "Uni Research Climate , Bergen Norway" to "NORCE Norwegian Research Centre, Bergen, Norway", because Uni Research carried out a fusion with NORCE and has taken up this new name. Likewise, the name "Uni Research Climate" has been changed to "NORCE Norwegian Research Centre" in he acknowledgments.

C22: Table 1 has been update to achieve consistency with the other revisions made.

[revised manuscript text omitted]